# Hydrogeomorphological analysis and modelling for a comprehensive understanding of flash-flood damaging processes: The 9[th] October 2018 event in North-eastern Mallorca

Joan Estrany[1,2*], Maurici Ruiz-Pérez[1,2,3], Raphael Mutzner[4], Josep Fortesa[1,2], Beatriz Nácher-Rodríguez[5], Miquel Tomàs-Burguera[6], Julián García-Comendador[1,2], Xavier Peña[4], Adolfo Calvo-Cases[7], Francisco J. Vallés-Morán[5]

[1]Mediterranean Ecogeomorphological and Hydrological Connectivity Research Team (http://medhycon.uib.cat), Department of Geography, University of the Balearic Islands, Carretera de Valldemossa Km 7.5, 07122 Palma, Balearic Islands, Spain

[2]Institute of Agro-Environmental and Water Economy Research –INAGEA, University of the Balearic Islands, Carretera de Valldemossa Km 7.5, 07122, Palma, Balearic Islands, Spain

[3]Service of GIS and Remote Sensing, University of the Balearic Islands, 07122 Palma, Balearic Islands, Spain

[4]Hydrique Engineers (http://www.hydrique.ch), Le Mont sur Lausanne, Vaud 1052, Switzerland

[5]Universitat Politècnica de València, Camí de Vera, s/n, València, 46022, Spain

[6]Estación Experimental de Aula Dei (EEAD-CSIC), Avenida Montañana, 1005, 50059 Zaragoza, Spain

[7]Inter-University Institute for Local Development (IIDL) Department of Geography, University of Valencia, Av. Blasco Ibáñez 28, 46010, Valencia, Spain

*Correspondence to: Joan Estrany (joan.estrany@uib.cat)

**Abstract.** A flash-flood event hit the northeastern part of Mallorca Island on 9[th] October 2018, causing 13 casualties. Mallorca is prone to catastrophic flash floods acting on a scenario of deep landscape transformation caused by Mediterranean tourist resorts. As global change may exacerbate devastating flash floods, analyses of catastrophic events are crucial to support effective prevention and mitigation measures. Field-based, remote-sensing and modelling techniques were used in this study to evaluate rainfall-runoff processes at the catchment scale linked to hydrological modelling. Continuous streamflow monitoring data revealed a peak discharge of 442 $m^3$ $s^{-1}$ with an unprecedented runoff response. This exceptional behaviour triggered the natural disaster as a combination of heavy rainfall (249 mm in 10 h), karstic features and land cover disturbances in the Begura de Salma River catchment (23 $km^2$). Topography-based connectivity indices and geomorphic change detection were used as rapid post-catastrophe decision-making tools, playing a key role during the rescue searchi. These hydrogeomorphological precision techniques were combined with the Copernicus Emergency Management Service and

'ground-based' damage assessment, which showed very accurately the damage driving-factors in the village of Sant Llorenç des Cardassar. The main challenges in the future are to readapt hydrological modelling to global change scenarios, implement an early flash-flood warning system and take adaptive and resilient measures on the catchment scale.

## 1 Introduction

Flash floods are related with high-intensity precipitation, mainly of convective origin and with a restricted spatio-temporal occurrence. For this reason, they usually impact basins <1000 km$^2$ with response times of a few hours or less. These spatial and temporal dimensions of flash-flood events are directly linked to (1) geomorphometric characteristics of the catchments and (2) activation mechanisms of runoff as a combination of intense rainfall, soil moisture and soil hydraulic properties (Versini et al., 2013). Land use modification, urbanization and recurrent wildfires can alter these activation mechanisms and the potential for flash flood casualties and damages. In Europe, 40% of flood-related casualties in the period 1950–2006 are due to flash floods (Barredo, 2007). However, catastrophic flash-floods are much more frequent in some parts of the Mediterranean region than in the rest of Europe due to the interaction between geomorphology, climate, vegetation, and the warm sea surface (Cassola et al., 2016) all combining to create a flood-prone environment. The abrupt reliefs surrounding the Mediterranean Sea are very close to the coastline, shaping small and torrential catchments where the convergence of low-level atmospheric flows and the uplift of warm wet air masses drifting from the Mediterranean Sea to the coasts generate heavy downpours in very short time-spans (Gaume et al., 2009).

Characterising the response of a catchment during an extreme flash-flood event is important because it clarifies flood severity and the activating hydrological processes and their dependency on natural and anthropogenic catchment properties (Borga et al., 2007). Numerous studies have tried to determine these driving factors (Braud et al., 2014) in which geological heterogeneities associated with the presence of karst features are crucial in Mediterranean catchments (Vannier et al., 2016; Wainwright and Thornes, 2004). Likewise, flash floods are closely related to land use: the devastation of plant cover in the Mediterranean may increase the risk of flooding because bare soil leads to larger runoff coefficients (Wainwright and Thornes, 2004). However, the limited spatial and temporal scales of flash-floods make these events particularly difficult to monitor and document. In the case of rainfall monitoring, the spatial scales of the events are in general much smaller than the sampling potential offered by apparently dense rain networks (Borga et al., 2008; Amponsah et al., 2016). In the case of streamflow monitoring, there is a lack of flash flood discharge ($Q$) data from stream gauge observations (Marchi et al., 2010) although $Q$ data are crucial to obtaining representative hydrometric values and characterizing the runoff response of such extreme flash-flood events (Borga et al., 2008). As a result, further field observations and modelling studies are required in order to assess the interdependencies of flash-flood drivers and, thus, better understand and reproduce the active hydrological processes (Sofia and Nikolopoulos, 2020).

Earlier flash flood forecasting systems were based on the Flash Flood Guidance (Georgakakos, 1986), which calculated the a-priori amount of rainfall needed to trigger specific $Q$ at the outlet of a catchment, depending on prior wetness conditions. At present, semi-distributed or distributed hydrological models are more widely used for such forecasting purposes (Artinyan et al., 2016; Gourley et al., 2010; Miao et al., 2016; Nguyen et al., 2016), whilst probabilistic and ensemble modelling assess the uncertainty of flash flood forecasting systems (Hardy et al., 2016). However, the uncertainty in hydrological modelling can be large. The main source of uncertainty is related to the spatio-temporal scales of rainfall pattern. The forecasting of intense thunderstorms by numerical weather prediction systems to provide accurate rainfall information is particularly challenging (Alfieri et al., 2015; Collier, 2007). Another source of uncertainty is related to the hydrological models that often need to be calibrated in order to reduce the uncertainty of the discharge estimate and to understand the physical processes occurring during flash flood events (Adamovic et al., 2016; Segura-Beltrán et al., 2016; Vannier et al., 2016). Consequently, the reliability of data measuring surface water resources requires temporal continuity and good maintenance of the gauging stations (Fortesa et al., 2019). Flash-flood events are also conditioned by high non-linearity in the hydrological response relating to threshold effects (Braud et al., 2014).

In the Mediterranean region, the planning and management of flood hazards are hydro-sociologically crucial (Gaume et al., 2016), especially in the current global change context. Therefore, the predictability of such events remains low. It is clear that the understanding of flash floods requires an integrated scientific approach (Marchi et al., 2010), in which technological advances create opportunities to investigate simultaneously in the areas of Earth and Social Sciences (Wohl et al., 2019). Firstly, geomorphometric techniques applied to topographic surveys can be valuable planning and decision-making tools for producing flood hazard maps that represent flood-prone areas (Kalantari et al., 2017). Geomorphometry and digital terrain analysis by means of a topography-based connectivity index could also be used to simulate preferential flowpaths for high-magnitude events to determine the main erosion and deposition areas, as a tool for better and faster responses to catastrophic flooding events. In addition, post-storm assessments to capture the landform signature of the event are needed. The use of high-resolution field measurements is here critical to understanding the effects of storm on fluvial dynamics (Westoby et al., 2012) and providing input data sets for numerical modelling. These data can be difficult to obtain, as traditional post-storm survey techniques are expensive and time-consuming (Duo et al., 2018). In recent years, unmanned aerial vehicles (UAVs) have been used to improve traditional, expensive, or time-consuming mapping approaches on catchment science research. UAVs permit a rapid deployment and achieve accurate high-resolution topographic data for monitoring geomorphic changes (Estrany et al., 2019; Langhammer and Vacková, 2018).

Secondly, the concept of vulnerability relates to the predisposition of certain stakes to damage or malfunction, implying that a multitude of direct and indirect factors, often interacting in a dynamic and complex way, should be integrated in their assessment. From this point of view, vulnerability particularly relates to the damage to exposed stakes (Defossez and Leone, 2017). Estimations of the elements at risk in flash floods and the damage-driving factors are insufficiently understood in the Mediterranean Region, even though assessment of damage processes from flash floods have been assessed in several European and national projects and also publications (cf. Llasat et al., 2013; Gaume et al., 2016). Observed spatial distributions

of costs and fatalities are the result of complex interplay between different explanatory factors. Useful information may reveal the economic and social impact of floods on our societies, but its interpretation is questionable (Gaume et al., 2016). In addition, the largest public world databases on disaster events that contain flood events do not include all the catastrophic events in the Mediterranean region (Llasat et al., 2013), which underlines the need for further research. Since 2012, the Copernicus Emergency Management Service (Copernicus-EMS) evaluates the intensity and scope of the damage caused by natural disasters, human-made emergency situations and humanitarian crises throughout the world (Copernicus Emergency Management Service, 2019). To understand the damage processes of flash floods better, comprehensive damage assessment linking hydrological process dynamics and intensities to damage and loss is needed (Laudan et al., 2017). In addition, a comparison of 'ground-based' assessment and 'remote-based' Copernicus EMS may shed light on the accuracy of this rapid and helpful tool for assessing most catastrophic flash floods.

This paper aims at improving the comprehension of the hydrological and socioeconomic processes of the devastating flash floods that frequently affect Mallorca (Estrany and Grimalt, 2014; Llasat et al., 2013), a paradigmatic Mediterranean flood-prone region under intense human occupation and geologically shaped by karstic features. The study focuses on the catastrophic flash-flood event that hit the north-eastern part of the island on 9th October 2018, causing 13 casualties. This was the worst local natural disaster in decades which caused a marked interest among the scientific community, producing early studies of the event. In this way, Lorenzo-Lacruz et al. (2019) reconstructed this same flash-flood event through the application of hydrological and hydraulic simulations, with a focus on the meteorological input. The specific objectives of this study are (1) to explain the runoff response and so clarify the dependency of flood severity on catchment properties and human influence, by using the flash flood $Q$ data from a stream gauge installed in 2015 by the Mediterranean Ecogeomorphological and Hydrological Connectivity –MEDhyCON Research Group; (2) to assess the uncertainty of semi-distributed hydrological modelling in such a severe flash flood in a karstic environment; (3) to investigate the socioeconomic and territorial flood damages linked to hydrogeomorphological processes; and (4) to analyse multi-temporally high-resolution digital elevation models (HR-DEM) for detecting and measuring geomorphic changes by using a UAV and a topography-based connectivity index for rapid response in post-catastrophe search and rescue tasks.

**2 Materials and Methods**

To obtain a better understanding of the flash flood as well as of damage processes, a comprehensive and combined analysis of the meteorological synoptic situation, precipitation and discharge was developed through the analysis of the rainfall-runoff processes at small spatial scale during this extreme event. Likewise, this analysis was linked to hydrological modelling to check the internal consistency of the information gathered by instruments and to flood damage and losses at Sant Llorenç des Cardassar village. Finally, high-resolution digital elevation models (HR-DEM) were generated by LiDAR 2014 data from the Spanish National Geographic Institute and by imagery captured through a low-cost UAV just six days after the catastrophe to calculate a sediment connectivity index (IC) and measure geomorphic changes (Fig. 1).

**2.1 Study area**

Mallorca is a Mediterranean flood-prone region, historically affected by flash floods. Since the Late Middle Age, devastating flash floods have been systematically documented, particularly in Palma, the capital of the island. In this town, a catastrophic event caused ca. 5,000 deaths in 1403 (20% of its population), showing that floods are the main natural hazard in this type of environment (Petrus et al., 2018). In the rest of the island, the historical distribution patterns of human settlements were related to fluvial systems, but avoided the occupation of floodplains until the increase of urban areas in the 19[th] century during the

Industrial Revolution. However, in the second part of the 20[th] century this urban expansion became exponential, with an increased urban and tourist settlements (Pons Esteva, 2003) including in flood hazard areas.

In such a high-energy environment, Mallorca's Eastern ( 'Llevant' in Catalan) county constitutes a dramatic combination of physical and human factors that have created a flood-prone environment with a very strong coupling of climate and geomorphology to constant urban expansion since 1850, wich lead to more than 10 catastrophic flash-flood events during this

period (Estrany and Grimalt, 2014). This county consists of two main relief units (Álvaro et al., 1991): (1) The Llevant Ranges occupy the headwaters with altitude ranging from 300 to 500 m a.s.l. They consist of a series of alpine mountains and hills mainly of Jurassic limestones and dolomites and Cretaceous marls; and (2) The Marinas, a reefal Upper Miocene tabular platform composed of calcarenites, calcisiltites and terra-rossa postreef sediments affected by significant karstic processes on the eastern slopes of the Llevant Ranges. A morphometric analysis of the catchments by Estrany and Grimalt (2014) showed

differences that depended on both the width of the platform (i.e., the distance between the coast and the base of the Llevant Ranges through the Miocene platform) and the hypsometry and geological settings at the headwaters. The torreniality and clinometric variables were the ones more closely related to the geological settings. Therefore, the catchments with the highest values of torrentiality (i.e. >30; which is topographically computed as a coefficient between the number of first-order streams and catchment area, multiplied by the drainage density; cf. Strahler, 1964) have impervious materials (i.e., lower Cretaceous

and lower Miocene marls) covering approx. > 40% of these catchments. These are located at headwaters where the clinometry and connectivity between slopes and channels are the highest.

This study focuses on the Ca n'Amer River catchment (78 km$^2$), due to the storm that struck several urban areas within it, especially the village of Sant Llorenç des Cardassar. Its main headwater tributary is the Begura de Salma River (23.4 km$^2$; Fig. 2c) with altitudes ranging from 71 to 485 m.a.s.l. (Fig. 2c). The mean slope of the catchment is 16% and the length of the main

channel 9.3 km (average gradient of 3%). The lithology is mainly composed of marls intercalated with limestone (60% of the area) of the Medium-Upper Jurassic (Dogger), dolomites (22% of the area) of the Upper Triassic and Lower Jurassic, and pelagic limestone marls (14% of the area) of the Lower Cretaceous (Fig. 2d). This lithological composition determines the surface water/groundwater interaction. On the one hand, a high degree of fracturing, fissuring and karstification of limestones favours percolation through karstic aquifers. On the other hand, the imperviousness of Dogger and Cretaceous marls (74% of

the area) does not allow the percolation, enabling runoff generation. The main land use in 2012 was agriculture (58%), mostly

located in lowland areas. Forest (26%) and scrubland (17%) were predominant at headwaters. Terraced fields still occupied 10% of the catchment, although most of them were abandoned (Fig. 2e). In 1956, natural vegetation covered 21% of the catchment. This rose to 42% in 2012 due to an afforestation process of former agricultural land in the second half of the twentieth century. In combination with other factors, afforestation triggered a higher fire risk: two wildfires burned an area of 165  1.7 km$^2$: 17% in 1983 and 83% in 2011 (Balearic Forestry and Soil Conservation Service, http://xarxaforestal.caib.es; Fig. 2e). The climate of the area is classified as Mediterranean temperate sub-humid on the Emberger scale (Guijarro, 1986). Mean annual rainfall (1968-2018, B630 AEMET station, see Fig. 2c) is 652 mm y$^{-1}$. A rainfall amount of 140 mm in 24 hours was estimated to have a recurrence period of 25 years (YACU, 2003).

**2.2 Meteorological context of the 9$^{th}$ October 2018 rain event**

The 9$^{th}$ October storm affected the two northernmost catchments of the Llevant County; i.e., the Ca n'Amer and Canyamel Rivers (Fig. 2) with 9 and 4 casualties respectively and significant damages. The synoptic situation was like the situations generating flash-flood events in the Western Mediterranean (Fig. 3a). A cut-off low at mid-level was located in the eastern part of the Iberian Peninsula and a shallow low level pressure was affecting the same region driving warm and wet air from 175  the Mediterranean Sea to Balearic Islands and the eastern part of the Iberian Peninsula. This occurred in early October, when the sea surface temperature is close to its annual maximum in the Western Mediterranean, providing high quantities of moisture. Moreover, the cut-off low showed a typical divergence at mid-levels on its eastern flank, affecting the Balearic Islands and favouring the development of deep convection. Convection started on the sea between the Balearic Islands and the Iberian Peninsula (Figures 3b and 3c) and due to SW winds at mid tropospheric levels, the convective cells started to move 180  towards the Balearic Islands, where they triggered the flash-flood event after a heavy rainfall episode (Figures 3d and 3e).

**2.3 Rainfall data**

In order to assess the rainfall-runoff processes during the flash flood event in the Begura de Salma River, the continuous 10-min precipitation record of the event was obtained from radar (see location in Figure 2b) images that were initially calibrated 185  through rainfall data downloaded at https://opendata.aemet.es/ and http://asomet.balearsmeteo.com/. The two webpages contain meteorological data for official stations of the Spanish Meteorological Agency (AEMET) and the Meteorological Association 'Balearsmeteo', respectively (see Fig. 2c: B526X-Artà-Molí d'en Leu, B496X-Son Servera, B614E-Manacor, B569X-Far de Capdepera and B603X-Colònia de Sant Pere from AEMET; and BM01-Sant Llorenç des Cardassar from Balearsmeteo). After calibration, all radar images were geo-statistically treated to obtain the hourly mean and total precipitation 190  (and its standard deviation) fallen in regular squares 1x1 km in size (Fig. 2c), thus allowing analysis of the spatio-temporal distribution of the rainfall event.

The radar, located in the west of Mallorca, obtains reflectivity data (in dBZ) at 1*1 km of spatial resolution and at a 10-minute temporal resolution for many altitudes. Thanks to the AEMET Open Data Platform (opendata.aemet.es), reflectivity data corresponding to the lower Plan Position Indicator (PPI), which corresponds to an azimuth of 0.5º, can be downloaded in near-real time. Using reflectivity data, rainfall was calculated by using the Marshall-Palmer relation (Marshall and Palmer, 1948).

$$Z = aR^b \tag{1}$$

where $Z$ is the radar reflectivity (mm$^6$ m$^{-3}$), R is the rain rate (mm h$^{-1}$) and $a$ and $b$ are two coefficients. As the obtained reflectivity data is in dBZ units instead of mm$^6$ m$^{-3}$ a conversion was necessary using the following relation

$$dBZ = 10 * \log (Z) \tag{2}$$

The relation between $Z$ and $R$ depends mainly on the drop size distribution (DSD), which is usually unknown, unless disdrometer data are available, which is not the case (Chapon et al., 2008). Hence, coefficients $a$ and $b$ cannot be well established for this event. Multiple pair of values for $a$ and $b$ coefficients can be found in the literature, mostly changing between precipitation types.

We tested two pair of coefficients: (i) the pair $a$=200 and $b$=1.6 was tested because AEMET commonly uses these coefficients to obtain near-real time rainfall estimations (http://www.aemet.es/es/eltiempo/observacion/radar/ayuda; Last access: 15 May 2020) from the same radar data that we used in this research; (ii) the pair $a$=300 and $b$=1.4 was tested because the NWS in the USA uses it at operational level (Fulton et al., 1998) and it is argued that these coefficients performs better in a convective environment than the first ones (e.g. Seo et al., 2020).

The use of the two pairs of coefficients results in the estimation of high amounts of precipitation (100 mm in the first case and 150 mm in the second case), but an amount much lower than the maximum rainfall observation, which is higher than 250 mm, which shows the complexity of calculating the precipitation of this event accurately by radar data. In the scientific literature, many methodologies to enhance precipitation calculation using radar data can be found. While some methods focus on the correction of the reflectivity data; others are based on the calibration of the estimation of precipitation using reflectivity data as these are the two main sources of errors (Harrison et al., 2000). Mountains may partially or totally block the electromagnetic radar signal and affect radar reflectivity and precipitation estimations (Germann and Joss, 2004). The study area is mountainous, but with low maximum altitudes (~400 m.a.s.l.). This low elevation combined with the regional orography, the distance of the Begura de Salma River from the radar (~50 km), the 0.5º azimuth of the PPI used, and the altitude of the radar location (113 m.a.s.l.), avoided any topographic interference with the radar signal. Thus, no any orographic blocking reflectivity correction technique was needed. To directly correct the estimation of precipitation, geostatistical methods are commonly implemented in combination with the use of rain gauge data (Barbosa et al., 2018), but the data available data in

the most affected area are too few to implement these methods. Available data from the surrounding area could be used, but due to the sharp differences in rainfall between the affected and surrounding areas we decided not to use these data.

With the set of coefficients a=300 and b=1.4, the maximum amount of estimated rainfall using radar data clearly underestimated the observed rainfall, with a PBIAS of –50.6% and an estimation of ca. 149 mm as the maximum rainfall amount, compared with the 257 mm recorded at the Sant Llorenç des Cardassar rain gauging station (see Fig. 2c). Instead of

using the recorded rainfall in gauging stations to calibrate the radar-based rainfall, a correction method of the rainfall estimation based on spatial resampling was posited here. Accordingly, the 2*2 km spatial resolution of radar data was resampled by assigning to each grid cell the value of the maximum amount of estimated rainfall at 1*1 km. By this method, the regression coefficient reached $R^2= 0.8$, a PBIAS of only +2.6% and 258 mm as the maximum estimated rainfall amount, wich fitted to the rainfall at that point (Fig. S1).


## 2.4 Discharge data

A hydrometric gauging station is located where the Begura de Salma River enters Sant Llorenç des Cardassar village, 50 m upstream from the Ma-3323 road bridge (Fig. 4a). The station was built by the water authority in the 1970s. After years of abandonment, in 2015 the MEDhyCON Research Group installed within the gauge house a Hobo Water Level U20L-04, which

measures the water stage by 1-minute readings, accumulating 15-minute average values. The station is located at the very beginning of the concrete channelizing that takes the river through the village, closing a drainage basin of 23 km² and located ca. 50 m upstream of the first of the five bridges that cross the river within the village (see Bridge 1 in Fig. 4b). At the cross section where the hydrometric station is located, the channel bed is 70.25 masl, whilst the top of its bank channels are 72.00 masl.

The transformation of water stage (hereinafter WS; m) to $Q$ (m³ s⁻¹) through the stage/discharge rating curves (hereinafter SDRC) into the qualitative range (from low to high $Q$ conditions) are broadly developed by power and also polynomial equations, characterised by physical-based parameters. With the absence of direct flow measurements for the $Q$ estimation, a complete two-dimensional hydraulic model was developed with HEC-RAS 5.0.6 software (November 2018 version) by the US Army Corps of Engineers, Hydrologic Engineering Center. In this case, a set of differential equations in partial derivatives

that govern flow behaviour, known as Saint Venant's equations, were solved in shallow water equations.

For the generation of the 2D hydraulic model, a HR-DTM derived from a 1 m LiDAR-based DEM dating from 2014 was used; http://pnoa.ign.es/coberturalidar. In addition, the entire concrete channeling of the Begura de Salma River was topographically surveyed with a dGPS Leica 1200 for its integration to the HR-DTM. Within the model, all the bridges with potential flow effects on the hydrometric station were also introduced (Fig. 3a). This 2D hydraulic model was applied in a longitudinal river

section established 150 m upstream and downstream from the hydrometric station. Previously, the geometry of the gauging section was measured to characterize its hydraulic functioning. This is a regular trapezoidal concrete channeling of ca. 300 m in length with low sinuosity (Fig. 4a), a longitudinal slope of 0.0052 m m⁻¹, 10 m wide at the bed channel and with the height

of embankments 1.75 m. The slope of the longitudinal section was hydraulically high, which meant that the critical depth at the beginning of the channeling was the constraining factor of its hydraulic capacity (ca. 78 $m^3$ $s^{-1}$). This $Q$ value was related to 1.75 m of WS, coinciding with the height of the embankments.

As boundary conditions, an input hydrograph was arranged at the most upstream part of the section studied, whilst the WS was subsequently arranged in accordance with the slope of the channel bed at the most downstream part. Hydraulic simulations were always started with no flow conditions. Taking into account the hydraulic function characteristics for obtaining the SDRC, the calculation method used in these simulations was the Diffusion Wave. To design an accurate SDRC for the hydrometric station, the hydraulic effects of concrete channelization and bridges were assessed, running iteratively the model by flowing the representative $Q$ values under stable hydraulic functioning conditions.

Accordingly, a first hydrograph was designed step-by-step by using $Q$ values for a power equation with 2 as exponent in a range between 2 to 512 $m^3$ $s^{-1}$. The duration of each constant $Q$ (step) of this hydrograph was established (from preliminary simulations) at 4 hours and 15 minutes, significantly longer than the transit of the flood wave through the longitudinal section studied. In this way, the flow transit for each step in steady state conditions was guaranteed. These conditions and this designed hydrograph gave a first approach to the SDRC. To improve the accuracy of the SDRC, a second hydrograph was designed with nineteen $Q$ values, also containing the previous nine $Q$ values of the first hydrograph, new intermediate values, and some other significant $Q$ values such as the maximum channel capacity and the $Q$ value at which bridges influences the hydraulics. To optimize the time consumption of the modelling, this second hydrograph employed with a constant $Q$ (step) of shorter temporal duration; i.e., 1 and 2 hours depending on the $Q$ range.

The SDRC was divided into two sections. The first was related to a scenario where bridges do not impact on the hydraulic functioning at the gauging section, despite flow being bankfull or even overbanking the channeling. The second section was a scenario under the influence of bridges; i.e., the flow below them and also over their decks. It is then clear that the design, dimensions and proximity to the hydrometric station of these bridges affected directly flow behaviour at the hydrometric station due to obstruction (see pictures in Fig. 4). The obstruction of Bridge 1 occurred when the upstream section of the concrete channeling reached the bankfull, such that the dragging and floating elements began to collide against the back of the bridge, triggering the obstruction process. Consequently, the hydraulic model had to integrate those obstructions observed at the downstream bridges, at least the two nearest ones (Fig. 4a). These post-event pictures and maximum water stages observed in situ 10 h after the event were useful for calculating the obstruction percentages of these two bridges:85% at Bridge 1, 40% at Bridge 2 and with no obstruction for the other ones. The overbank flow coefficients were 2.2 for bridges 1, 4 and 5, and 2.1 for bridges 2 and 3. The SDRC obtained allowed calculation of this phenomenon which is activated with $Q$ values ranging 130-160 $m^3$ $s^{-1}$. Nevertheless, a calibration in the hydraulic model of overbank flow coefficients and the degree of obstruction of these structures was needed and attained by complete flow equations (Full Momentum) without changes in boundary conditions. For this purpose, the WS calculated in open channel conditions were compared with those WS observed in the field during and after the event in the hydrometric station and its surrounding floodplain area. Maximum WS observed in situ 10 hours after the event, within a period of time in which the high-water marks were still preserved, were mapped through

ground control points. Three of them were selected as representative locations around the hydrometric station (Fig. 4b), and the maximum WS reached at the hydrometric station (4.55 m) was also included. The modelling results in open-channel conditions for the ground control points involved a < 5% error in WS or even <1% in the case of the WS recorded at the

hydrometric station.

The SDRC was finally designed (see Fig. 5), showing clearly the two differentiated sections with a gap between them in WS of ca. 1.5 m because 1.4 m is the edge of Road Bridge 1 located just 50 m downstream from the hydrometric station. The SDRC is fitted to a power equation for both sections, obtaining high values of significance ($R^2 > 0.990$). Thus, in the case of the first section of the SDRC, the adjustment was $R^2$ 0.999, operating in open channel flows up to 160 $m^3$ $s^{-1}$ in bankfull or

even overbanking the channeling, but always without the bridges influence. The second section, with a $R^2$ 0.996, defined for $Q > 160$ $m^3$ $s^{-1}$, was already influenced by the presence of the bridges. That is in open channel flows but conditioned by the backwater generated in the flow under pressure inside the span of Bridge 1 as well as over its concrete decks.

## 2.5 Hydrological modelling

A semi-distributed hydrological model was used to reproduce the hydrological response of the catchment during the flash-flood event, the Routing System (RS) model (García-Hernández et al., 2007; Jordan, 2007). The SOCONT was the rainfall-runoff model used. This comprised an appraisal of hydrological processes such as snowmelt as well as surface and subsurface infiltration-induced flow and groundwater flow due to percolation (Jordan, 2007; Schaefli et al., 2005). The catchments were divided into elevation bands to incorporate the influence of temperature evolution with altitude and orographic effects within

mountainous catchments. In this model, the sub-catchments were divided into 100 meter elevation bands. The SOCONT model was applied to each elevation band, which was addedat the outlet of each sub-catchment. The input data of the SOCONT was the temperature obtained from meteorological stations and the precipitation derived from radar measurements (see previous section).

The SOCONT model is shown in Fig. S2. First, the temperature was interpolated for each elevation band based on inverse

distance weighing using the Shepard method (Shepard, 1968). Resampled 1 km resolution radar data (see subsection 2.3) were used in the model to obtain precipitation for each elevation band by including all 1 km resolution points falling within each elevation band. The soil-infiltration model was based on modified GR3 equations (Schaefli et al., 2005). Infiltration and evaporation were determined by the soil saturation; i.e., infiltration is higher for lower soil saturation whereas evapotranspiration is higher for high soil saturation. Surface runoff was computed with the SWMM model. Soil-infiltration

was modified to simulate karstic hydrological dynamics, as shown in Fig. S3. Precipitation infiltrated the soil, as in Schaefli et al. (2005), as a function of soil saturation (Fig. S3a). The resulting outflow from the reservoir ($Q_{GR}$) is also dependent on the soil saturation: outflow increases with higher soil saturation. In the modified equations, soil saturation can increase up to a certain level ($H_{GR,threshold}$), with this being a parameter to be adjusted in the model (Fig. S3b). When this threshold is reached, the soil reservoir releases all the available volume contained between the $H_{GR,threshold}$ and the minimum water level ($k_{karst}$ in Fig.

S3c). The released volume of water is then transferred to the SWMM model described in Schaefli et al. (2005). The relevant parameters for the modified version of the soil infiltration model were the maximum soil capacity ($H_{GR,max}$), the threshold for the karstic behaviour ($H_{GR,threshold}$) and the release coefficient ($k_{karst}$).

## 2.6 Damage assessment

Rapid Mapping is a mature Earth Observation (EO) service with many years of user oriented development since the International Charter `Space and Major Disaster' was established in 1999. EO satellite data-derived disaster mapping during emergencies is provided to civil protection and humanitarian user communities at national, continental and worldwide scales. Given the great helpfulness of the Copernicus EMS reports attained by rapid mapping techniques, the damage assessment of the event focused on the comparison between two information sources. The first one, a 'ground-based' report, was the damage

analysis carried out by the Directorate General of Emergencies of the Balearic Islands Government (Pol, 2019a). This 'ground-based' report provided a detailed description of the resources mobilized in the emergency phase and also a damage inventory. The second information source was the 'remote-based' damage assessment by the Copernicus EMS (https://emergency.copernicus.eu/mapping/ems-product-component/EMSR323_01SANTLLORENC_02GRADING_MAP/2), which also included the flooded areas established by the

Copernicus EMS within Sant Llorenç des Cardassar village.

The damage assessment comparison between the two sources was developed by means of a cartographic overlay with GIS tools. To provide greater accuracy, detailed territorial information and flow direction were also incorporated. First, the type of buildings and land use at the urban plot scale from the General Directorate for the Cadastre (http://www.sedecatastro.gob.es/) were included. Second, the flow direction in the urban network was calculated with Arc Hydro Tools (ESRI, 2019). This gave

a preliminary assessment of the role of hydraulic processes in physical damage., Due to the flow direction, this is mainly related to the velocity vector component perpendicular to the building element surface (Amirebrahimi et al., 2016). The results are given in a set of tables and maps that summarize the effects of the event and help to reflect on its causes and consequences.

## 2.7 Sediment Connectivity

As well as the hydrogeomorphological monitoring tasks, the 15th October 2018 MEDhyCON Research Group collaborated in the Emergency operation to search for a missing person during the flash flood who had not yet been found. Firstly, and taking into account the emergency situation, the index of (water and sediment) connectivity at the catchment scale was applied to find the areas with the greatest sediment deposition potential, which were where victims could have been buried by the flash flood. The sediment connectivity index (IC) proposed by Borselli et al. (2008) and modified by Cavalli et al. (2013) determined the

preferential flow-paths by exploring the water and sediment transference patterns in different landscape compartments of the entire Ca n'Amer River. Thus, the IC is a dynamic property of the catchment that indicates the probability of a particle at a

certain location reaching a defined target area, which in this study was established at the catchment outlet (Trevisani and Cavalli, 2016). This morphometric index was mainly derived from a HR-DTM, in this case, a 1 m LiDAR-based DEM dating from 2014; http://pnoa.ign.es/coberturalidar. The IC was calculated as follows:

$$IC = \log 10\left(\frac{D_{up}}{D_{dn}}\right) = \log 10\left(\frac{\bar{W}\,\bar{S}\,\sqrt{A}}{\sum_i \frac{di}{W_i S_i}}\right) \tag{3}$$

where $D_{up}$ and $D_{dn}$ are up- and down-slope components respectively, $\bar{S}$ average percentage slope, $A$ the size of the upslope contributing area, $\bar{W}$ an averaged weighting factor representing terrain roughness and a flow length $di$ of the $i^{th}$ cell along the steepest downslope direction. IC was calculated by using the freely available *SedInConnect* (Version 2.3) software developed by Crema and Cavalli (2018).


## 2.8 Geomorphic change detection

HR-DEMs facilitate the improvement of sediment connectivity as a powerful tool to determine preferential flow-paths and those areas with the greatest potential sediment deposition. The evaluation of the flash-flood landform signature by UAVs is the second part of creating a tool for a rapid response of post-catastrophe search and rescue tasks by applying
hydrogeomorphological precision techniques. The estimation of overbank sedimentation allowed the calibration of the predicted large sedimentation by IC mapping and its reliability in detecting sites where victims might be buried by flood sediment.

The latest technological advances in remote data acquisition (i.e., UAVs) and topographic modelling (i.e., Structure for Motion –SfM– and Multi-View Stereo –MVS–) led to a huge advance in Earth and environmental sciences. Following the
incorporation of MEDhyCON to the emergency operations, several UAV flights were carried out all along the Ca n'Amer River, from the headwaters (Begura de Salma River) to its outlet into the Mediterranean Sea at the village of S'Illot (Fig. 2c). This fieldwork involved the establishment and survey of more than 250 ground control points (GCPs), needed for an appropriate geo-referencing of the aerial photographs taken by the drone. Therefore, on 15[th] October 2018, just six days after the flash flood, evidence of erosion was recorded by aerial photographs taken with a small unmanned aerial vehicle (UAV *DJI*
*Phantom 4 Pro*, < 2 kg) and its conventional camera. The sensor dimensions are 12.83 x 8.60 mm, 5472 x 3648 px. The camera was calibrated by means of the *Agisoft Lens*, an automatic lens calibration routine included in *Agisoft Metashape* that uses the LCD screen as a calibration target and enables the full camera calibration matrix, including non-linear distortion coefficients (*Manual* Agisoft Lens, 2018), to be calculated. Resolution was set at 20 Mpix, shutter speed at 1/2,000 s and focal length was 8.60 mm. Most of the active zones of the main stream –including the floodplain corridor– with evidence of erosion and
deposition were surveyed, which also ensured the recording of high-water marks.

Imagery acquired during the aerial campaign enabled (1) the creation of mosaics of aerial georeferenced images and (2) the generation of high-resolution digital terrain models. These were produced by *Agisoft Metashape Pro® v1.5.3* using automated digital photogrammetry techniques. This software obtains high-quality results easily from algorithms known as 'Structure-

from-Motion' (SfM). Further details on the implemented algorithms can be found in Lowe (2004) and Westoby et al. ( 2012).

For the proper acquisition of the imagery, flight altitude was set at 70 meters, ensuring ground resolution close to 0.02 m pix⁻¹, and the camera was programmed to shoot every 15 m, flying at an average speed of 5 m s⁻¹.

Once all the drone images were geo-referenced and properly mosaicked, topographic modelling (i.e., Structure from Motion) generated the post-flash-flood very-high-resolution DEM (i.e. 5 cm pixel size). The comparison of that DEM to that of the catchment prior to the catastrophic event (LiDAR-based DEM dating from 2014) allowed the quantification and assessment

of the actual magnitude (competence) of the event in terms of the volume of sediments eroded and/or deposited and the alteration of the fluvial morphology. It is worth noting that no geomorphic changes were observed between 2014 and October 2018, by photointerpretation of aerial imagery (PNOA, 2015) and the continuous measurement of water stages since January 2015, with no overbanked flood events. Consequently, geomorphic changes were estimated in a floodplain downstream from Sant Llorenç des Cardassar to evaluate the amount of overbank sedimentation in the area of the rescue where IC suggested the

search. Measurements were taken with a procedure, similar to DoD, that compared the elevation of the ground class points extracted from the LiDAR topography collected in 2014 (http://pnoa.ign.es/coberturalidar) and the points extracted from the 0.05 m-resolution DEM obtained by the UAV flight at the same coordinates. Errors (RMSE) in xyz of the UAV DEM were calculated for 12 precise coordinate points (different from those GCPs used for image geo-reference and located on surfaces not modified by the flash flood) within the floodplain area of volumetric measurements, which were being < 0.175 m.

**3 Results**

**3.1 Catchment hydrological dynamics**

The hydrogeological and geomorphological characteristics of the Mallorca river catchments control its surface water/groundwater interactions and thus generated different streamflow regimes (cf. Estrany et al., 2009). The headwaters of all sub-catchments and the tributaries that drain the Llevant Ranges and Marinas are ephemeral due to the high degree of

fracturing, fissuring and karstification, which favour infiltration and percolation through perched karstic aquifers unconnected to the main stream channels.

The hydrological monitoring period assessed in this paper by using data from the hydrometric station was from 10th January 2015 to 30th September 2018 (Fig. 6). The month of October 2018 month was reserved to develop a singular and deeper study that could describe better the catastrophic flash-flood event (see results in sub-section 3.2). This gives a series of almost 4

hydrological years under hydrometeorological conditions illustrating ephemeral behaviour of the Begura de Salma River that was average in terms of precipitation (see the inset table of Fig. 6). In terms of $Q$, this inset table also shows the behaviour during the study period of several hydrological parameters. However, these values cannot be compared in the long term due to errors of up to two orders of magnitude in $Q$ values measured by the hydrometric network managed by the Balearic Islands Government (cf. Fortesa et al., 2019). Events of different magnitudes occurred during this study period, some of them

representative of recurrence $\approx$ 5 years in terms of rainfall. However, only two events recorded peak $Q$ (hereafter $Q_{peak}$) values

> 1 m³ s⁻¹, both occurring in January, when the hydrological pathways were completely active due to saturation processes. In January 2015, 120 mm of rainfall within 48 h at the AEMET-B630 Ses Pastores rainfall station (see location in Fig. 2), created a flow response with a $Q_{peak}$ of 2.8 m³ s⁻¹. Finally, 153 mm of rain accumulated in January 2017 in 72 h with a $Q_{peak}$ of 4.8 m³ s⁻¹, the maximum recorded at the hydrometric station during the study period before the catastrophic flash flood.


## 3.2 Hydrological response of the flash flood

The hydrological response of the flash flood was analysed through variables derived from the rainfall (Table 1a, 7 variables) and runoff (Table 1b, 9 variables) of the catchment: Event rainfall duration: duration from the beginning of rainfall until it stopped; Time of maximum rainfall: time of the highest rainfall intensity; Centroid storm: central time of the rainfall event;

Average radar rainfall: mean rainfall obtained by radar; $IP_{max}$ average radar: average of the highest rainfall intensities obtained from radar rainfall points; $IP_{max}$ radar: highest rainfall intensity obtained from radar data; IP average radar: average of rainfall intensities obtained from radar rainfall points; Runoff: discharge volume amount divided by the catchment area; Runoff ratio: ratio between runoff and rainfall, also known as runoff coefficient when is expressed as a percentage; Event duration: duration of the flood event; $Q_{max}$: peak discharge; Time $Q_{max}$: time the peak discharge lasted; T centroid storm − T $Q_{max}$: duration

between the time of the rainfall centroid and the time of the discharge peak; $Q_{average}$: discharge average during the flood event; Unit peak discharge: peak discharge divided by catchment area, allowing the comparison of peak discharge regardless of catchment size; Reduced Unit peak discharge: discharge peak divided by catchment area in square kilometres elevated by 0.6. The exponent was obtained from Gaume et al. (2009), who applied this last parameter to compare reduced unit peak discharge from different flash-flood events. The duration of the rainfall event was ca. 10 h and the average catchment rainfall amount

was 249 mm for both the Blanquera and the Begura de Salma catchments. Average and maximum rainfall intensities in 10 minutes were respectively 25 mm h⁻¹ and 45 mm h⁻¹ (Table 1a). However, spatial differences in rainfall depth within the catchments could be seen. Thus, the total rainfall amount ranged spatially within the catchment from 170 mm (see R1 Fig. 7) to 285 mm (See R5 and R6 Fig. 6), with the highest rainfall amount in 1 h occurring at R12 (77.2 mm). These highest rainfall values occurred at the headwater parts of the Begura de Salma River catchment (R12 Fig. 7) at 15:00 h, the beginning of the

event. At 17:00 h, the convective train was moving very slowly causing a new peak of rainfall amounts in 1 h located in the downstream part of this catchment with values between 60 and 70 mm recorded at R5, R6 and R9 (Fig. 7). During the last part of the event, at 19:00 h, rainfall amounts in 1 h of 60 mm h⁻¹ were recorded from R2 to R5.
Rain started to fall at 15:00 h (official time; UTC + 2 h). At 18:00 h, its amount was already 104.2 ± 20 mm h⁻¹, but the runoff response was insignificant with $Q$ 0.089 m³ s⁻¹. However, one hour later, at 19:00 h, with rainfall reaching an amount of 144.6

± 36.8 mm h⁻¹ $Q$ was already bankfull, i.e., 120 m³ s⁻¹. This was probably because the catchment's soil infiltration capacity was exceeded, which caused a rapid overland flow. Consequently, only 15 minutes later, $Q_{peak}$ was recorded (i.e., 442 m³ s⁻¹) which triggered catastrophic flood. In addition, $Q$ values continued to be high (i.e., > 135 m³ s⁻¹) until 20:45 h due to the convective train maintaining rainfall intensities > 24 mm h⁻¹. At 00:00 h, the rainfall event finished and $Q$ dropped sharply to

0.016 m³ s⁻¹. Table 1b summarizes the most important runoff parameters, shedding further light on the hydrology of this flash-flood event.

### 3.3 Reproducing the flashy hydrological response of the catchment

The hydrological model described above helped to understand the process better during the event. The results of the hydrological model simulation can be seen in Figure 7. The input data used in the model were the continuous radar dataset described above and the temperature measured at the surrounding meteorological stations; i.e., three stations within a radius of 12 km. The model was calibrated to reproduce the event and the final parameters were set to:

- $H_{GR,max}$ = 1.4 m
- $H_{GR,threshold}$ = 0.215 m
- $K_{karst}$ = 0.045 m
- Initial conditions of the GR reservoir: $H(t_0)$ 0.08 m, corresponding to a low soil saturation.

The relative volume error was 6% between the simulation and the measurement. The simulated peak ratio was 437.7 m³ s⁻¹; with an estimated runoff coefficient of 37.8%. The recession limb was not as sharp as the measured $Q$. It is worth noting that it was not possible to simulate $Q$ with the same magnitude as the $Q$ measured during the episode with a non-modified version of the GR3 model.

The same parameters were used for the entire headwater catchment. Accordingly, the $Q_{peak}$ from the other headwater tributary located on the west flank of Sant Llorenç des Cardassar village (Sa Blanquera) was estimated by the model to be 77.2 m³ s⁻¹, corresponding to an estimated runoff coefficient of 22.7%.

### 3.4 Socioeconomic and territorial flood damages linked to hydrogeomorphological processes

The flash flood caused great social and economic repercussions in Llevant County and the whole of Mallorca, as well as extensive national and international media coverage. The flash-flood event was a catastrophe with 13 deaths and economic damage with great importuct on the population and infrastructure. The number of casualties in one of the most important international tourist resorts, considered traditionally safe, shook national and international opinion. For further assessment of the media impact, see Table S1.

The damage assessment report carried out by the Emergency Services of the Balearic Islands Government showed an unprecedented mobilization of resources in the Region during the first week after the catastrophe in line with the high number of victims and amount of damage (Table 2; Pol, 2019). The initial costs of the emergency works exceeded 1.5 million € including the following actions: cleaning and restoring river channels, demolition of walls and structures affected, removal of potential polluting sources (Pol, 2019b). The declaration of disaster area is regulated by Spanish Law 2/2018, other

complementary laws (BOE, 2018, 2019) and Decree 33/2018 (GOIB, 2018). These laws established public support for alleviating the basic needs of families, deaths, housing assistance, aid for loss of vehicles and support for the economic sectors affected. The laws provided assistance for the repair of public infrastructures and environmental damage, specifying the amount of aid and the administrative procedures to receive it. These regulations referred to all the affected areas, including the municipalities of Sant Llorenç des Cardassar, but also Artà, Capdepera, Son Servera and Manacor. Recovery was financed

jointly by the different public administrations: the Spanish Government, the Balearic Islands Autonomous Government, the Insular Government, and the Sant Llorenç des Cardassar City Council. In April 2019, the expenditure of the Regional Government had reached 30.4 million euros in recovery and mitigation actions (GOIB, 2019). This expenditure included aid to the affected towns and villages of 11.27 million € (2.7 million € for the Sant Llorenç des Cadassar City Council), aid to companies of 3.3 million, for rehabilitation of homes (1.6 million), vehicle recovery (1.5 million), social aid (1.2 million) and

0.264 million euros for deaths.. The Sant Llorenç City Council, deployed various funds from the Spanish Government and Autonomous and Insular Governments for an investment plan in Sant Llorenç des Cardassar of 3.51 million € (Ajuntament de Sant Llorenç des Cardassar, 2018). In parallel, the Insurance Compensation Consortium (CCS, 2018), the Spanish public agency that handles payments to affected people in cases of damage caused by catastrophic events, processed claims for the flash flood as well as all the payments following damage assessment after the disaster in Sant Llorenç des Cardassar. A total

of 774 claims were processed, with 6,842,468 € paid out (see Table S2).

A territorial and hydrological analysis of the damage assessment is developed here. The location of the affected buildings and the WS reached in the streets and buildings provided by the Emergencies Department of the Balearic Islands Government and by the Copernicus EMS enabled three affected zones within the urban area of Sant Llorenç des Cardassar to be mapped (Figure 8a). Zone 1 w due to the overbank flow of the Begura de Salma River and corresponded mostly to the affected areas defined

by the Copernicus EMS. In this Zone 1, the highest WS in the streets was reached, exceeding 3.3 m. Zones 2 and 3 were those urban areas affected by the overbank flow of the Sa Muntanyeta Creek, located in the northernmost area of the village. The streets of Sant Llorenç des Cardassar rerouted the overbank flow from both the Begura de Salma River and the Sa Muntanyeta Creek (Fig. 8b), causing significant damages to vehicles and movable public property. In addition, as most of the buildings in Sant Llorenç des Cardassar use the ground floor as a home or business, the event caused major flooding by water and mud that

made their use impossible and required cleaning and restoration. According to the Balearic Islands Government, 392 damaged buildings and plots were inventoried in the urban area of Sant Llorenç des Cardassar, most of them in Zone 1 (Figures 8c and 8d). The flow direction illustrated how the N-> S direction, parallel to the Begura de Salma River, caused most damage in Zone 1, with 349 affected buildings and a WS average of ca. 1.03 m. Zones 2 and 3 had lower-intensity damage, with 37 and 6 affected buildings and a maximum WS of 1.80 m and 1.60 m, respectively. In these Zones 2 and 3, the flow direction had

no clear pattern because the Sa Muntanyeta Creek has a small catchment (2.2 km$^2$) and the urban street network and plots are not parallel to its natural flow direction.

The maps included in the Balearic Flood Risk Management Plan (GOIB, 2016) indicate the urban area of Sant Llorenç des Cardassar as a maximum risk area. Accordingly, the Plan developed an analysis of the potentially affected areas by recurrence

periods of 10, 100 and 500 years (Figure 8e). In addition, Table 3 analyses the damaged buildings to see if they are included

in these official flood risk areas in accord with the recurrence periods. None of the flood risk maps for different return periods

encompassed the areas affected as a result of the event. The 10-year recurrence map only included 25% of the affected areas;

the 100-year covered 48% of the area damaged; while the 500-year map only reached 60% (Table 3).

On comparing the affected zones where damaged buildings are also depicted in the Copernicus EMS, some differences between

the initial flash-flood definition carried out by the Copernicus EMS-EU and the distribution of damaged buildings were found

(Figure 7f) in Zones 2 and 3. It is worth noting that the post-event definition of the Copernicus EMS (Copernicus Emergency

Management Service, 2018) covered ca. 90% of the real damage.

## 3.5 Sediment Connectivity and geomorphic change detection as emergency tools

The search for the only person missing during the flash flood who had not yet been found 6 days after the storm caused

considerable social consternation in the Balearic Islands and beyond. Subsequently, hydrogeomorphological precision

techniques were crucial. A very intense topographical survey constructed very high-resolution (i.e. 5 cm pixel size) digital

elevation models and orthophotomosaics.

First, given the emergency situation, the index of (water and sediment) connectivity at the basin scale was used to identify

those areas with the greatest sediment deposition potential (Fig. 9a). The IC allows good understanding of the sediment transfer

processes within drainage catchments. The most connected areas of a basin are those in which their different compartments

are more powerfully linked. That favours the largest water surface flow generation, and thus erosion and, potentially, larger

soil losses. On the contrary, the zones with low connectivity are those whose topographical characteristics disconnect water

and sediment flows, acting as storage or deposition areas. The IC was applied to the whole Ca n'Amer River basin but was

only analysed from the point where the missing person was last seen (Fig. 9b, point 1). That was the exact point where the car

in which he was circulating was swept away by the flood wave. Therefore, the preferential water and sediment paths most

likely to be followed by the flood flows were identified, as well as the most important deposition areas downstream from the

last point person was seen. The most likely deposition zone was identified and immediately communicated to the Emergency

Authorities, upstream from the bridge of the road Ma-15 which crosses the Ca n'Amer River about 1 km below Sant Llorenç

des Cardassar. The search activities concentrated on that area, which is where the last victim was found (Fig. 9b, point 2) when

the Emergency Authorities had decided to move their search activities to the mouth of the Ca n'Amer River and beyond into

the Mediterranean.

In addition to the ability of HR-DEMs to the improve sediment connectivity as a powerful tool to determine the preferential

flow-paths and deposition areas, the present study evaluated the landforms signatures of the event by using UAVs as a tool for

a rapid response of post-catastrophe search and rescue tasks along the whole downstream section of the Ca n'Amer River from

the village of Sant Llorenç des Cardassar, in order to measure effectively the sediment deposits generated by the flash flood

and to locate and quantify the most important deposition areas downstream from where the person was last seen. As the last

missing person was found by using the connectivity index, in the end the sediment deposition quantification was not needed during the Emergency operation. However, this study checked its validity by assessing the floodplain area where the last person was found. Accordingly, for each of the 7103 LiDAR points on the right bank of the Ca n'Amer River the elevation was compared From the differences interpolated (TIN), an elevations raster for a total volume of 844.28 m³ was calculated, for an area of 12,254 m². The irregular distribution of the sediments in Fig. 9c responds to the rescue mobilisation. In the gaps visible in the sedimentation area of Fig. 9c, vehicles and search machinery were removed and not included in the volumes.

## 4 Discussion

The flash-flood event described in this study fits with the monthly distribution of flash floods in Spain carried out by Gaume et al. (2009): October is the month with the highest number of this type of flood events. In addition, the hydrological characteristics of the event were comparable with the flash-flood requirements established by Amponsah et al. (2018) for inclusion in the EuroMedeFF database, which are a $Q_{peak}$ unit higher than 0.5 m³ s⁻¹ km², a spatial extent lower than 3,000 km² and a storm duration shorter than 48 h. In this case, the $Q_{peak}$ unit was 19 m³ s⁻¹ km² and the storm duration was 10 h. In addition, the characterization of 60 extreme flash-flood events carried out by Marchi et al. (2010) offers a framework for comparing the event on the Begura de Salma River with other flash floods, in terms of rainfall amount, rainstorm duration, catchment area, lag time, runoff coefficient and $Q_{peak}$ unit. With a rainstorm duration of 10 h and a mean rainfall amount of 249 mm, the event is located within the flash-flood group of events with the highest rainfall intensities, which is a key factor for extreme events due to the question of controlling the magnitude of the runoff response. This group of events consists mainly of Mediterranean and Alpine-Mediterranean catchments. The relationship between catchment area and lag time is located within the lowest flash flood response time reported still now. The lag time of the event (2.1 h) was the lowest of extreme flash-flood events with streamgauge data reported in Marchi et al. (2010). In addition, the maximum rainfall accumulated in the whole catchment occurred at 19.00 h (45 mm; see Table 1a), just 15 minutes before the $Q_{peak}$. This short response time was caused by a combination of geographic characteristics of the catchment as well as the occurrence in time and space of maximum rainfall amounts and intensities (Fig. 7), as is explained in sub-section 3.2. In addition to rainfall characteristics, other factors that play a key role in flash floods are lithology and prior wetness conditions. On the one hand, low runoff coefficients have been reported in karst areas with carbonate lithology due to high infiltration rates (Li et al., 2019). On the other hand, Marchi et al. (2010) reported differences in the median runoff coefficient up to 23%, which were higher on flash floods occurring when prior conditions were wet. The flash-flood event of the Begura de Salma River occurred under dry antecedent conditions because the rainfall amount for the 9 preceding days was only 6.4 mm in a period when evapotranspiration was still high as temperatures were quite warm (i.e. 20ºC). Despite these dry antecedent conditions, the runoff coefficient of the event (i.e., 35%) was analogous to the median runoff coefficient under average wetness conditions (37%) reported by Marchi et al. (2010), rather than dry ones (20%). This runoff response resulted from the combination of rainfall intensity and its spatial distribution,

complex geology and land cover disturbances in generating a high $Q_{peak}$ (i.e., 442 m$^3$ s$^{-1}$) with high potential for generating

geomorphological changes. Thus, the $Q_{peak}$ unit obtained (i.e., 19 m$^3$ s$^{-1}$ km$^2$) can be classified as the third highest value of all the reported values in Marchi et al. (2010) and the highest of those values obtained from streamflow measurements in a hydrometric station and not by post-event analysis. The hydrologic response analysis in the course of a flash flood shows how storm structure and evolution result in a scale-dependent flood response (Borga et al., 2007). Consequently, spatial rainfall organisation, geology combined with orography and land cover disturbances led to pronounced contrasts in the flood response

at the Begura de Salma River. Spatial rainfall on the catchment scale showed that the highest accumulation at the beginning of the storm was located at the headwaters of the catchment (at 15:00 h), whilst during the last part of the event the most important rainfall amounts were located in the downstream part. Examination of the flood response illustrated how the extent and the position of the karst terrain (Zanon et al., 2010) and soil conservation practices (Calsamiglia et al., 2018; Tarolli et al., 2014) provided major geological and anthropogenic control of runoff response. Impervious materials cover 74% of the Begura

de Salma River catchment, mostly located at the headwaters, which are responsible for the highest values of topographic torrentiality (Estrany and Grimalt, 2014), facilitating rapid overland flow generation. During the first part of the storm, when the highest rainfall amounts affected the headwaters, runoff response was delayed by the laminar effect of check-dam terraces and field terraces massively constructed over Cretaceous marls and Lias limestones respectively (Calvo-Cases et al., 2020) and by the predominance of percolation in those areas covered by limestone, mostly in the intermediate parts of the catchment.

During the last part of the event, when the highest rainfall intensities were in the downstream part, the excess of soil infiltration capacity and the collapse of headwater check-dam structures triggered the sudden increase in discharge from 120 to 442 m$^3$ s$^{-1}$ in only 15 minutes at the hydrometric station. Moreover, the increase of 5 km$^2$ (21% of the catchment area, see more details in section 2.1) of natural vegetation since the 1960s as a result of afforestation processes, increased fuel loads and the risk of wildfires led to 1.7 km$^2$ (7% of the catchment) being burned since 1980. The removal of vegetation by fires has a similar effect

(less interception, less soil storage), which has been experimentally documented after major fires. These factors are a major reason why the history of the steady devastation of plant cover in the Mediterranean is likely to enhance flood risk (Wainwright and Thornes, 2004) and increase desertification tendencies.

The hydrological model was calibrated specifically for the flooding event. The parameters of the modified GR reservoir as well as the initial conditions were adjusted to best represent the flooding event. A very sensitive parameter is the $H_{GR,threshold}$,

which regulated the time when additional water reserve in the soil was released. Modelling results were also very sensitive to initial conditions (soil saturation) before the rainfall event. During the calibration process, it was necessary to simulate, on the one hand, smaller flood events observed at the hydrometric station. On the other hand, the model had to reproduce the historical 2018 flood event. However, the flood event could not be reproduced when the model was calculated for a long time period, due to initial conditions that were not adjusted prior to the event. In this context, the initial condition $H(t_0)$ was manually

adjusted, as numerical models applied to simulate catchment runoff response are often unsuccessfully implemented for Mediterranean-climate catchments due to show very heterogeneous responses over time and space (Merheb et al., 2016). The uncertainty for the results regarding the Sa Blanquera River was higher because of the lack of hydrometric data in this

catchment. There was no karstic behaviour of the model within this subcatchment, which was the main modelling uncertainty for this ungauged subcatchment. The model analysis clearly showed that, without any massive water storage during the first part of the rainfall event, which was released at the $Q_{peak}$ of the event, it was impossible to reproduce the correct flood magnitude and the very short lag time. This water storage may be due to underground karstic volumes combined with pipes, or storage / dam break effects. Only future large flood events will enable validation of the chosen parameters, as the 2018 flood event was the only one needing a karstic component in the rainfall model to be correctly represented by the model.

The predictability of flash-flood events is unresolved, especially because forecasting of intense thunderstorms has also not been solved by operational meteorology. Even using one of the best state-of-the-art weather forecasting models, Harmonie/AROME, the Spanish National Weather Service (AEMET) only activated a yellow warning for one-hour accumulated precipitation of 20 mm beforehand. In contrast, the synoptic situation was forecasted well by global forecasting models some days before the event. An experienced forecaster could anticipate the occurrence of an intense thunderstorm by using these models, but would lack any quantitative or geographical precision, which are two key factors in flash-flood forecasting. However, now-casting products, based on radar, satellite and ground truth may anticipate severe weather situations better. These products are updated every often (several minutes to one hour) and compensate the weather forecasting models which are updated less often. The main challenge in using the hydrological model as a flash-flood early warning system is to include correctly initial soil saturation conditions as well as accurate rainfall forecasts. For the latter, the scientific community is working on now-casting products that typically deliver short-term (few hours lead time) rainfall forecasts that are updated very often, from a base of 10 minutes to an hour. These forecasts are based on real-time measurements that combine data from radar, satellites and meteorological stations. However, it is hard to  calculate initial conditions automatically, as the river is dry most of the time and there are no soil moisture measurements in the catchment. Data assimilation and automatic adjustment of initial conditions, which are usually applied in operational forecasts, are therefore not relevant here. However, an early warning system can be built using the model proposed in this paper by assessing the uncertainty of the forecast. At present, Mallorca does not have any sort of early warning system to assist flood risk management, and nor of course has Sant Llorenç des Cardassar. Similarly, no hydrometeorological early warning was issued by the competent authorities, as the Balearic Islands have no operational hydrological control network releasing real-time information on discharges. In October 2018, Sant Llorenç des Cardassar was one of the four municipalities in Mallorca with a flood risk emergency plan. However, it was not operational at the time the emergency was declared. As a result, the population was completely unaware of how to defend themselves, even during the emergency phase, although Sant Llorenç des Cardassar municipality had significant social vulnerability to floods, as most of the casualties were tourists and the elderly.

The addition of the MEDhyCON research group on 15[th] October 2018 to the Emergency operational allowed the application and testing of hydrogeomorphological precision techniques. The fundamentals are that flood risk plans and Emergency activities are based on a thorough understanding of linkages between sediment and catchment compartments at all stages of flood events. Integrating topography-based connectivity assessment (Kalantari et al., 2017) and geomorphic change detection may be a crucial support to decision-making in flood risk planning and in Emergency surveys, as this study shows. The

combination of hydrological and sediment connectivity (IC in various forms) with other key natural characteristics (i.e., soil type and topography by using LiDAR-based HR-DEM), along with the integration of territorial information such as land cover/uses by using Cadastre data bases (Piaggesi et al., 2011), results in a powerful tool. Accordingly, the easy-to-calculate

IC can be an effective tool for rescue tasks after extreme flash-flood events with a huge erosion capacity.

In addition, the post-event delimitation and damage assessment released by the Copernicus EMS (Copernicus, 2018) identified ca. 90% of the real damage in this traditional Mediterranean village, Sant Llorenç des Cardassar, consisting of compact blocks of buildings and plots. The Synthetic Aperture Radar (SAR) technology with very high spatial resolution (1-3 m; Plank, 2014) is fundamental to obtaining high efficiency and accuracy of this rapid mapping tool at low cost. Consequently, Emergency

resources can be directly concentrated on the most damaged areas without having to check the entire affected area on the ground.

The increase in the torrentiality of rainfall as a result of climate change in the Mediterranean region may exacerbate the level of exposure of urban areas and infrastructures to floods. Catastrophic events will increase in quantity and intensity. Local government bodies will need to adapt continuously prevention and management of flood risk tools to these new scenarios. The

legal framework for flood risk planning and management (GOIB, 2016) showed that the level of risk exposure was extensively known. In addition, the analysis of current regulations shows that the appropriate preventive measures were being taken to minimize possible damage in a potential event in the Balearic Islands. However, the magnitude of this flash-flood exceeded any type of forecast carried out by the risk and emergency plans. The consequences of the catastrophe reveled deficiencies in prevention by Local Government, both at the level of urban planning and infrastructure and in risk management itself. In

addition, the population was also unprepared due to a very low level of risk culture.

## 5 Conclusions

The hydrogeomorphological analysis and damage assessment developed in this paper has provided a comprehensive understanding of the Sant Llorenç des Cardassar flash-flood event of 9[th] October 2018 by means of an integrated approach with a meteorological, hydrological, geomorphological, damage and risk data analysis. The use of rainfall radar data –corrected

with measurements from rainfall stations in the surrounding region– combined with $Q$ data from stream gauge observations showed how spatio-temporal distribution of rainfall amounts and intensities, karstic features and land use/cover resulted in an unprecedented runoff response in a Mediterranean environment, triggering this natural disaster. It was shown that the application of different direct estimation approaches may reduce the uncertainty of hydrological modelling and thus increase the credibility and practical value of the whole analysis. Undoubtedly, the inclusion of streamflow monitoring data for this kind

of flash-flood event proved crucial, as did accurate calibration with a two-dimensional hydraulic model also integrating the influence of bridges' obstruction in flow routing.

The flash-flood event was a catastrophe that caused 13 casualties, huge economic damage and an unprecedented mobilization of human resources in the Balearic Islands. Rapid mapping from Copernicus EMS and detailed damage reported by regional

authorities, linked to territorial information from the Cadastre and hydrogeomorphological processes, showed very accurately the damage-driving factors in the urban area of Sant Llorenç des Cardassar village. Although flood risk planning showed the high level of risk exposure, the disaster was generated by a very high exposure of buildings and infrastructures to floods, the absence of early warning systems with efficient action protocols, and the lack of municipal regulations to instruct the population on how to act when struck by an event of this magnitude. The incorporation of hydrogeomorphological precision tools during Emergency post-catastrophe operations was revealed as a powerful tool. Then, the simple application of a geomorphometric index from easy-access LiDAR-based topographic data resulted in a rapid identification of deposition zones in the different compartments of a catchment, which helped in the search and rescue of missing persons. In addition, the evaluation of landform signatures by using UAVs measured effectively the sediment deposits generated by the flash flood and/or mobilised by the Emergency operations during rescue searches.

This study represents a first step to further improvement of flash-flood risk management in Mediterranean flood-prone regions such as Mallorca, which are likely to recur due to global change. Mediterranean regions are subject to violent flash floods that may intensify –especially in terms of peak discharge– in the future due to forest fire, land uses and/or climate changes. These future consequences of global change should lead to the modification and adaptation of hydrological and flood risk models, allowing the development of a rule-based system with adaptive and resilient measures to take at the catchment scale.

*Author contributions.* JE, MR, RM, AC and FV developed the experimental design; whilst JE, JF and JG were responsible for data curation, fieldwork and figure elaboration and MT carried out the meteorological analysis. BN and FV performed the hydraulic modelling. RM and XP developed the hydrological model code and performed the simulations. MR completed the damage assessment. AC performed the sediment connectivity and geomorphic change detection. Resources and funding acquisition were supervised by JE and MR. JE prepared the manuscript with contributions from all co-authors.

**Acknowledgements**

This research was supported by the Spanish Ministry of Science, Innovation and Universities, the Spanish Agency of Research (AEI) and the European Regional Development Funds (ERDF) through the project CGL2017-88200-R "Functional hydrological and sediment connectivity at Mediterranean catchments: global change scenarios –MEDhyCON2". Josep Fortesa has a contract funded by the Vice-presidency and Ministry of Innovation, Research and Tourism of the Autonomous Government of the Balearic Islands (FPI/2048/2017). The contribution of Miquel Tomàs-Burguera was supported by the project CGL2017-83866-C3-3-R also funded by the AEI. Julián García-Comendador is in receipt of a pre-doctoral contract (FPU15/05239) funded by the Spanish Ministry of Education and Culture. Compensation payments were facilitated by the Spanish Insurance Compensation Consortium, whilst the type of buildings and land use at the urban plot scale were provided by the Spanish Directorate General for the Cadastre and the damage report was by the Directorate General of Emergencies of the Balearic Islands Government. Meteorological data were facilitated by the Spanish Meteorological Agency (AEMET). We are grateful to BalearsMeteo for providing subhourly rainfall data of the event. The authors want to thank Xurxo Gago, Carlos

J. Oliveros, José A. López-Tarazón and Hassan Ouakhir for their assistance during fieldwork. Finally, we want to pay tribute to all the professionals and volunteers who worked with such determination in the rescue tasks.

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

 **Tables and Captions**

(a)

| Event rainfall duration (h) | Time of maximum rainfall | Centroid storm | Average radar rainfall (mm) | IP$_{max}$ average radar (mm h$^{-1}$) | IP$_{max}$ radar (mm h$^{-1}$) | IP average radar (mm h$^{-1}$) |
|---|---|---|---|---|---|---|
| **10** | 09/10/18 19:00 | 09/10/2018 17:07 | 249 | 45 | 77 | 25 |

(b)

| Runoff (mm) | Runoff ratio | Event duration (h) | Q$_{max}$ (m$^3$ s$^{-1}$) | Time Q$_{max}$ | T centroid storm-T Q$_{max}$ (h) | Q$_{average}$ (m$^3$ s$^{-1}$) | Unit peak discharge (m$^3$ s$^{-1}$ km$^2$) | Reduced Unit peak discharge (m$^3$ s$^{-1}$ km$^2$) |
|---|---|---|---|---|---|---|---|---|
| **86** | 0.35 | 12 | 442 | 09/10/18 19:15 | 2.1 | 26 | 19 | 67 |

**Table 1** (a) Rainfall and (b) runoff variables of the flash flood at the Begura de Salma River catchment estimated from the continuous water stage monitoring at the MEDhyCON hydrometric station located in the village of Sant Llorenç des Cardassar.

| People and goods | | Buildings | |
|---|---|---|---|
| Death toll | 13 | Emergency interventions into the structure | 52 |
| Slightly injured | 4 | Demolitions | 3 |
| Initial missing persons | 74 | Affected buildings | > 1000 |
| **Infrastructure damages** | | **Movable properties** | |
| Cut roads | 4 main roads | Motor vehicles | 426 |
| Affected roads | 22 road sections | | |
| Bridges | 8 with structural damages | **Actions undertaken** | |
| Public hydraulic domain | High impact | Rubble removal | 7,000 tonnes |
| Drinking and wasting water network | Several points of damage | Human Emergency resources mobilized | > 200 |
| Telecom infrastructures | Severe damage | | |
| Electricity network | Undetermined severe damage 8355 affected users | Rescue assistance | 342 persons |

**Table 2** Damage summary and emergency actions after the 9th October 2018 violent flash-flood in Llevant County, Mallorca. Source: Pol (2019).

| Damage level | Flood risk cartography | | | | | | Copernicus Emergency Management System | | Total |
|---|---|---|---|---|---|---|---|---|---|
| | *10 years* | *%* | *100 years* | *%* | *500 years* | *%* | *Affected area* | *%* | |
| COLLAPSED | 5 | 50 | 8 | 80 | 9 | 90 | 10 | 100 | 10 |
| DAMAGED & HABITABLE | 52 | 20 | 107 | 41 | 141 | 54 | 225 | 86 | 261 |
| DAMAGED & NOT HABITABLE | 15 | 41 | 26 | 70 | 31 | 84 | 36 | 97 | 37 |
| DAMAGED PLOT | 6 | 35 | 9 | 53 | 9 | 53 | 13 | 76 | 17 |
| DAMAGED & RESTRICTED USE | 19 | 28 | 39 | 58 | 47 | 70 | 63 | 94 | 67 |
| TOTAL | 97 | 25 | 189 | 48 | 237 | 60 | 347 | 89 | 392 |

**Table 3** Damaged buildings in the village of Sant Llorenç des Cardassar caused by the violent flash-flood on 9[th] October 2018 and those encompassed in the official flood risk maps for 10, 100 and 500 years recurrence periods.

945    **Figures and Captions**

950

955

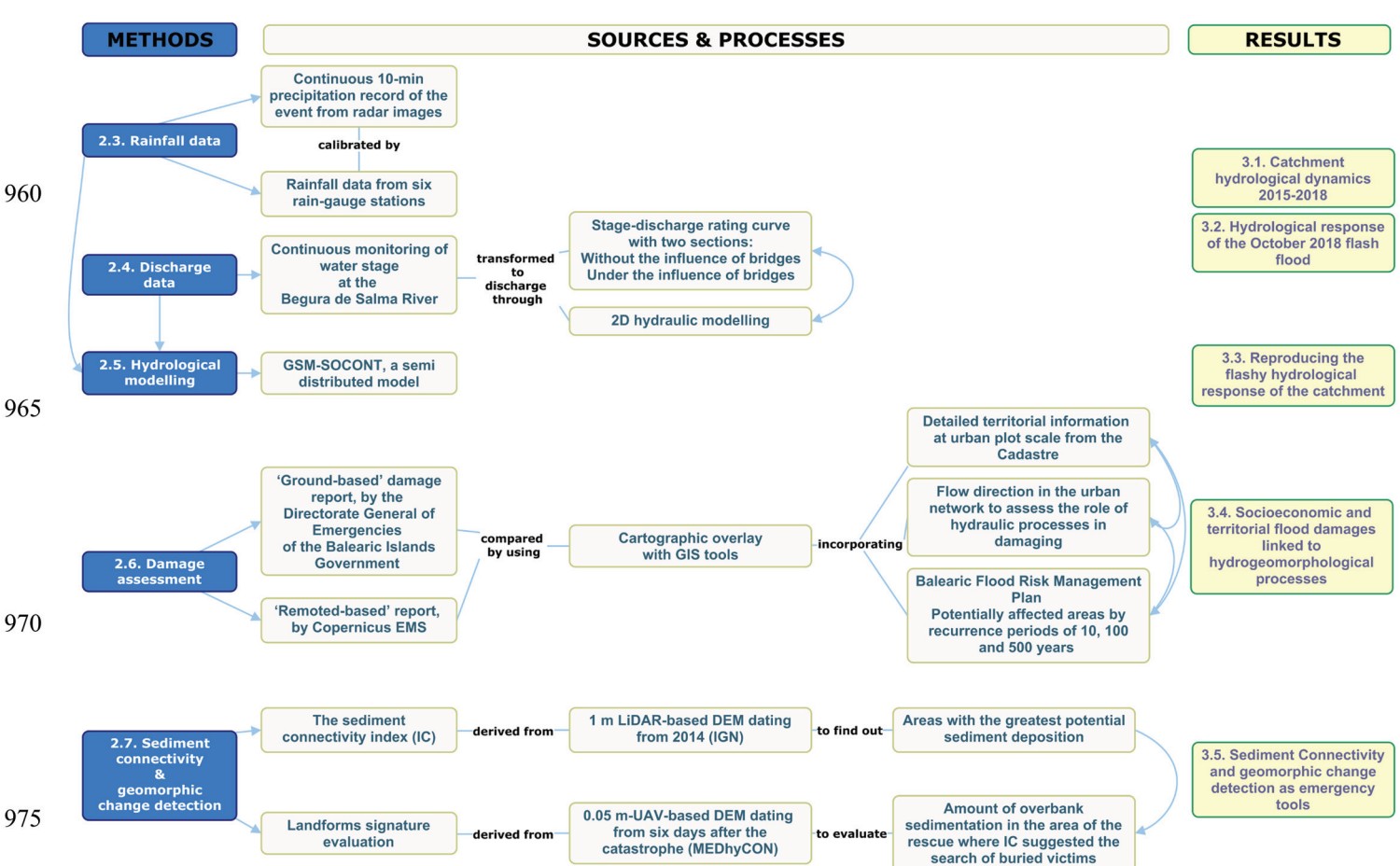

**Figure 1** Workflow of the experimental design.

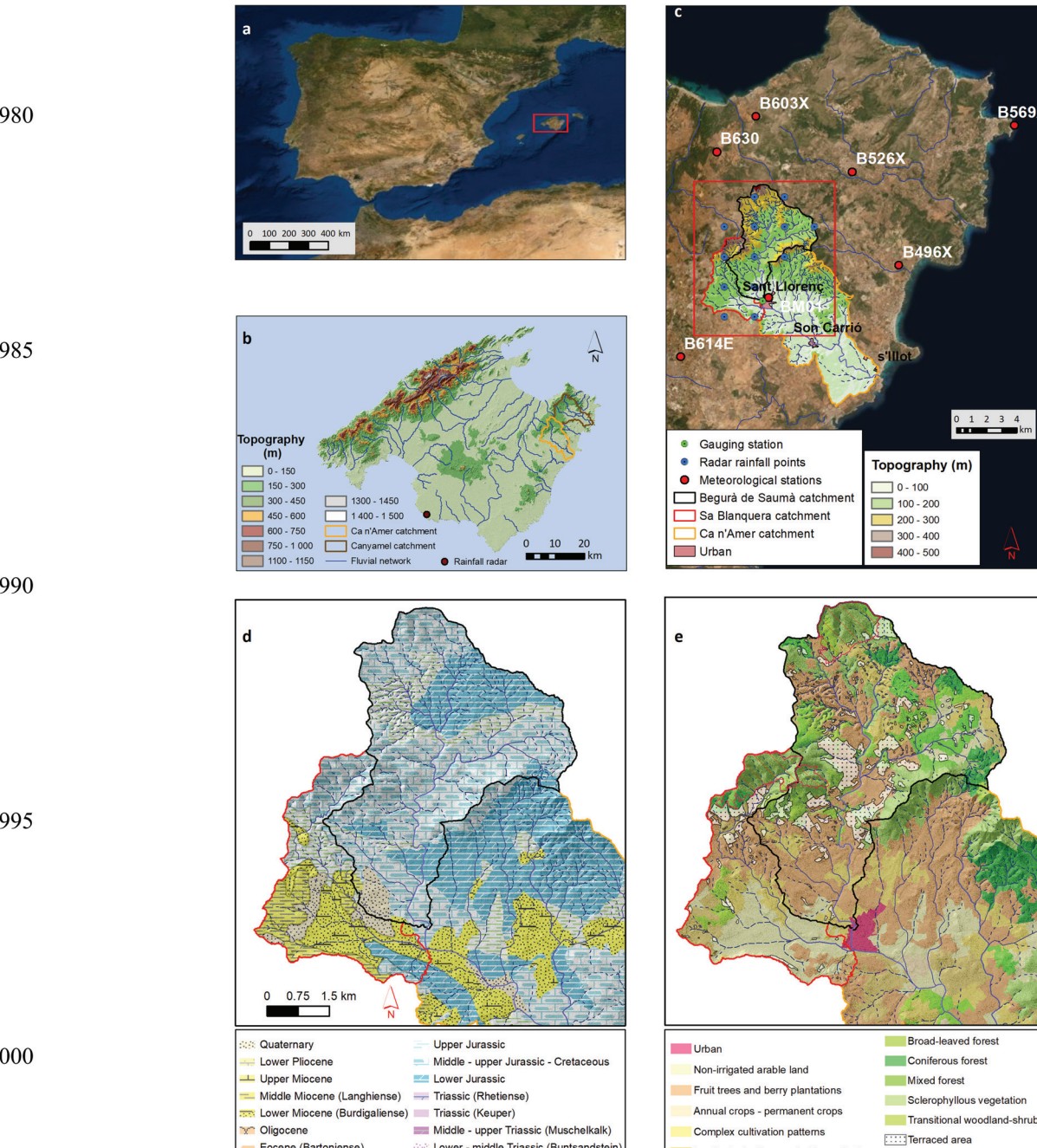

**Figure 2** Main characteristics of affected basins during the 9[th] October 2018 flash-flood. (a) Location of Mallorca in the western Mediterranean. (b) Topography and fluvial network of Mallorca with the location of the main basins affected: Canyamel and Ca n'Amer rivers. (c) Blanquera and Begura de Salma headwater river catchments within the Ca n'Amer River, with locations of rainfall and hydrometric stations; and radar rainfall points derived from a regular mesh of 1x1 km. Source: https://opendata.aemet.es. Background: aerial photography and DEM data (PNOA, 2015). (d) Lithology of both Blanquera and Begura de Salma catchments. (e) Land uses and terraced areas for the same headwater catchments. Source: Corine Land Cover (2018).

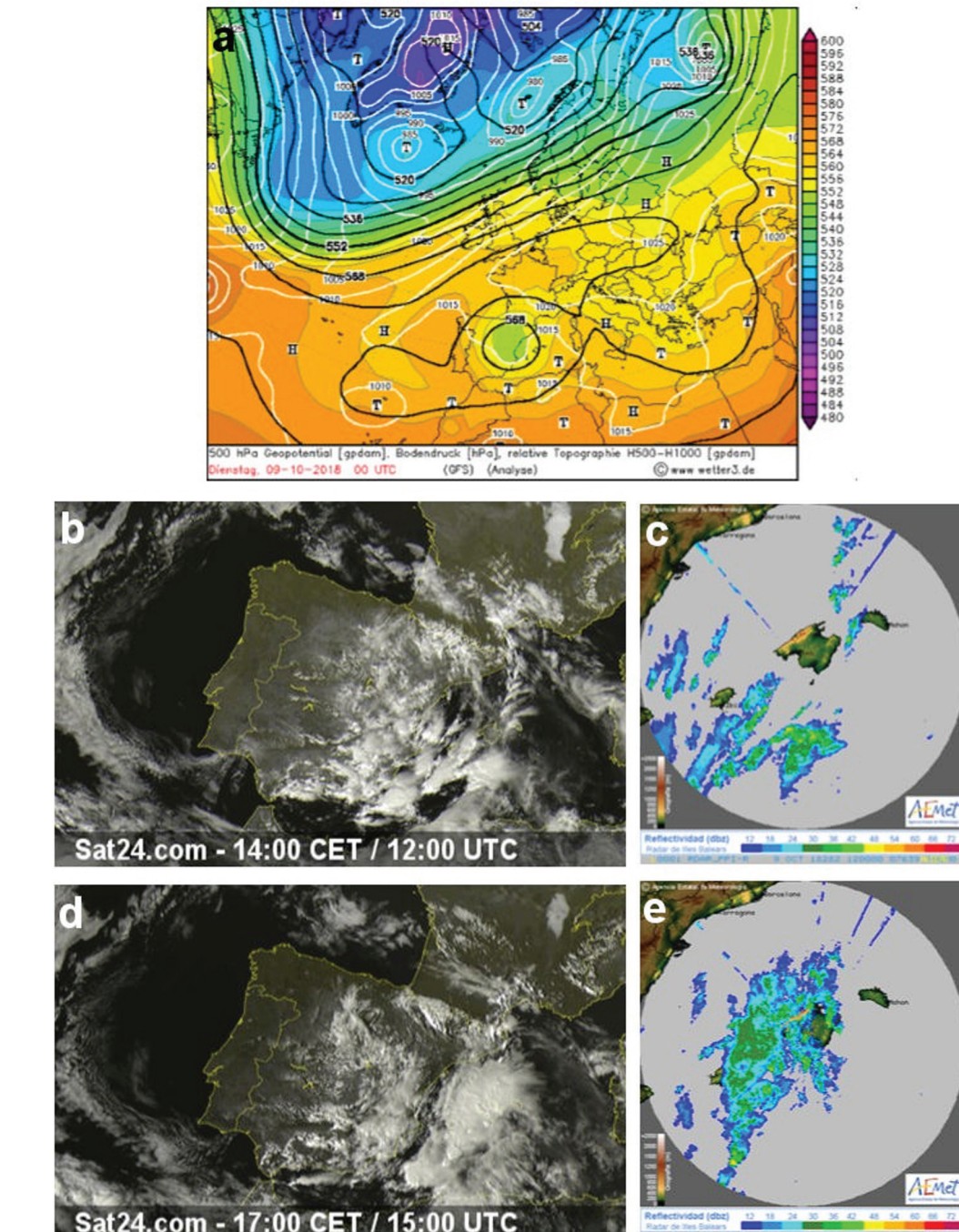

**Figure 3** (a) Surface pressure and 500-hPa height analyses at 1200 UTC on 9[th] October 2018 Source: **http://wetter3.de**; i.e. , at the beginning of the precipitation event. Satellite image at (b) 12.00UTC and (d) 15.00UTC Source: http://www.sat24.com. EUMETSAT and radar images at the same hours (c and e) Source: http://www.aemet.es.

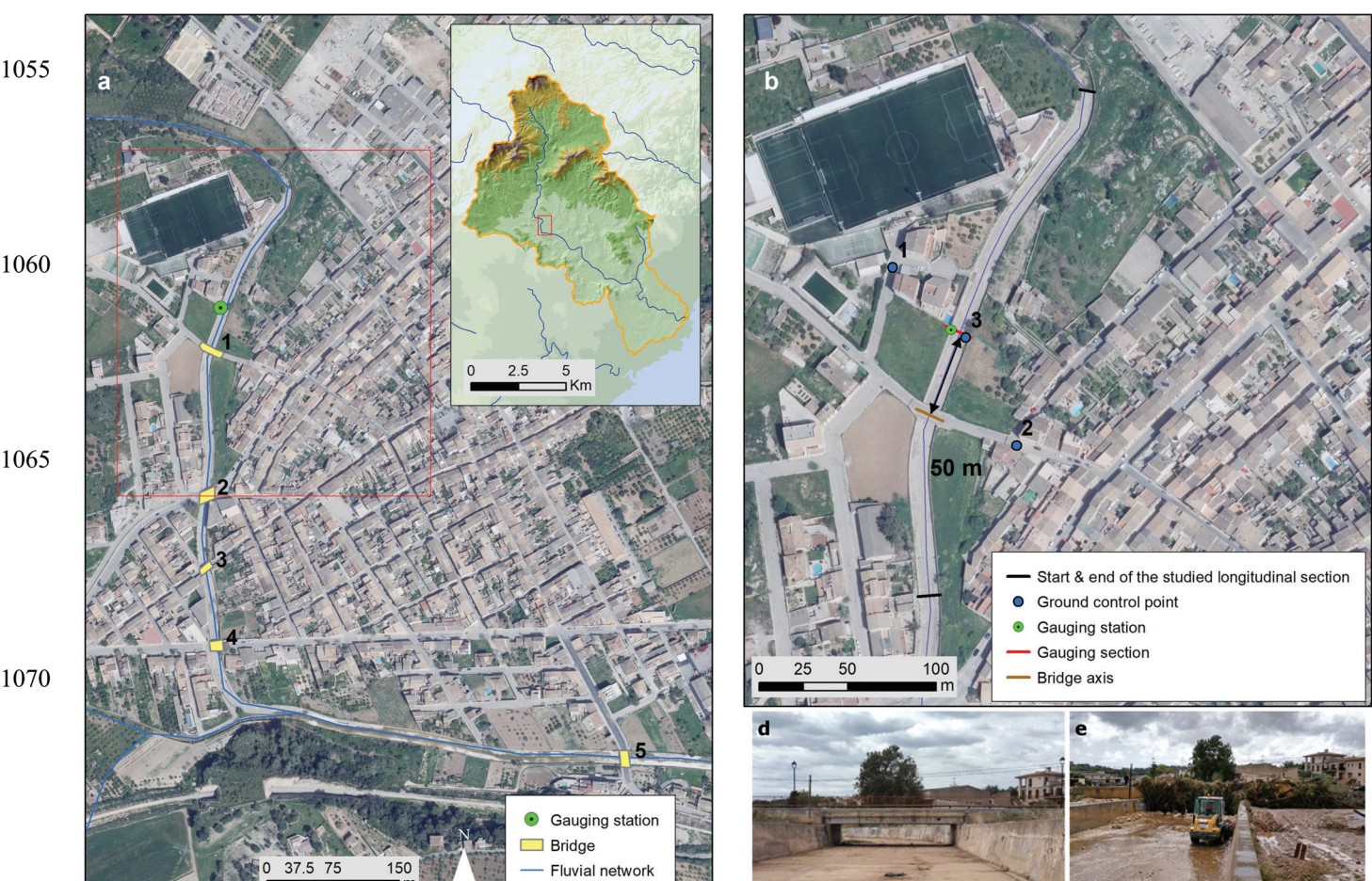

**Figure 4** (a) Aerial view of the concrete channeling of the Begura de Salma River that crossing Sant Llorenç des Cardassar village and the location of bridges. (b) Detailed aerial view of the very beginning of this concrete channeling, where the hydrometric station is located. The photographs show a view of Bridge 1 from the hydrometric station when (d) the digital equipment was installed, on 10[th] June 2015 and (e) a few hours after the flash flood, on 10[th] October 2018. Background: aerial photography and DEM data (PNOA, 2015).




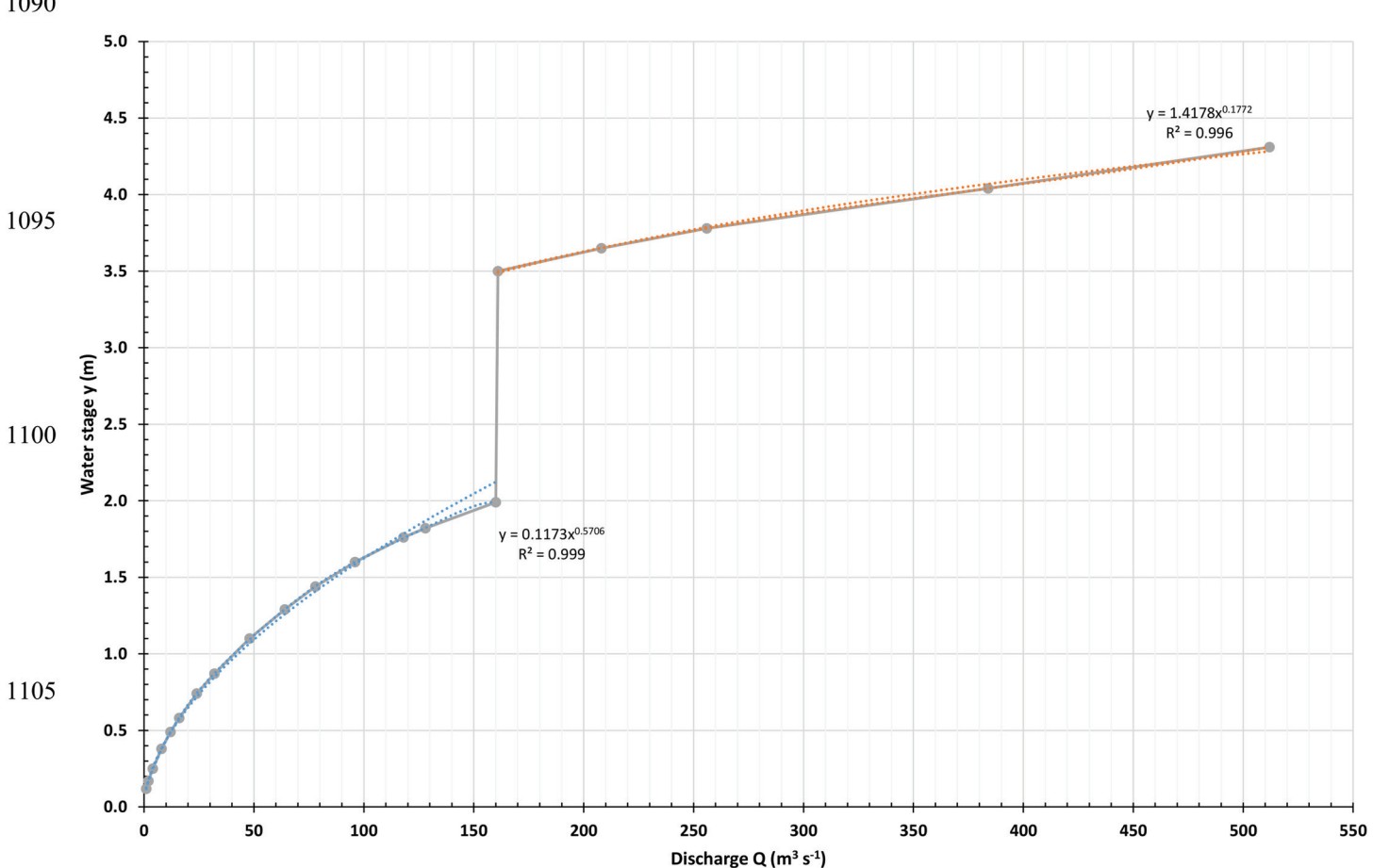




**Figure 5** Stage-discharge rating curve performed by means of two-dimensional hydraulic modelling with two differentiated sections in
accord with the influence of Bridge 1 (see Figure 3a) and its potential obstruction.

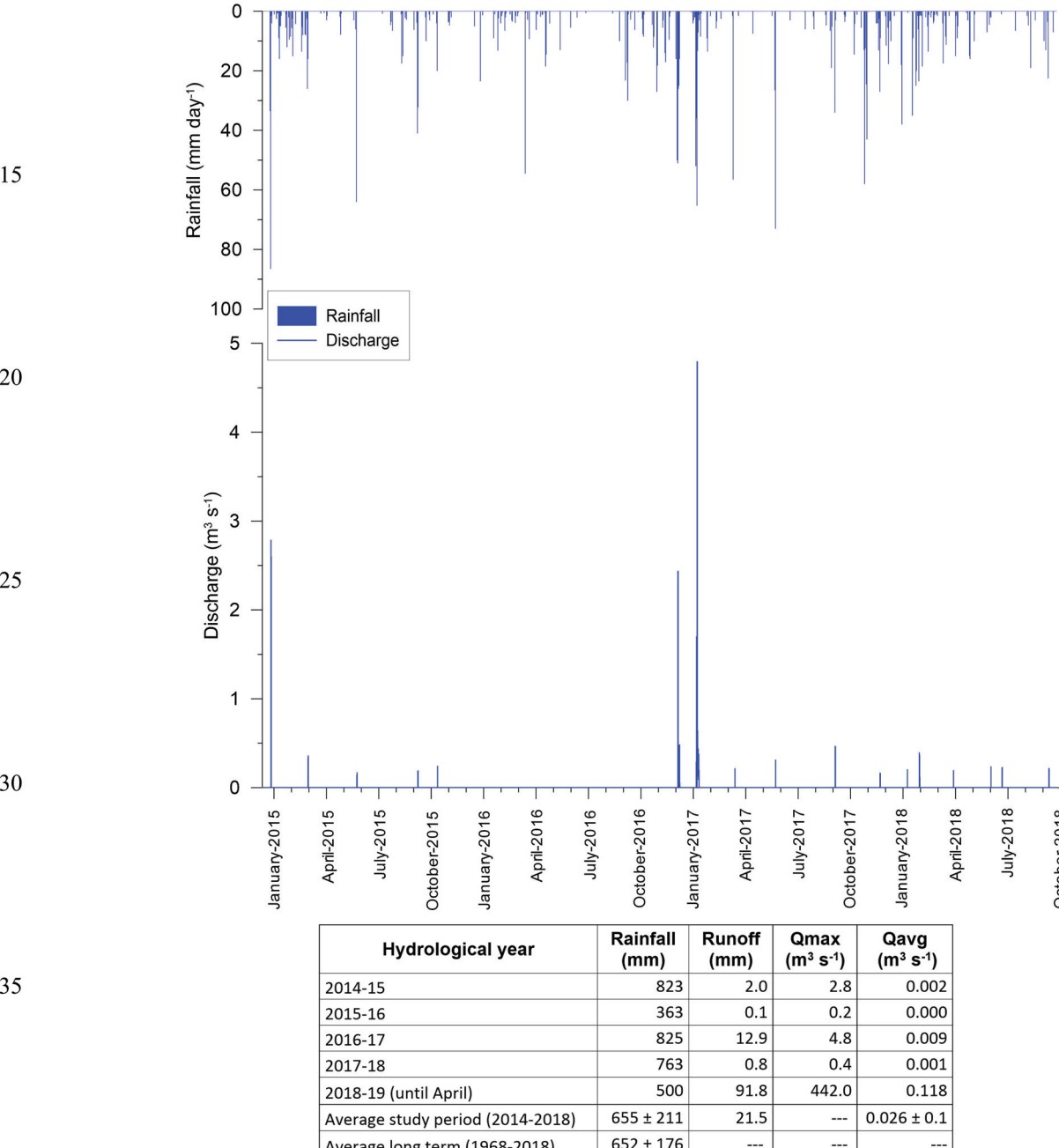

| Hydrological year | Rainfall (mm) | Runoff (mm) | Qmax (m³ s⁻¹) | Qavg (m³ s⁻¹) |
|---|---|---|---|---|
| 2014-15 | 823 | 2.0 | 2.8 | 0.002 |
| 2015-16 | 363 | 0.1 | 0.2 | 0.000 |
| 2016-17 | 825 | 12.9 | 4.8 | 0.009 |
| 2017-18 | 763 | 0.8 | 0.4 | 0.001 |
| 2018-19 (until April) | 500 | 91.8 | 442.0 | 0.118 |
| Average study period (2014-2018) | 655 ± 211 | 21.5 | --- | 0.026 ± 0.1 |
| Average long term (1968-2018) | 652 ± 176 | --- | --- | --- |

**Figure 6** Discharge at 15-min intervals measured at the MEDhyCON hydrometric station located at the beginning of the concrete channeling of the Begura de Salma River in Sant Llorenç des Cardassar. Likewise, the daily rainfall measured at the AEMET- B630 Ses Pastores during the monitored period (10th January 2015-30th September 2018), prior to the catastrophic flash flood of 9th October 2018. Bottom set table: Rainfall, runoff and peak discharge for hydrological years during study period. Rainfall data are from AEMET-B630 Ses Pastores, located 10.5 km from the Begura de Salma catchment outlet and representative of the rainfall dynamics of the Llevant Ranges headwaters.

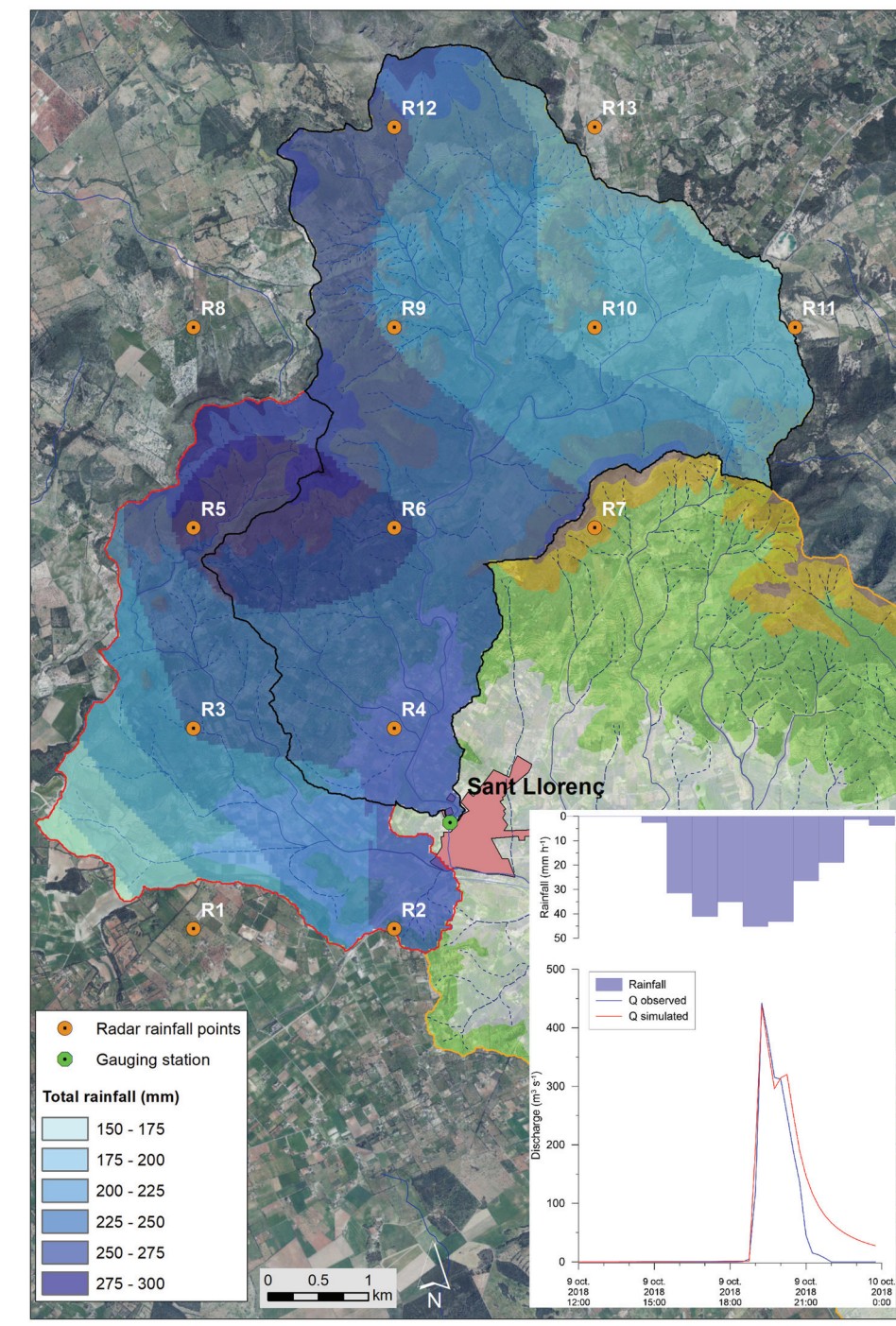

**Figure 7** Map of isohyets of the rain storm of 9[th] October 2018 in the two headwater catchments of the Ca n'Amer River; i.e. the Blanquera and Begura de Salma rivers. Source: 10-minute radar images obtained from the web https://opendata.aemet.es/. The inset figure illustrates the observed discharge measured at the MEDhyCON hydrometric station as well as the result of the rainfall-runoff simulation using a modified version of the GR3 model. Background: aerial photography and DEM data (PNOA, 2015).

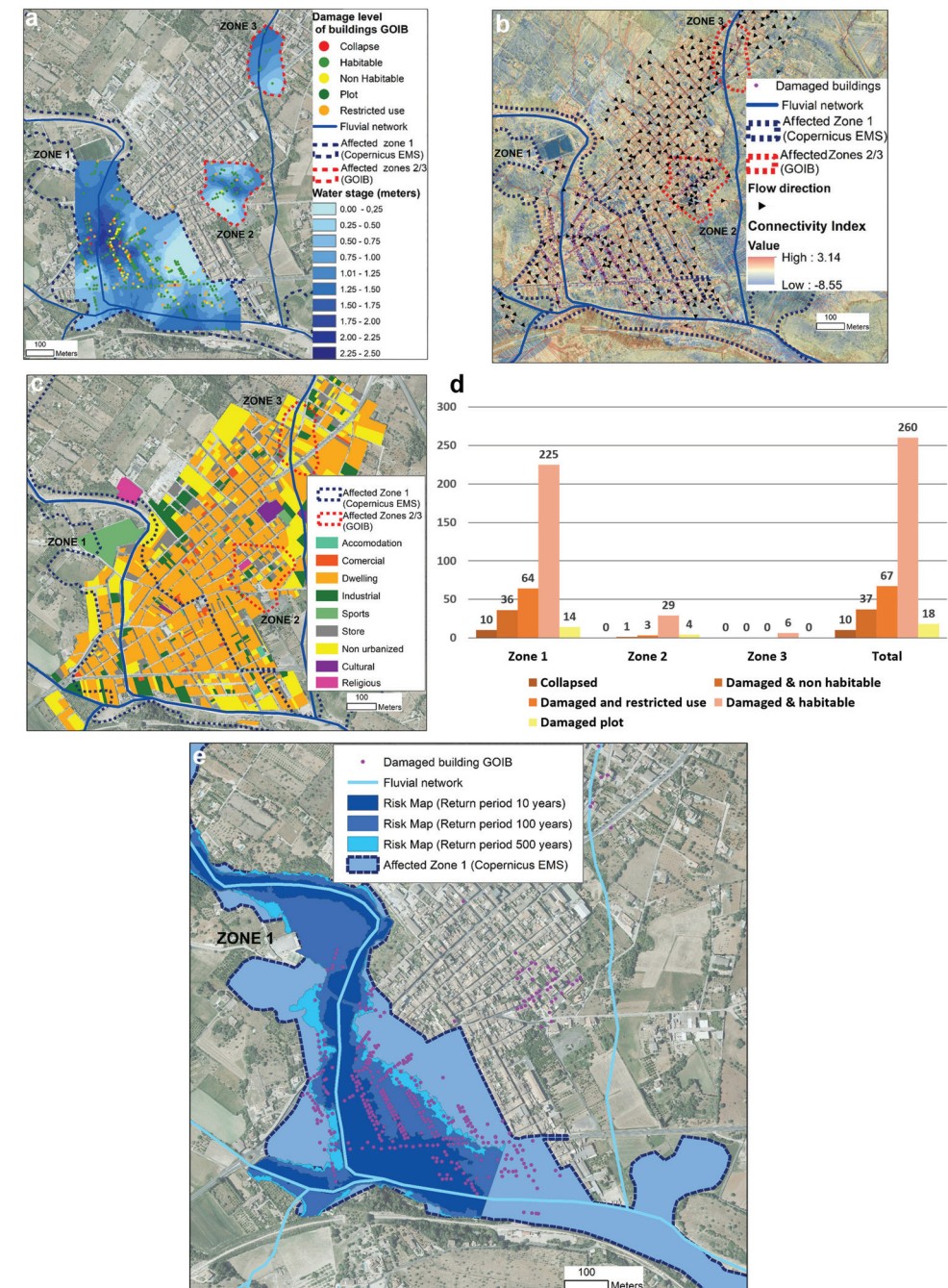

**Figure 8** (a) Map of the damage level classification of buildings and water stage reached in the different affected zones at Sant Llorenç des Cardassar according to the Balearic Islands Autonomous Government in comparison with the flood delimitation by Copernicus EMS. Flow direction and hydrological connectivity in the affected zones (b) 1, 2,and 3 in the Sant Llorenç des Cardassar urban network. (c) Economic activities at building scale in the urban area of Sant Llorenç des Cardassar and the delimitation of affected zones by the flash-flood. (d) Damage level classification of buildings in the different affected zones at Sant Llorenç des Cardassar. (e) Official flood risk maps and flood delimitation by Copernicus EMS at the Sant Llorenç des Cardassar village with the location of buildings affected by the flash flood of 9[th] October 2018. Background: aerial photography (PNOA, 2015). In Fig. 7d, the source of land uses at the urban plot scale is the General Directorate for the Cadastre (http://www.sedecatastro.gob.es/).

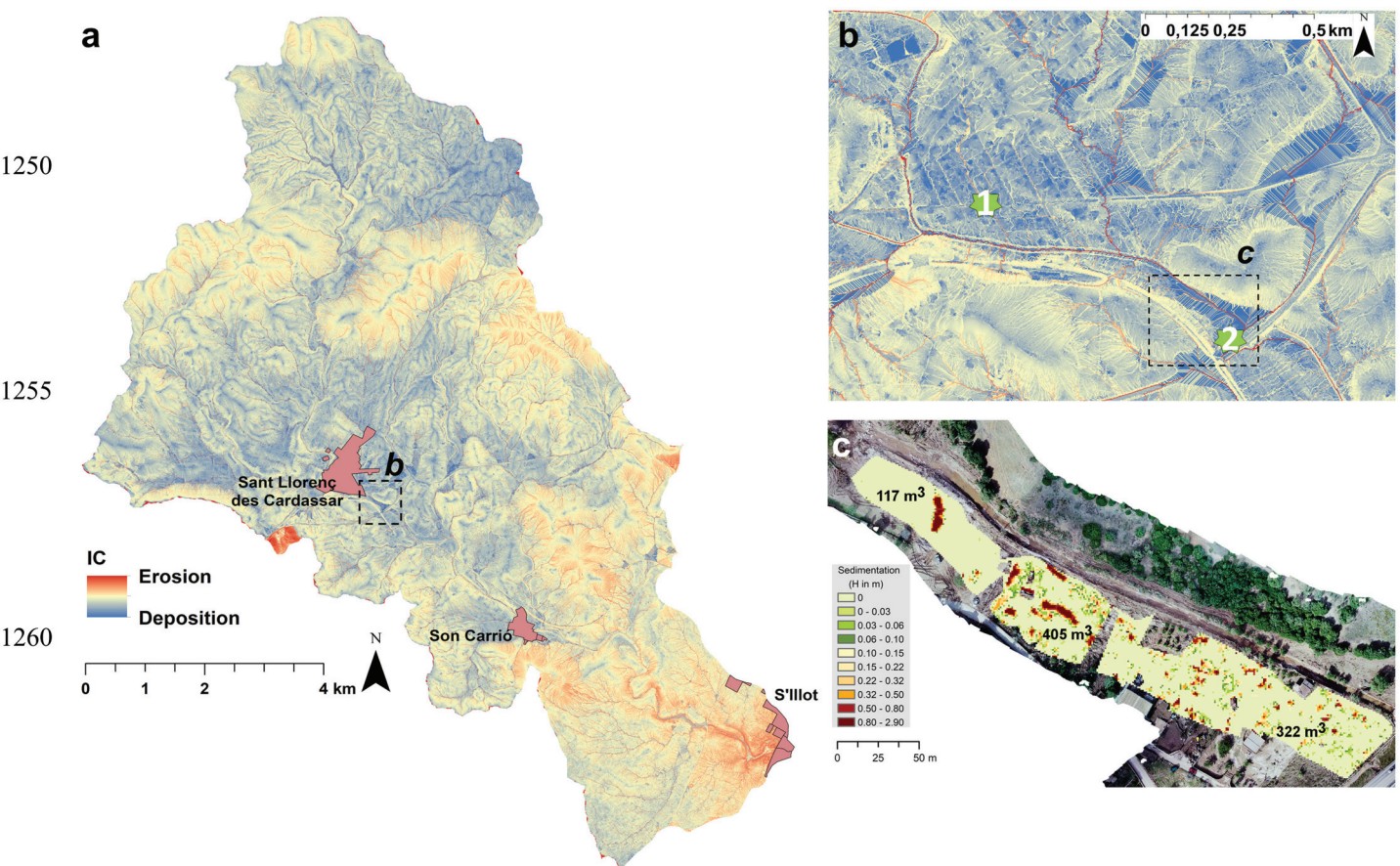

**Figure 9** Spatial patterns of hydrological and sediment connectivity (deposition zones in blue colours) (a) in the Ca n'Amer River basin, (b) in the south-east part of Sant Llorenç des Cardassar with numbers indicating (1) the point where the missing person was last seen and (2) where this person was found by using this connectivity index from a digital terrain model (MDT) of 2 m resolution (Instituto Geográfico Nacional, 2014). (c) Overbank sedimentation estimated after the flash-flood from a DEM performed with SfM from a UAV flight (15th October 2018) in relation to the ground points of the 2014 LiDAR data. Back ground aerial orthophotography of ca. 2 cm resolution obtained also from the drone images. Numbers indicate the total volume of deposited sediments in the three measured areas.