# Peer review of "Hydrogeomorphological analysis and modelling for a comprehensive understanding of flash-flood damaging processes: The 9th October 2018 event in North-eastern Mallorca"

_Natural Hazards and Earth System Sciences, 2019_

## Referee Comment (RC1) · Anonymous Referee #1 · 18 Dec 2019

The paper of Estrany et al. provides an analysis of the devastating flood that hit the North-eastern side of the Mallorca Island in October 2018, considering: 1) the hydrological response of the catchment; 2) damage assessment; and 3) geomorphic changes. The analysis presented is quite detailed, and represents a very good starting point that, linked to a study from a meteorological perspective and to a hydraulic study about the flooding dynamics (two aspects that –I acknowledge- go beyond the scopes of the paper), would provide a rather comprehensive picture of the event from a civil protection perspective. The data provided by the water level station are particularly

interesting and valuable.

My major comment is that the paper has the potential to go beyond the 'simple' description of a case study, where three single pieces (rainfall-runoff modelling, damage assessment and geomorphic changes) are discussed separately. Discussion section could help to bridge this gap. It introduces several topics (e.g., land-use changes, fires, etc.), which, however, are treated in an increasingly qualitative and general way during the discussion itself. What are their actual (and relative) effects on this event? Can they be quantified? Also, the triggering effect of the karstic reservoir(s) should be somehow addressed with more detail (I mean, the authors should try to go beyond the conceptual modelling and provide insights about the physical process, which involves, e.g., specific geological features in specific areas). I wonder about the sudden increase in discharge from 120 to 442 cms (an impressive peak flow rate per unit area) in 15 minutes (very fast response time). The reason for this behaviour is not totally clear. Is it mainly due to the karstic environment or to other reasons (e.g., the failure of a temporary dam)?

Another important point is that authors should take care of the English language and grammar. At the end of the review, I provide some examples, limited to the Abstract and the Introduction, but a thorough review should be carried out throughout the paper.

Finally, please find below a list of other specific comments. I hope that my comments help to improve the quality of the paper.

Abstract: it could be much more concise, avoiding unnecessary comments (e.g., "comprehensive analyses of catastrophic events are crucial. . ."). It is of the foremost importance that the abstract is as much straightforward as possible

L24: maybe remote sensing is better

L31: Copernicus EMS: it's better to avoid acronyms in the abstract without explanation

L45: also the interaction with the (warming) sea surface is an extremely important and

peculiar feature of the Mediterranean area (e.g., Cassola et al., 2016; Avolio et al., 2019)

L79: ok, but the main uncertainty in predictability is linked not only to hydrological uncertainty but also (mostly, I would say) to the meteorological uncertainty. This aspect should be also introduced.

L109: since the structure of the paper is complex, a brief introduction to the next Sections could be useful

L136: please explain what you exactly mean with "torrentiality". This index could be ignored by most of the audience

L193: a reference is needed to justify the sentence

L200: Please add in a figure (Fig. 2?) the radar location

L201: "due to these effects". What effects? Not clear. Please explain and justify with adequate reference(s)

LL206-213: this paragraph is not clear. What were the driving data for this analysis? Those provided by the rain gauges surrounding the catchments? If so, the authors should provide some proofs about the reliability of their analysis (e.g., a scatterplot, even in the Supplement).

Furthermore, if rainfall data are gridded in 2x2 square cells, why in fig. 6 it looks like they are spatially interpolated? Maybe, because they are related to the GSM-SOCONT scheme (for which the authors refer to inverse distance weighting)?

L218: When introduced, the acronym MEDhyCON (such as any other) should be explained

L251: "a second hydrograph was designed": with what Q values?

L262: I would like some more details about the percentages of obstruction detected

and how they were calibrated. Do they rely (only) on the three ground control points? Why those points? When/how were the water levels measured on them? Does the simulation consider 0% obstruction up to below the bankfull and 85% immediately after? No transient state?

L321: in my opinion, the best tool for this kind of assessment would have been a complete 2D hydraulic study. Please discuss briefly your choice and its advantages (e.g., it's time-saving, etc.)

Table 1 needs more explanation. Terms like IPmax should be explained

Figure 7 is not very clear. Maybe it could be divided into more figures. However: in Fig. 7a, are the red dashed polygons all derived from the Copernicus EMS? Also zone 2 and 3? Do the latter perfectly correspond to the Government survey? Figs. 7b and 7c are not very readable/useful, in my opinion.

LL510-511: I guess it is Fig. 8c.

LL535-536: "Despite these antecedent . . . as reported by these authors." Why?

LL547-548: it's not clear why the authors need to adjust the initial conditions manually. Does the model not perform well if used for long periods?

LL556-585: the discussion introduces many arguments in a general and qualitative way. I suggest to skip/reduce much of the discussion (especially that about the weather predictability, which is not addressed in this study) and/or try to quantify the different effects (please refer to my main concern).

English language and grammar review:

L25: at the catchment scale

L26: peak discharge of 442. . .

L28: "i.e." not needed

L38: For that reason, they usually affect/impact basins...

L46: very close to the coastline

L48: "scarce soils". What do the authors exactly mean?

L51: elucidates

L51: "the hydrological processes from an extreme flood"? Do you mean the hydrological processes activating during an extreme flood or so?

L54: small spatial scale usually means low-resolution (e.g., scale 1:50000 is a smaller scale than 1:500). Please rephrase to make the sentence clearer

LL67-68: in order to reduce the uncertainty of the Q estimate, I suppose

L69: "... also adding that..." please rephrase

L87: regarding vulnerability

L88: "understanding... are developed". Understanding is not developed. Please rephrase

L92: evaluates

L97: "with that Copernicus EMS one". Not clear

L112: why "precipitation as well as the Q"? I would write either "Precipitation and discharge" or "P and Q"

Captions: please check also all figure captions (e.g., "left" and "right" pictures in Fig. 3, in Figure 7 there are two references to (f) )

Supplement: please check grammar also here (e.g., L'independent; "Schema"). The caption of Figure S2 should declare the meaning of the variables.

Cited references:

Cassola, F., et al.: The role of the sea on the flash floods events over Liguria (north-western Italy). https://doi.org/10.1002/2016GL068265, 2016.

Avolio, E., et al.: Brief communication: Preliminary hydro-meteorological analysis of the flash flood of 20 August 2018 in Raganello Gorge, southern Italy. https://doi.org/10.5194/nhess-19-1619-2019, 2019.

---

## Short Comment (SC1) · 29 Dec 2019

After a careful reading of this manuscript I have several questions and concerns about the data (rainfall, discharge) and methods (modelling) the authors used for analysing the flash-flood occurred in October 2018 in eastern Mallorca: 1. The abstract states: "Continuous streamflow monitoring data revealed a peak discharge of 442 cumecs"... but it is not clear if the event was continuously monitored, as in line 226 the authors say "with the absence of direct flow measurement for the Q estimation"; and in line 382 they say that the hydrological monitoring period excluded the October 2018 month,

when the flash flood occurred. Therefore, there was no direct measurement during the flash flood and this statement cannot be used neither in the abstract nor in the rest of the manuscript. 2. Related to the discharge and following line 226, with the absence of direct flow measurements, the next line explains that the authors applied the two-dimensional hydraulic model HEC RAS 5.0.6. However, where this simulation can be found in the paper? 3. The peak discharge of the flash flood was estimated through a stage-discharge rating curve (figure 4) calibrated with only two events with values over 1 cumec. Therefore, the estimated value of the maximum discharge is suspicious or may contain a significant error. There is a study made by Lorenzo-Lacruz et al. (2019) (https://doi.org/10.5194/nhess-19-2597-2019) addressed on the same event, and published in NHESS one month ago. In this published article, Lorenzo-Lacruz et al. obtained peak discharges of 306.9 cumecs and 303.4 cumecs with FEST and KLEM models, respectively, and successfully validated it with in situ measurements after the event. As a new contribution, Estrany et al. must discuss the significant difference with the result they have obtained (442 cumecs). Nevertheless, the authors not even mention the article of Lorenzo-Lacruz et al., (2019), in spite of it was published for open discussion three months before Estrany et al. submitted its manuscript for discussion and review. 4. Figure 6 show the total rainfall during the flood event. It is weird how the authors distributed the rainfall showing that the highest values were recorded in the western-central part of the basin and not in the headwaters, where the relief is higher (around 400 meters). The figure also shows 13 rainfall radar points for what purpose? to calibrate the rainfall? In Lorenzo-Lacruz et al. (2019) they show a complete reconstruction of the radar data for the same flood event and it is completely different to the data depicted in Figure 6. Lorenzo-Lacruz et al. draw the highest values (up to 400 mm) on the headwaters. This highest rainfall recorded in the upper parts of the catchment would explain why the flash flood also occurred in the surrounding basins (barranc de Sa Canova and s'Ametllerar), which share the headwaters with the studied catchment. 5. The authors used the hydrological modelling GSM-SOCONT and it is also surprising, since this model is "a
semi-distributed glacio-hydrologic model to simulate daily discharge from catchments with glacier cover" (https://www.mathworks.com/matlabcentral/fileexchange/43452-gsm-socont-glacio-hydrological-model). Surface runoff was computed with the SWMM (a conceptual glacio-hydrological model for high mountainous catchments). It seems not to be the most appropriate model for this watershed and for a catastrophic flash-flood analysis or reconstruction. There are other studies addressed in the Mediterranean region in which more appropriate models for flash flood episodes were used (Amengual, A., Homar, V. and Jaume, O., 2015: Potential of a probabilistic hydrometeorological forecasting approach for the 28 September 2012 extreme flash flood in Murcia, Spain. Atmospheric Research, 166, 10-23. Amengual and Carrio, 2017: A Comparison of Ensemble Strategies for Flash Flood Forecasting: The 12 October 2007 Case Study in Valencia, Spain https://doi.org/10.1175/JHM-D-16-0281.1). 6. Therefore, it is embarrassing to say that the "observed and modelled hydrograph and corresponding peak flow (442 cumecs)" showed in Figure 6 contains major flaws as: i) there is no observed hydrograph, ii) a glacio-hydrological model was used for a small Mediterranean watershed with low elevations and semiarid conditions, and iii) what is more surprising, the peak flow is reached one hour before the peak of rainfall. Is there any reasonable argument to defend such hydrological behaviour? 7. It is at least strange that this study not even mention the work by Lorenzo-Lacruz et al. (2019) "Hydro-meteorological reconstruction and geomorphological impact assessment of the October, 2018 catastrophic flash flood at Sant Llorenç, Mallorca (Spain)", accepted for publication in October 22nd and published one month ago in NHESS. The study by Lorenzo-Lacruz et al. can be used for discussion about the different results that Estrany et al. obtained on rainfall data, peak discharge, hydraulic and hydrological modelling, geomorphological reconstruction, etc. This is, indeed, the objective of an open discussion journal. Explicitly speaking there is a resemblance of some of the content of this contribution (Estrany et al.) with the contents of Lorenzo-Lacruz et al. (2019). Not only the title and structure of the manuscript are similar, some figures included in this paper show a direct and evident inspiration on the published paper by Lorenzo-Lacruz

et al. (i.e.: figure 2 of the published paper and figure 2 of this manuscript; figure 6 and figure 7; figure 9.c of the published paper with figure 8.c of this manuscript). In science, the right of any particular to carry out research freely on any topic is respected, of course, but it is also necessary to cite and mention the work already done by others to avoid plagiarism.

―――――――――――――――――――――

---

## Author Comment (AC1) · 27 Jan 2020

The response of the questions and comments provided by Dr. Jorge Lorenzo-Lacruz will be addressed in a point by point format.

GENERAL COMMENT OF DR. LORENZO-LACRUZ. After a careful reading of this manuscript I have several questions and concerns about the data (rainfall, discharge) and methods (modelling) the authors used for analysing the flash-flood occurred in October 2018 in eastern Mallorca

[Figure]

We are very grateful to Dr. Lorenzo-Lacruz for this short comment because it provides an interesting opportunity to underpin the research work carried out by the MEDhyCON Team in collaboration with a multi-disciplinary group of scientists from different academic institutions in Spain and a spin-off from Switzerland. Beyond the continuous streamflow monitoring in the Begura de Saumà River hydrometric station, it is also worth saying that most of the data exposed in this paper was collected in the very intense fieldwork campaign during the fortnight following the catastrophe, in which the MEDhyCON Team was fully collaborating in search and rescue tasks as required by the Directorate General of Emergencies of the Balearic Islands Government to help in finding the last missing victim. We are therefore in a unique position to clarify some questions which have arisen about this particular event.

The research group in Hydrology and Ecogeomorphology in Mediterranean environments –MEDhyCON (http://medhycon.uib.cat) is part of the Institute for Agro-Environmental Research and Water Economy – INAGEA (http://inagea.com) at the University of the Balearic Islands (Spain). During the last decade, we have been working on the installation of a hydrometric monitoring network throughout the island of Mallorca. The deployment of our network was specially motivated by the lack of maintenance and the structural deficiency (cf. Fortesa et al., 2019) of the official network managed by the Autonomous Government of the Balearic Islands. It is important to note that the implementation of this new network by MEDhyCON, in some cases, necessitated taking advantage of some useful abandoned parts of the aforementioned old official network. This research infrastructure has been crucial for going beyond in the scientific assessment of global change impacts on Mediterranean catchments hydrology (see a list of SCI papers published by the Group at http://medhycon.uib.cat/publications.html).

Mallorca is a Mediterranean flood-prone region historically affected by devastating flash floods, which have systematically been well documented but always ungauged (Grimalt and Rosselló-Geli, 2011; Estrany and Grimalt, 2014; Petrus et al., 2018). Under this

geographical framework, MEDhyCON research team deployed its hydrometric network in some of the catchments most affected by these catastrophic events. As a result, our hydrometric monitoring network is currently composed of 34 stations (see Fig. 1 of this point-by-point document) with digital probes to measure the water stage by readings of 1 minute intervals accumulating in 15-minute average values. This type of monitoring provides continuous streamflow records.

One of these stations was installed in June 2015 in the Begura de Saumà River hydrometric station, located in Sant Llorenç des Cardassar village (see Fig. 1 of Estrany et al. MS). The small spatial and temporal scales of flash floods, relative to the sampling characteristics of typical hydro-meteorological networks, make these events particularly difficult to monitor and document. A lot of flash flood events occurred in ungauged catchments. Even in those catchments with hydrometric stations, streamflow monitoring can be damaged during these catastrophic events. This was not the case of the worst natural disaster that occurred in Mallorca during the Late Modern period, because the water stage was completely recorded by our pressure probe installed in a bank of the artificial channel where the water stage is the same by "connected vessels" but avoiding potential damage (see Fig. 2 of this point-by-point document). It is a historical landmark, being the first time that real data is recorded in the Balearic Islands in such type of events.

References used in this comment

Estrany, J. and Grimalt, M.: Catchment controls and human disturbances on the geomorphology of small Mediterranean estuarine systems. Estua. Coast. Shelf. Sci, 150: 1–12. DOI: 10.1016/j.ecss.2014.03.021, 2014.

Fortesa, J., García-Comendador, J., Calsamiglia, A., López-Tarazón, JA., Latron, J., Alorda, B. and Estrany J.: Comparison of stage/discharge rating curves derived from different recording systems: Consequences for streamflow data and water management in a Mediterranean island. Sci. Total Environ., 665: 968-981. DOI:

10.1016/j.scitotenv.2019.02.158, 2019.

Grimalt, M., Rosselló-Geli, J.. Anàlisi històrica de les inundacions a les Illes Balears. (Translation to English: Historical analysis of floods in the Balearic Islands; following the methodological guide developed by the Spanish Ministry of Agriculture, Food and Environment). Conselleria d'Agricultura, Medi Ambient i Territori del Govern de les Illes Balears, 442 pp + Atlas, 2011.

Petrus, JM., Ruiz, M and Estrany J.: Interactions between Geomorphology and Urban Evolution Since Neolithic Times in a Mediterranean City. Urban Geomorphol., 9–35. DOI: 10.1016/B978-0-12-811951-8.00002-3, 2018.

POINT 1 LORENZO-LACRUZ. The abstract states: "Continuous streamflow monitoring data revealed a peak discharge of 442 cumecs"... but it is not clear if the event was continuously monitored, as in line 226 the authors say "with the absence of direct flow measurement for the Q estimation"; and in line 382 they say that the hydrological monitoring period excluded the October 2018 month, when the flash flood occurred. Therefore, there was no direct measurement during the flash flood and this statement cannot be used neither in the abstract nor in the rest of the manuscript.

Our hydrometric station located at Sant Llorenç des Cardassar provided a complete and continuous water stage recording of the extreme flash-flood event, as we have explained in the previous general comment. We are really surprised by this comment because it denotes a lack of knowledge about one of the basics in surface hydrology: the estimation of river discharges. In a hydrometric station, discharges are traditionally estimated from the continuous measurement of the river's water stage. As is explained in the sub-section 2.3 Discharge data, "the MEDhyCON Research Group installed within the gauge house a Hobo Water Level U20L-04, which measures the water stage by readings of 1 minute accumulating 15-minute average values" (Lines 218-219). Therefore, the continuous measurement of water stage is directly related to the continuous streamflow monitoring.

It is obvious that the hydrological monitoring period also included the October 2018 month, when the flash flood occurred. However, to better assess the hydrological dynamics of the catchment, we structured these hydrological results into two sub-sections. The first one is the 3.1 Catchment hydrological dynamics, where these dynamics were described during almost 4 hydrological years. The second one is the 3.2 Hydrological response of the flash-flood, where the continuous streamflow monitoring allowed evaluating this response with a high accuracy. This structure is also linked with the performance of two different figures for each sub-section where the NHESS readers can directly observe the ephemeral regime during the monitoring period 2015-2018 (Fig. 5 of Estrany et al. MS) and the difference of three units of magnitude in the observed discharge during the catastrophic flash flood if compared with the previous monitored period (Fig. 6 of Estrany et al. MS).

Regarding Line 382 and considering the explanation provided in the previous paragraph, we must however thank to Dr. Lorenzo-Lacruz because the writing could generate confusion to the NHESS readers. We will therefore change the sentence in the revised version of the Estrany et al. MS.

The original version was:

"The hydrological monitoring period in the hydrometric station was from 10th January 2015 to 30th September 2018 (Fig. 5), excluding the October 2018 month when the catastrophic flash flood event occurred (see results in sub-section 3.2)".

The modified version will be:

"The hydrological monitoring period assessed in this paper by using data from the hydrometric station was from 10th January 2015 to 31st October 2018 (Fig. 5). In addition, the October 2018 month was reserved to develop a singular and deeper study in order to better describe the catastrophic flash flood event (see results in the next sub-section 3.2)".

POINT 2 LORENZO-LACRUZ. Related to the discharge and following line 226, with the absence of direct flow measurements, the next line explains that the authors applied the two-dimensional hydraulic model HEC RAS 5.0.6. However, where this simulation can be found in the paper? Linked to the previous Point 1 provided by Dr. Lorenzo-Lacruz, it is essential to emphasize that specific discharge measurements, known as gaugings, are needed to establish a relationship between the water stage and the discharge. This is called a stage-discharge rating curve, and its purpose is to translate water stages into discharges (Chow, 1964; Braud et al., 2018).

In the Begura de Saumà hydrometric station we did not perform specific discharge measurements by using direct flow velocity measurements, although during flash-flood events it is not possible to carry out these measurements with traditional tools; i.e., current meters. Therefore, rating curves are often extrapolated for high flows, far beyond the range of gauged values, which leads to very high uncertainties regarding flood discharges. In order to avoid these uncertainties, a two-dimensional hydraulic model was applied as a tool to perform and evaluate the stage-discharge rating curve as well the peak discharge during the event. The 2D hydraulic model was also thoroughly described between lines 224 and 272 and Fig. 3 of Estrany et al. MS. Data used for the DEM and mesh construction, area and length of the studied reach, location of significant structures and hypothesis regarding its hydraulic behaviour, boundary conditions, calculation method and simulations running in order to obtain the stage-discharge rating curve are shown in Fig. 4 of Estrany et al MS. All significant factors and processes regarding the simulation assumptions and hypothesis were also explained.

It is worthy of note that the main scope of our paper is not the hydraulic modelling of the event, it is focused on addressing other features such as the hydrological response of the catchment and its modelling, the damage assessment and geomorphic changes.

Reference used in this comment

Braud, I.. Vincendon, B., Anquetin, S., Ducrocq, V. and Creutin J-D.: The Challenges

of Flash Flood Forecasting. In Mobilities Facing Hydrometeorological Extreme Events 1. Elsevier; 63-88, 2018.

Chow V Te.: Handbook of applied hydrology; a compendium of water-resources technology. McGraw-Hill, 1964.

POINT 3 LORENZO-LACRUZ. The peak discharge of the flash flood was estimated through a stage-discharge rating curve (figure 4) calibrated with only two events with values over 1 cumec. Therefore, the estimated value of the maximum discharge is suspicious or may contain a significant error. There is a study made by Lorenzo-Lacruz et al. (2019) (https://doi.org/10.5194/nhess-19-2597-2019) addressed on the same event, and published in NHESS one month ago. In this published article, Lorenzo-Lacruz et al. obtained peak discharges of 306.9 cumecs and 303.4 cumecs with FEST and KLEM models, respectively, and successfully validated it with in situ measurements after the event. As a new contribution, Estrany et al. must discuss the significant difference with the result they have obtained (442 cumecs). Nevertheless, the authors not even mention the article of Lorenzo-Lacruz et al., (2019), in spite of it was published for open discussion three months before Estrany et al. submitted its manuscript for discussion and review.

First of all, the stage-discharge rating curve shown in Fig. 4 of Estrany et al. MS is not the result of a calibration made with only two events with values over 1 m3 s-1. This stage-discharge rating curve was obtained by means of the two-dimensional hydraulic model described in sub-section 2.3, simulating the 2D flow hydrodynamics with the required free surface flow equations; i.e., Shallow Water Equations.

A hydrometric gauging station is located at the beginning of the artificial channelization of the Begura de Saumà River that crosses the village of Sant Llorenç des Cardassar, as was explained in detail within sub-section 2.3. However, in this case, water stage measurements cannot be directly transformed into discharges by using a typical stage-discharge rating curve, despite this being the normal procedure in gauging stations. During the 9th October event, two main factors could have led to wrong results by using this transformation. Firstly, as the flow overbanked the artificial channelization, the cross section used in the traditional stage-discharge rating curve is no longer a rectangular section, although the entire cross section of the flooded area must be completely incorporated into the discharge estimation. Secondly, as it was also carefully explained, several bridges cross the artificial channelization. Specifically, one of them is located 50 m downstream of the hydrometric station (see Fig. 3a and sub-section 2.3 in Estrany et al. MS). This bridge was severely obstructed by vegetation during the event (see inset pictures in Fig. 3 in Estrany et al. MS). The obstruction influenced the channel hydraulics also creating backwater effects that altered the stage-discharge relationships, causing an increase of overflow in sections located upstream of the bridge.

Accordingly, it is not possible to apply a unique theoretical stage-discharge rating curve due to the necessity of having to apply a different approach to compute the stage-discharge rating curve during the flash flood event.. The complete geometry of the study area, including bridges, were implemented in a 2D hydraulic model, being run with different boundary conditions (different discharge values entering the system). In addition, for each of these conditions, the water stage –calculated at the location of the hydrometric station– was retrieved. It can be observed that if the flow depth is below the low chord of the first bridge's deck, the resulting stage-discharge rating curve follows the theoretical expression, according to the channel characteristics. However, once the water stage reaches the first bridge's deck, backwater starts affecting the hydraulic flow behaviour at the gauging station, and the streamflow starts overbanking the channel section. The influence of bridges is therefore well evaluated and represented in the stage-discharge rating curve in our paper, performing a 'jump' in the water stage.

Moreover, during the event of 9th October, the partial blockage of some of the bridge's openings (clearly visible in the post-event pictures in Fig. 3 of Estrany et al. MS) also occurred. One of the main uncertainties regarding the discharge simulation of the flash-flood event is the level or degree of obstruction at each bridge. In this case,

we carefully implemented a model calibration parameter, estimated by observing photographs taken just after the event; i.e., 12 hours after, 10th October morning. The calibration procedure was carried out with the maximum water stages measured at significant locations (see Fig. 3b in Estrany et al. MS). With the values described in lines 262 and 263, the simulated water stages matched these ground control points with and error < 5% (even < 1% in the case of the water stage recorded at the hydrometric station). With a small increase or decrease in the flow discharge, these errors are rapidly increased.

Regarding to the discussion proposed by Dr. Lorenzo-Lacruz on the discrepancy in peak discharges between our manuscript (Estrany et al.) and those obtained in the paper Lorenzo-Lacruz et al. (2019), we have carefully read the methods and results sections. Thus, Lorenzo-Lacruz et al. (2019) explained that peak discharge was obtained with FEST and KLEM models (fully distributed hydrological models), being validated with in situ measurements after the event. There is a clear lack of information on this key method issue, although we can assume a hypothetical validation by means of hydraulic modelling of those peak discharges. However, there are some important issues within the model that Lorenzo-Lacruz et al. (2019) used that, from our point of view, invalidate the obtained results.

First, it is stated that Lorenzo-Lacruz et al. (2019) applied a 1D model (Line 231). The software is the same (HEC-RAS) used in our paper, but 1D models are strongly not-recommended for the simulation of flood events. Once the flow overbanks the main channel, this flow can take multiple directions, and hence have varying water surface elevations and velocities in each of the directions. The HEC-RAS 2D Modelling User's Manual (2015) indicates ex professo this recommendation. Therefore, the 1D models only incorporate one main direction of the flow (in this study case, the direction of the channel). However, with overbank flows, a significant second direction is starting to take place.

1. Lorenzo-Lacruz et al. (2019) clearly describes how the water overflowed the channelization from the beginning of the flash-flood event:

Line 362: "The hydraulic simulation (Fig. 4) shows how water starts flowing through Sant Llorenç town at 1900 h (LT) and, in barely 10-20 minutes, it starts overflowing the artificial channel at several points within the sector between bridge#2 and bridge#3, reaching 365 depth values of about 3 m. The football field locate in the meander between bridge#2 and bridge#3 was completely flooded in less than half an hour".

2. They also described how the overflows reached the centre of the village which is a complex net of buildings and streets. As a note in urban geography, we must say that the old village where the centre is located was not affected by the overflows. In terms of hydraulics, urban environments are characterized by numerous transitions to super-critical flow and numerical shocks. As a result, the simulation of the flow behaviour during this catastrophic flash-flood event, a 2D model is required (Hunter et al., 2008).

Line 366: "It only takes between 10 and 20 minutes for the peak flow modelled at the entrance of the town to reach the maximum extent of flooded area in the city centre: water covers the entire longitudinal path of the channel through town, with the most affected areas in the vicinities of bridges and river corners".

Line 372: "Water depth reached almost 1.5 m in the Town Hall square and the surrounding blocks, which are located 150-200 metres away from the main river channel".

3. Furthermore, they highlighted the need to excessively increase the width of the cross-sections in order to reach the points located in areas affected further away from the channelization (it must be noted that the width of the channel is just 10 m).

Line 267: "The number of cross-sections was limited by the meandering shape of the Ses Planes river at Sant Llorenç and the extraordinary extent of flooding in the city centre (cross-sections wider than 300 m were required to cover the affected areas at some points)".

Although there is no universal rule to determine whether using a 1D or a 2D model,

some practical rules can be applied. For instance, when the cross-section width is longer than 50 times the water depth, a 2D model is recommended. However, if the flow itself is clearly 2D (flow at channel bends, flow with significant velocity differences between channel and floodplain, flow divergence, or flows in urban areas), a 2D model is compulsory (ACA, 2003).

Therefore, it is difficult to understand why the authors chose a 1D model to represent a flooding process that is clearly 2D, and why they affirm that results obtained with that model can be used to validate the peak discharge obtained with the hydrological model. In line 495, Lorenzo-Lacruz et al. (2019) introduced the possible inaccuracies caused by the 1D approach:

Line 495: "In this context, the use of the 1-D approach for analysis of bridge effects may not properly simulate the interactions between bridge and flow, and backwater effects may be underestimated or neglected (Costabile and Macchione, 2015). This seems to not be the case for Sant Llorenç since water velocity greatly increased at bridges, and sections between bridges #2 and #3 were flooded for a longer time, denoting that the model registered the occurrence of intense backwater processes at these bridge locations".

However, 1D models are perfectly useful for simulating flow through bridges, for both low and high water stages. This cannot be the reason why the 1D model should or should not be considered valid. As the flow overbanked the main channelization occupying the urban area, this is the clear reason that a 1D model is not valid to simulate this catastrophic flash-flood event. The difference between Lorenzo-Lacruz et al. (2019) peak discharge –peak discharges of 306.9 m3 s-1 and 303.4 m3 s-1 with FEST and KLEM models– and Estrany et al. MS peak discharge (442 m3 s-1) can be easily explained. Beyond Estrany et al. MS directly measured the water stage by a pressure probe, the use of 1D models underestimate discharges in the case of overbanking flood events, because one of the two main directions of the flow is being neglected (and therefore, not all the water volume flow is estimated).

Additionally, regarding the use of KLEM model, it is also known that the rescaled width function (which is the root of the KLEM model) is sensitive to the method of channel network extraction (Mutzner et al., 2016). This is particularly true if the drainage density is uneven. Lorenzo-Lacruz et al. (2019) used a threshold area method without giving neither the value of the threshold, nor the resolution of the DEM used to extract the network. However, the threshold area method assumes a constant drainage density. The magnitude and the ratio of the hillslope and channel velocities are also determinant and also have a large influence on the timing and peak of the modelled flood (Mutzner et al., 2016). Then of course the calibration of these two parameters helps to obtain a very good agreement with the timing of observed discharge (if existing).

Some other relevant comments can be made about Lorenzo-Lacruz et al.'s hydraulic model:

1. In Line 265 "The HEC-RAS model was set up using 40 cross-sections distributed along the 4303 m of reach length". This separation distance between cross-section performed more than 100 m distance between sections when the recommended distance should not exceed 20-25 m. Lorenzo-Lacruz et al. (2019) explained this selection in:

Line 267: "The number of cross-sections was limited by the meandering shape of the Ses Planes river at Sant Llorenç and the extraordinary extent of flooding in the city centre (cross-sections wider than 300 m were required to cover the affected areas at some points), since cross-sections could not intersect each other".

In our opinion, this confirms the need to use a 2D model, instead of a 1D model that does not accurately represent the geometry of the study area.

2. It was not specified in Lorenzo-Lacruz et al.'s paper how the model assesses the vegetation blockage that occurred in some bridges, especially those bridges located very closed to the hydrometric station, although they do highlight this fact in the paper:

Line 402: "The bridge (pictured in Figure 6), which occupies more than half the flooded section, blocks the water and increases its velocity upstream due to flux accumulation and generation of intense backwater effects caused by bridge clogging".

Line 511: "Along the study reach, several trees were eroded, transported and deposited downstream, arriving at the first bridge at the entrance of the village".

3. In Line 282 "The computation time step was set up at 10 minutes (same temporal resolution as the discharge hydrograph) and the output time interval was set at 1 minute". The computational time step should be determined according to numerical computational criteria, such as Courant's condition, not to the temporal resolution of the input boundary condition. Computational time step and the temporal resolution of the hydrograph are two completely different parameters. Such a large interval may lead to instabilities and inaccuracies. Furthermore, it makes no sense to retrieve output results in time steps that are smaller than the computational time step.

In view of all these scientific evidences, the hydraulic model used in Lorenzo-Lacruz et al. (2019) cannot be considered adequate to validate any maximum discharge hypothesis.

References used in this comment

Hunter NM, Bates PD, Neelz S, Pender G, Villanueva I, Wright NG, Crossley AJ 2008. Benchmarking 2D hydraulic models for urban flooding. In Proceedings of the Institution of Civil Engineers-Water Management Vol. 161, No. 1: 13-30. Thomas Telford Ltd.

Engineers, U.A.C.O. 2015. HEC-RAS River Analysis System, 2D Modeling User's Manual. Agència Catalana de l'Aigua. Departament de Medi Ambient 2003. Guia Tècnica. Recomanacions tècniques per als estudis d'inundabilitat d'àmbit local.

Mutzner R, Tarolli P, Sofia G, Parlange MB, Rinaldo A. 2016. Field study on drainage densities and rescaled width functions in a high-altitude alpine catchment, Hydrol. Process, 30: 2138-2152. DOI: 10.1002/hyp.10783.

[Figure]

POINT 4 LORENZO-LACRUZ. Figure 6 show the total rainfall during the flood event. It is weird how the authors distributed the rainfall showing that the highest values were recorded in the western-central part of the basin and not in the headwaters, where the relief is higher (around 400 meters). The figure also shows 13 rainfall radar points for what purpose? to calibrate the rainfall? In Lorenzo-Lacruz et al. (2019) they show a complete reconstruction of the radar data for the same flood event and it is completely different to the data depicted in Figure 6. Lorenzo-Lacruz et al. draw the highest values (up to 400 mm) on the headwaters. This highest rainfall recorded in the upper parts of the catchment would explain why the flash flood also occurred in the surrounding basins (barranc de Sa Canova and s'Ametllerar), which share the headwaters with the studied catchment. Regarding the rainfall estimation process using radar data we would first like to emphasize that in our case we only used the lowest available radar PPI, which performed a vertical inclination of 0.5°. This PPI is justified because we only used the radar data that we downloaded during the event thanks to the Open Data of AEMET (https://opendata.aemet.es/centrodedescargas/inicio). In contrast, Lorenzo-Lacruz et al. (2019) used in their paper two PPI, the one showing a vertical inclination of 0.5° and the one with 1.5°.

It is possible that some of the differences mentioned by Dr. Lorenzo-Lacruz in the rainfall estimations by Lorenzo-Lacruz et al. (2019) and our rainfall estimation emerged due to the differences in the processed data. Furthermore, Lorenzo-Lacruz et al. (2019) decided to implement a technique designed to correct the orographic-blocking of the radar signal to their data. In Estrany et al. MS, we decided not to implement this procedure because we did not detect major blockings to the signal, after calculating the trajectory of the 0.5° PPI and considering the location of the radar. In this sense, it is a bit worrying to observe the radar location that Lorenzo-Lacruz et al. (2019) used because in their Fig. 1 the rain radar is incorrectly located. It seems that they located the rain radar at Palma Airport, but the AEMET rain radar is located almost 25 km faraway, in a site called Cap Blanc. However, at Palma Airport an AEMET radar of the lightning detection system is installed, but the AEMET rain radar is located at Cap

Blanc. This information can be found, for instance, in many messages of the official twitter account of @AEMET_Baleares, the entity which owns and manages the rainfall radar. (https://twitter.com/AEMET_Baleares/status/798930116969644032).

It is also worthy of note that while Dr. Lorenzo-Lacruz mentioned a "complete reconstruction of the radar data" in his comment, we consider more precise the expression "estimation process". We are aware that two different estimation processes can lead to two different estimation quantities, as it occurs with our work. In this sense, and surprisingly for us, Lorenzo-Lacruz seems to assume in this comment that a rainfall episode always shows a strong relief influence. In Mallorca, it is true that an orographic effect can generate an increase over the precipitation total amounts. However, this process is commonly restricted to some synoptic situations in the Tramuntana Range; i.e., northeastern winds (Sumner et al. 1995; Llop and Alomar, 2012). In reality, this process is very effective when the rainfall is heavily influenced by the uplift generated in the surface by the orography, which can or cannot occur during a convective episode. Despite Cuevas-Tascón et al. (2019) explaining how a convergence front in the mesoscale probably existed, the exact location of this convergence front it is still unknown. Accordingly, the convergence front was a key factor in explaining the persistence of heavy rainfall and the total rainfall amount, but it is not possible to assume that the highest amount of rainfall should have been estimated at the higher relief areas of the Begura de Saumà River catchment, as Dr. Lorenzo-Lacruz assumes in his comment.

Regarding the 13 points showed in the Fig. 6 of Estrany et al. MS we want to point out that these points corresponded to the centroids of our rainfall estimation and they were used to generate the map of isohyets as well as to obtain the estimation of rainfall at each time step.

Regarding the following part of comment 4 "This highest rainfall recorded in the upper parts of the catchment would explain why the flash flood also occurred in the surrounding basins (barranc de Sa Canova and s'Ametllerar), which share the headwaters with the studied catchment". We do not fully understand the aim of this comment. It is true

that a flash-flood also occurred in surrounding catchments, but with peak discharges much lower than the peak discharge recorded in the Begura de Saumà River (see Fig. 3 of this point-by-point document). The inset figure illustrates the rain fallen the 9th October 2018 in the middle and downstream parts of Na Borges River catchment as well the continuous streamflow calculated from the MEDhyCON Research Group at the (3) Sa Vall and (4) Ses Pastores hydrometric stations. Peak discharges were >10 m3 s-1 during the rain event of 9th October 2018, at least two order of magnitude lower than the peak discharge recorded in the Begura de Saumà River. With such intense rainfall recorded in the upper parts of the catchment as stated by Lorenzo-Lacruz et al. (2019), the MEDhyCON hydrometric station located in Ses Pastores (number 4 in Fig. 3 of this point-by-point document) probably experimented a flash-flood peak discharge of a similar magnitude to the one recorded in the Begura de Saumà River. Nevertheless, the peak discharge recorded by the MEDhyCON gauging station was one order of magnitude lower. In our opinion, this reinforces the spatial distribution of the rainfall obtained in our paper against the one presented by Jorge-Lorenzo et al. (2019).

Furthermore, in this specific part of his comment, Dr. Lorenzo-Lacruz seems to underestimate the possibility of rainfall amounts ca. 150-200 mm to generate flash-flood events. This is quite curious, as in their own paper, Lorenzo-Lacruz et al. (2019) explained –in the sub-section 2.1 Catchment description– the following:

"The Ses Planes torrent has been affected by some severe flood events over the last half century: 12th October 1973, 3rd September 1982, 25th October 1985 and 6th September 1989 (Grimalt and Rodríguez-Perea, 1989). The latter was especially important; 156 mm of precipitation recorded in 2 h generated a flash flood that affected some areas of the town and led local authorities to undertake the artificial channelization of the Ses Planes torrent where it crosses the Sant Llorenç urban area".

The event of 1989 was general in the eastern part of Mallorca island, generating a flash-flood episode that affected most of the catchments in that part of the island, as explained in Grimalt and Rodríguez-Perea (1989). At the same time, only a few weather

stations reported a rainfall amount > 200 mm, and in general, the recorded rainfall was in the range of 100-200 mm.

References used in this comment

Cuevas-Tascon, G., Pascual, R., Roa, A., Esteban DA (2019). Catastrophic floods on October 9th 2018 in Mallorca Island. 3rd European Nowcasting Conference. Madrid (Spain). 24-26 April 2019

Llop Garau, J., Alomar Garau, G. (2012). Clasificación sinóptica automática de Jenkinson y Collison para los días de precipitación mayor o igual a 200mm en la isla de Mallorca. Territoris, 8: 143-152

Sumner, G., Ramis, C., Guijarro, J.A. (1995). Daily rainfall domains in Mallorca. Theoretical and Applied Climatology, 51:199-221

POINT 5 LORENZO-LACRUZ. The authors used the hydrological modelling GSM-SOCONT and it is also surprising, since this model is "a semi-distributed glacio-hydrologic model to simulate daily discharge from catchments with glacier cover" (https://www.mathworks.com/matlabcentral/fileexchange/43452-gsm-socont-glacio-hydrological-model). Surface runoff was computed with the SWMM (a conceptual glacio-hydrological model for high mountainous catchments). It seems not to be the most appropriate model for this watershed and for a catastrophic flash-flood analysis or reconstruction. There are other studies addressed in the Mediterranean region in which more appropriate models for flash flood episodes were used (Amengual, A., Homar, V. and Jaume, O., 2015: Potential of a probabilistic hydrometeorological forecasting approach for the 28 September 2012 extreme flash flood in Murcia, Spain. Atmospheric Research, 166, 10-23. Amengual and Carrio, 2017: A Comparison of Ensemble Strategies for Flash Flood Forecasting: The 12 October 2007 Case Study in Valencia, Spain https://doi.org/10.1175/JHM-D-16-0281.1). We are delighted with Dr. Lorenzo-Lacruz for his comment. We also recognize that referring to GSM-SOCONT in the manuscript can be confusing, as the snow and ice contributions

are null in this case and only the SOCONT part of the model was used. We will adapt the manuscript and refer to the SOCONT model only, as only the soil contribution of the model is determinant here.

However, we are concerned by the lack of knowledge and interest of Dr. Lorenzo-Lacruz regarding the model used in our study. In the link given by Lorenzo-Lacruz et al. (2019), it is possible to access the paper of Schaefli et al. (2005) where the SOCONT model is described in detail. We quote: "It is carried out through a conceptual reservoir-based model named SOCONT developed by Consuegra and Vez (1996) and similar to the GR-models (Edijatno and Michel, 1989)" and later in the same section: "Several applications of the SOCONT model to non-glaciered catchments (Consuegra et al., 1998; Guex et al., 2002) have shown that this model is able to reproduce all the major characteristics of the discharge such as floods, flow-duration-curves or the hydrological regime". As described in our manuscript, the SOCONT model takes into account infiltration, surface and sub-surface runoff, karstic behaviour, evapotranspiration and flow routing.

Moreover, the GR-models are widely used, in particular in France by national authorities to produce flood forecasts, including Mediterranean areas, see for instance https://webgr.inrae.fr/en/models/hydrological-forecasting-model-grp/. Hydrique Engineers successfully used SOCONT model for flood forecasting in many parts of the World, including flash flooding systems. More importantly, these models are applied in very different hydrological regimes (Mediterranean regions, karstic environments, snowmelt/glaciermelt fed hydrographs in Europe and China). We have therefore tested empirically that this kind of model is very versatile and is perfectly suited to analyse the kind of event described in the paper.

Regarding the other studies focused on the application of more appropriate hydrological models in Mediterranean catchments. Initially, we believe that it is not ethical to only cite those studies developed by some of the co-authors of the paper that you are leading because, secondly, a lot of models are applied and developed in Mediterranean

catchments.

Reference used in this comment

Schaefli, B., Hingray, B., Niggli, M., and Musy, A.: A conceptual glacio-hydrological model for high mountainous catchments. Hydrol. Earth Syst. Sci., 9: 95-109. DOI: 10.5194/hess-9-95-2005, 2005.

POINT 6 LORENZO-LACRUZ. Therefore, it is embarrassing to say that the "observed and modelled hydrograph and corresponding peak flow (442 cumecs)" showed in Figure 6 contains major flaws as: i) there is no observed hydrograph, ii) a glacio-hydrological model was used for a small Mediterranean watershed with low elevations and semiarid conditions, and iii) what is more surprising, the peak flow is reached one hour before the peak of rainfall. Is there any reasonable argument to defend such hydrological behaviour? We must again thank to Dr. Lorenzo-Lacruz about a warning on the observed hydrograph, despite the pejorative and abusive language. We address the three main concerns on Figure 6 from Estrany et al. MS, as follows:

i) there is no observed hydrograph. In Figure 6, the observed and the modelled hydrographs are perfectly performed in blue and red lines respectively in Estrany et al. MS (see Fig. 4 of this point-by-point document). In addition, as we have stated in the previous points, continuous streamflow monitoring and the application of a calibrated and validated hydrological model as the GSM-SOCONT represents an accurate and precise methodological approach to investigate the hydrological response of this catastrophic flash-flood event. For this reason, the use of adjectives like "embarrassing" is totally unprofessional and unjustified in science communication and in any context. However, we will modify the width of these lines to improve the performance of one of the main results of Estrany et al. MS.

ii) a glacio-hydrological model used for a small Mediterranean watershed with low elevations and semiarid conditions. As commented in the previous point 5, the SOCONT model is well suited to analyse the event. We believe that Dr. Lorenzo-Lacruz was

confused by the GSM part of the SOCONT model applied in our study.

iii) what is more surprising the peak flow is reached one or before the peak of rainfall. Is there any reasonable argument to defend such hydrological behaviour? We have revised the plotted hyetograph that represented the hourly average rainfall derived from all the radar points exposed in Fig. 6 of Estrany et al. MS. We have updated the hyetograph using only the rainfall radar points within the Begura de Saumà catchment. In the updated version of Fig. 6 of Estrany et al. MS, the peak of rainfall is at 19:00 h, 15 minutes earlier than the Q peak.

After this change, variables from Table 1 (rainfall and runoff variables of the flash flood) will be updated in the Estrany et al. MS, despite the fact that the differences do not modify the general patterns of the hydrological response of the catastrophic flash-flood. As a result, the time of maximum rainfall has been moved from 18.00 h to 19.00 h, whilst average total rainfall over the catchment derived from radar images has increased from 240 to 249 mm, IPmax average radar decreased from 46.4 to 45.3 mm h-1, IP average radar increased from 24.01 to 24.91 mm h-1 and runoff ratio decreased from 0.36 to 0.35. The updated rainfall and runoff values must be discussed in the same way as the previous values as changes were insignificant.

POINT 7 LORENZO-LACRUZ. It is at least strange that this study not even mention the work by Lorenzo-Lacruz et al. (2019) "Hydro-meteorological reconstruction and geomorphological impact assessment of the October, 2018 catastrophic flash flood at Sant Llorenç, Mallorca (Spain)", accepted for publication in October 22nd and published one month ago in NHESS. The study by Lorenzo-Lacruz et al. can be used for discussion about the different results that Estrany et al. obtained on rainfall data, peak discharge, hydraulic and hydrological modelling, geomorphological reconstruction, etc. This is, indeed, the objective of an open discussion journal. Explicitly speaking there is a resemblance of some of the content of this contribution (Estrany et al.) with the contents of Lorenzo-Lacruz et al. (2019). Not only the title and structure of the manuscript are similar, some figures included in this paper show a direct and evident inspiration

on the published paper by Lorenzo-Lacruz et al. (i.e.: figure 2 of the published paper and figure 2 of this manuscript; figure 6 and figure 7; figure 9.c of the published paper with figure 8.c of this manuscript). In science, the right of any particular to carry out research freely on any topic is respected, of course, but it is also necessary to cite and mention the work already done by others to avoid plagiarism.

We must again thank Dr. Lorenzo-Lacruz for this last comment –as we pointed out at the beginning of this document– because it represents a very good opportunity to better situate our research within the international scientific audience.

The short comment of Dr. Lorenzo-Lacruz is questioning (see the issues raised in this short comment from the abstract to the end of our manuscript) the validity of our data and therefore our ethical integrity as scientists. Sentences like "(. . .) there was no direct measurement during the flash flood (. . .)" demonstrate that accusations or insinuations of falsifying an investigation are extremely serious and damage up the confidence in our discipline itself, beyond our own reputation.

Furthermore, this barrage of denunciations –openly accusing us of plagiarizing Lorenzo-Lacruz et al. (2019)– is completely and demonstrably false, as we have argued throughout this point-by-point document. However, to shed further light on the evidence, we will provide a chronology of the case:

1. Last April 2019 we started the pre-submission of our manuscript on the NHESS Editorial Platform, detail that can be proven by the NHESS Editorial services, as the natural way for the significant working scientific effort carried out.

2. We were aware of the open discussion of Lorenzo-Lacruz et al. (2019) in the beginning of September 2019, when we had almost finished our manuscript, it was submitted on 13th September 2019. At this point, we had already detected the inconsistencies in the Lorenzo-Lacruz et al. (2019) manuscript. However, in view of the very transparent rules of NHESS –with an open discussion in parallel with a very rigorous peer-review process, we decided to wait until the second round of revision to cite this piece of work

in a "classic way"; i.e., until finishing the open discussion and anonymous reviews.

3. In addition, Lorenzo et al. (2019) was not definitely published by the date of us submitting our manuscript (13th September 2019), the status being "Open discussion". Thus, we decided to wait for both the definitive publication of Lorenzo-Lacruz et al's piece of work and the second round of revision of our MS. One of our main aims for citing them in this second round of revision was that Lorenzo-Lacruz et al. (2019) seemed to have based their piece of work on the fact that the Begura de Saumà River was ungauged (see Line 14 –Abstract–, and Line 86 –1. Introduction– of their open discussion MS), which is demonstrably false as we stated in our manuscript and within this point-by-point document.

4. Moreover, the supposed lack of knowledge of our streamflow data was at the very least surprising because the catastrophic nature of the event caused that MEDhyCON Research Group was required by the Emergency Authorities to directly participate in the post-catastrophe management by helping to rescue the last missing victim by applying a rapid mapping of sediment connectivity index and a UAV flight campaign for detecting detailed geomorphic changes. However, rapid mapping was enough to find this last missing victim. This approach is explained with detail in section 3.5 Sediment Connectivity and geomorphic change detection as emergency tools of Estrany et al. MS.

5. In November 2018, the Emergencies General Directorate of the Balearic Islands Autonomous Government required the MEDhyCON Research Group to provide an official report in order to perform a forensic reconstruction of this natural catastrophe. Within this report, MEDhyCON Research Group provided detailed information –although not calibrated– on the rainfall-runoff response during the flash-flood event. These key data were later used in the final report explaining the management of the catastrophic rainfall event that affected the Llevant County of Mallorca. This report was signed by the Operational and Technical Head of the Balearic Emergencies Resources and approved on 18th January 2019 by an agreement of the Balearic Council of Ministers (in Catalan)

(http://www.caib.es/govern/pidip/consells/dadesAcord.do?lang=ca&codi=9188230&extra=N). Specifically, page 7 of the second report (http://www.caib.es/pidip2front/jsp/adjunto?codi=2243620&idioma=ca) illustrates a detailed description of the rainfall-runoff process by using MEDhyCON data from the hydrometric station located in the Begura de Saumà River.

6. Lorenzo-Lacruz et al. (2019) must know these data and such statements are not helping the understanding of the event from the general public and cannot be considered plagiarism of their paper.

7. As we reiteratively stated along this point-by-point document, we are using our own data collected and carefully treated from the pressure probe installed and managed by the MEDhyCON Research Group in a hydrometric station since 2015 being combined with hydrological and hydraulic modelling by experienced scientists.

Regarding the mention of plagiarism on the structure of the paper, we must specify that Lorenzo-Lacruz et al. (2019) and Estrany et al. MS are analysing the same catastrophic flash-flood event in the same catchment. Therefore, the potential similarities between both papers can only be interpreted in terms of the study area. Identically, the NHESS applies a rigorous filter of plagiarism. Hence, the Editorial System did not detect any plagiarism with Lorenzo-Lacruz et al. (2019) in Estrany et al. MS.

In addition, Estrany et al. MS is focused on the following specific objectives (see 1. Introduction section, Lines 103-109): "The specific objectives of this study are to (1) explain the runoff response for elucidating the dependency of flood severity on catchment properties and human influences by using the flash flood Q data from a stream gauge installed in 2015 by the MEDhyCON Research Group; (2) assess the uncertainty of semi-distributed hydrological modelling in such severe flash flood in a karstic environment; (3) investigate the socioeconomic and territorial flood damages linked to hydrogeomorphological processes; and (4) multi-temporally analyse high resolution digital elevation models (HR-DEM) to detect and quantify geomorphic changes by using a UAV and a topography-based connectivity index for a rapid response of

post-catastrophe search and rescue tasks".

The specific objectives (methods) of Lorenzo-Lacruz et al. (2019) were stated in the Introduction 1, as follows: "The methodology used for the reconstruction of the event was organized on three main stages: (i) 10 min precipitation has been derived from radar reflectivity observations; (ii) two distinct fully distributed hydrological models have been implemented to accurately simulate the discharge hydrograph; and (iii) a hydraulic simulation has been performed to map the affected areas, including flooding extent and timing, water depth and flow velocity. Some of the geomorphological impacts on the main channel have been also assessed by using very high resolution orthophotographs and digital elevations models for comparison of pre- and post-flooding conditions".

Estrany et al. MS are providing a comprehensive picture of the event from using valuable continuous streamflow discharges very useful to help calibrate the application of hydrological models of extreme flash-floods in karstic environments as well as the application of geomorphic precision approach as a rapid management post-catastrophe civil protection tool. In conclusion, Lorenzo-Lacruz et al. (2019) and Estrany et al. MS are reasonably different perspectives of this catastrophic event.

Regarding the mention about plagiarism on specific figures, it is worthy of note that all these figures were already presented on the "International Seminar on flood risk planning and management in Mediterranean environments" organized by the Institute of Agroenvironment and Water Economy at the University of the Balearic Islands, in March 2019: http://diari.uib.cat/arxiu/Seminari-internacional-sobre-planificacio-i-gestio.cid579735. This seminar was specially organized to assess the Sant Llorenç des Cardassar catastrophe as well as generating a debate among the different public stakeholders aiming at improving the planning and prevention of these catastrophic events. The full Seminar can be viewed on the UIB youtube Channel: https://youtube.com/watch?v=mVjJfmUj0Lcwe.

We can also specifically point-by-point comment each figure that Dr. Lorenzo-Lacruz

is accusing us of plagiarising:

a) "figure 2 of the published paper and figure 2 of this manuscript". It is true that our Figure 2 has some coincidences with Figure 2 of Lorenzo-Lacruz et al. (2019). Specifically, a synoptic weather map of the day of the event and a satellite image were used in both Figures. It has to be considered that the usage of compositions like this are very common in papers focusing in flash-floods episodes. It was not the aim of the authors of the paper to copy any Figure or idea of Lorenzo-Lacruz et al. (2019). Our aim was to use a 'the facto standard' composition to facilitate the comprehension of the synoptic situation of that specific day.

b) "figure 6 and figure 7". Carefully comparing Figure 6 from Lorenzo-Lacruz et al. (2019), we are detecting that our Figure 7f is depicting as background the official flood risk maps and flood delimitation by Copernicus EMS at the Sant Llorenç des Cardassar village, but in the Figure 6a from Lorenzo-Lacruz et al. (2019) this information constitutes the basics of this Figure 6a. Accordingly, the purpose of both subfigures is completely different; i.e., the Fig. 7f from Estrany et al. MS illustrated the damage assessment, showing the location of affected buildings by the 9th October 2018 flash-flood compared with different levels of flood risk. In the case of the Fig. 6a from Lorenzo-Lacruz et al. (2019), it presented the checking of flooding extension of the 9th October 2018 flash-flood and selected return periods of flood risk.

c) "figure 9.c of the published paper with figure 8.c of this manuscript". As we widely stated through this point-by-point document, almost all data exposed in our manuscript were collected from our hydrometric station and during the very intense fieldwork campaign in the fortnight following the catastrophe, in which the MEDhyCON Team was collaborating with the Emergency Resources to help in finding the last missing victim. The aim of the Figure 8c from Estrany et al. MS was showing the overbank sedimentation estimated after the flash-flood by using our own data; i.e., a DEM performed with SfM from an imagery collected during a UAV flight campaign (15th October 2018) to find the last missing victim. It is also quite curious that the imagery and all data used

in Lorenzo-Lacruz et al. (2019) was ceded from a private company or public agencies (see the Acknowledgments of this paper), not being original from the authors.

**Elevations (m.a.s.l.)**

- ☐ 0 - 50   ☐ 350 - 700
- ☐ 50 - 200   ☐ 700 - 1,000
- ☐ 200 - 350   ☐ 1,000-1,445

- ● Hydrometric station equipped with water stage level and temperature, conductivity and turbidity
- ● Hydrometric station equipped with water stage level and temperature
- ▲ Rainfall stations

- —— Fluvial network
- ☐ Catchments

0   10   20 km

**Fig. 1.** Hydrometric network deployed by the MEDhyCON Research Team applying continuous streamflow monitoring.

[Figure]

[Figure]

(a)        (b)

**Fig. 2.** Hydrometric station of the Begura de Saumà River. (a) Installation of the digital equipment, 10th June 2015. (b) Data downloading the day after the flash-flood -10th October 2018-.

**Fig. 3.** Map of the north-eastern part of Mallorca Island, showing the catchments commented by Dr. Lorenzo-Lacruz with AEMET rainfall stations and MEDhyCON hydrometric stations.

[Figure]

**Fig. 4.** Comparison between the (a) Figure 6 of the Estrany et al. version in the Open Discussion stage and the (b) Reviewed Figure 6 that it will be submitted in the reviewed version of Estrany et al.

---

## Referee Comment (RC2) · Anonymous Referee #2 · 3 Feb 2020

Summary:

This manuscript analyses a flash flood event in a small catchment in the North-Eastern part of the Spanish Island of Mallorca, that left 13 people dead and caused severe damages to local properties. The analysis looks into four main aspects of the event, namely the meteorological conditions, the hydrological and hydraulic response, the damage assessment and a geomorphological analysis with the aim to improve the understanding of the drivers of this respective event. The authors conducted field measurements on the geomorphology few days after the event and present those findings alongside

measurements of the rainfall, discharge and a damage assessment of a severely hit village in the catchment based on ground-based records and remote sensing information. The authors use hydraulic and hydrologic models to model the runoff processes in the catchment. The presented data and results are discussed by topic and summarized in the conclusion.

General comments:

The paper is very interesting to read and provides important information on frequently underreported local flash flood events. The four aspects of the event are presented in great detail with very detailed information on the technical background of the data collection and modelling. However, overall the paper appears very fragmented with little connection between the different analysis. From reading the paper I was not able to fully understand how the presented data sets and models relate to each other and what are the main conclusions from the analysis. While the authors claim that their study uses an "[...] integrated approach with meteorological, hydrological, geomorphological, damage and risk data analysis" (L616f), the different analysis are presented largely isolated and independent including the discussion. Here, it would help if the authors would A) provide an overview figure that shows how the data sets and models are linked and B) A joint discussion that highlights how the individual results are linked and how this contributes to a better understanding of what made the event so devastating. It also appears that there is quite a disconnect between the results, discussion and conclusion sections, where topics such as driving factors of the damage in urban areas are for the first time explicitly mentioned in the conclusions, while the previous chapters mainly focus on the methodological aspects of the damage assessments. Similarly, language and grammar vary considerably throughout the paper and rigorous copy editing is necessary prior to accepting the manuscript for publication. Given the otherwise interesting and very relevant contribution the paper makes in the field of flash flood post event studies, I recommend considering the manuscript for publication after major revisions.

Specific comments

Structure

Introduction

The introduction is very technical and has a very narrow focus on flash flood processes. It also appears to address a lot of specific subjects in no particular order rather then leading to the research questions the authors are aiming to answer. Restructuring the introduction so it clearly leads to the research questions and highlights the importance of the work would therefore really improve the quality of the paper. As this is not the first study of its kind, I would also recommend including a literature review on previous post event studies (both flash flood related and potentially other natural hazards) and their findings. This would give the reader the opportunity to better evaluate the contribution of the paper to the scientific discourse and what knowledge gaps it addresses.

Description of the study area

For the sake of readability, I would recommend separating the meteorological conditions that lead to the event from the actual description of the study area.

Conclusion

The conclusion appears to be quite detached from the rest of the manuscript addressing several points that have not been previously mentioned in the manuscript but are important to fully understand the analysis. For example, how the meteorological, hydrological, geomorphological, damage and risk data analysis are linked. Or what the actual damage driving factor in urban areas are based on the different findings.

Rainfall

This paper focusses on the hydrological response as a main driver of the flash floods and the authors argue in the introduction that "the uncertainty in hydrological modelling can be large and hydrological models often need to be calibrated [. . .]. Therefore, the

predictability of such events remain low also adding that predictability is lowered by a high non-linearity in the hydrological response related to threshold effects". This implies that the uncertainty in the hydrological models are a key barrier in the predictability of flash floods. However, most other studies on flash floods and flash flood early warning systems find the spatio-temporal uncertainties in the rainfall prediction to be the largest obstacle in accurately forecasting and modelling flash floods (see for example Alfieri et al. 2017). This issue is also addressed in the description of the rainfall data, but the authors do not report to what extend the results of the subsequent hydrological and hydraulic models are sensitive to the uncertainties of the rainfall input. Therefore, I would recommend adding a short sensitivity analysis in regard to the rainfall input to the discussion section. It would also be interesting to see to what extend the results vary between the radar and gauge data.

Risk management and early warning

Given the high casualties and damage during this event it would be important to also cover the vulnerability of assets and people in the case study area for a comprehensive analysis of the damaging factors. This aspect however is only very briefly mentioned in the discussion and conclusion. Key questions would include: did people in the village receive some sort of early warning? Are their any risk management strategies in place apart from the mentioned flood zones? Discussing these aspects would also help to conclude with more specific recommendations for the improvement of risk management practises.

Damage classes

In Figure 7(e), the distribution of the damage classes for the three different zones and the total of all zones are shown. It seems that the total does not correspond to the sum of the three zones as the by far largest group in total are houses being "Damaged & Non habitable" with 260 houses, while the sum in this group for all three zones is 37 homes. That might be either an error or it should clearly be stated what is meant by

"Total".

Sediment connectivity and geomorphic change

While using the sediment connectivity to support search and rescue missions after flash flood events is a very innovative approach, it is not entirely clear what one can learn from the sediment volume calculation. Discussing this number in the context of the other analysis and its implications for a better understanding of the flash flood processes would help to further improve the manuscript. It would also be interesting to learn what is the accuracy of the mentioned approach given the different spatial resolutions and accuracies between the 2014 and 2018 surface models. Can changes in volume attributed to this specific event or does this number also include other changes to the geomorphology (both human and natural) that happened between 2014 and 2018? I would also recommend to clearly separate the sediment connectivity analysis that was used to support the search and rescue efforts and the geomorphic change detection to make clear that the two analysis had different aims.

Additional comments

As mentioned earlier, the manuscript would benefit from English language copy editing. Instead of giving point-by-point corrections I would like to provide a a few examples, which I find difficult to understand:

L 51f: "Characterising the response of a catchment during flash flood events is important because elucidate the hydrological processes from an extreme flood and their dependency on catchment properties and flood severity (Borga et al., 2007)" should probably be: "Characterising the response of a catchment during flash flood events is important because it helps understanding the hydrological processes of extreme floods and their dependency in regard to the properties of the catchment and the severity of the event."

L 112f: "[. . .] was developed affording the analysis of the rainfall-runoff processes at

small spatial scale during this extreme event." I did not understand what "affording" means in this context.

L125: "high-energy environment" I did not understand what "high-energy" means in this context

L156: "under a recurrent affection of wildfires": does that mean that these areas are regularly affected by wildfires or that these areas are prone to wildfires?

References

Alfieri, L., Berenguer, M., Knechtl, V., Liechti, K., Sempere-Torres, D., & Zappa, M. (2019). Flash flood forecasting based on rainfall thresholds. In Handbook of Hydrometeorological Ensemble Forecasting (pp. 1223-1260). Springer-Verlag.

---

## Author Response (AR1)

**Point-by-point reply to all referee comments (RCs) on "Hydrogeomorphological analysis and modelling for a comprehensive understanding of flash-flood damaging processes: The 9[th] October 2018 event in North-eastern Mallorca" by Joan Estrany et al.**

Joan Estrany[1,2*], Maurici Ruiz-Pérez[1,2,3], Raphael Mutzner[4], Josep Fortesa[1,2], Beatriz Nácher-Rodríguez[5], Miquel Tomàs-Burguera[6], Julián García-Comendador[1,2], Xavier Peña[4], Adolfo Calvo-Cases[7], Francisco J. Vallés-Morán[5]

[1]Mediterranean Ecogeomorphological and Hydrological Connectivity Research Team (http://medhycon.uib.cat), Department of Geography, University of the Balearic Islands, Carretera de Valldemossa Km 7.5, 07122 Palma, Balearic Islands, Spain

[2]Institute of Agro-Environmental and Water Economy Research –INAGEA, University of the Balearic Islands, Carretera de Valldemossa Km 7.5, 07122, Palma, Balearic Islands, Spain

[3]Service of GIS and Remote Sensing, University of the Balearic Islands, 07122 Palma, Balearic Islands, Spain

[4]Hydrique Engineers (http://www.hydrique.ch), Le Mont sur Lausanne, Vaud 1052, Switzerland

[5]Universitat Politècnica de València, Camí de Vera, s/n, València, 46022, Spain

[6]Estación Experimental de Aula Dei (EEAD-CSIC), Avenida Montañana, 1005, 50059 Zaragoza, Spain

[7]Inter-University Institute for Local Development (IIDL) Department of Geography, University of Valencia, Av. Blasco Ibáñez 28, 46010, Valencia, Spain

*Correspondence to*: Estrany J. (joan.estrany@uib.cat)

The response of the questions and comments provided by both anonymous RC1 and RC2 reviews will be addressed in a point by point format, being the comments from RC1 and RC2 in bold.

**RC1 - Anonymous Referee #1

**GENERAL COMMENT**. **The paper of Estrany et al. provides an analysis of the devastating flood that hit the North-eastern side of the Mallorca Island in October 2018, considering: 1) the hydrological response of the catchment; 2) damage assessment; and 3) geomorphic changes. The analysis presented is quite detailed, and represents a very good starting point that, linked to a study from a meteorological perspective and to a hydraulic study about the flooding dynamics (two aspects that –I acknowledge- go beyond the scopes of the paper), would provide a rather comprehensive picture of the event from a civil protection perspective. The data provided by the water level station are particularly interesting and valuable.**

We are truly delighted with the comments of Anonymous Referee #1, due to is considering we are providing a valuable insight in improving the comprehension of extreme flash-flood events in Mediterranean environments, highly subject to human pressures.

Regarding meteorological perspective and hydraulic study, we are also aware that they are both out of scope of this first approach carried out within this manuscript. However, the authors we are working to go beyond this first study, precisely (1) addressing a detailed 2D hydraulic modelling in which urban flooding dynamics are being investigated; and (2) evaluating geomorphic changes in two headwaters small catchments through high-resolution digital elevation models built from images captured by a UAV.

**My major comment is that the paper has the potential to go beyond the 'simple' description of a case study, where three single pieces (rainfall-runoff modelling, damage assessment and geomorphic changes) are discussed separately. Discussion section could help to bridge this gap. It introduces several topics (e.g., land-use changes, fires, etc.), which, however, are treated in an increasingly qualitative and general way during the discussion itself. What are their actual**

**(and relative) effects on this event? Can they be quantified? Also, the triggering effect of the karstic reservoir(s) should be somehow addressed with more detail (I mean, the authors should try to go beyond the conceptual modelling and provide insights about the physical process, which involves, e.g., specific geological features in specific areas). I wonder about the sudden increase in discharge from 120 to 442 cms (an impressive peak flow rate per unit area) in 15 minutes (very fast response time). The reason for this behaviour is not totally clear. Is it mainly due to the karstic environment**

**or to other reasons (e.g., the failure of a temporary dam)?**

It is a very challenging question to assess the effect of land-use change on the event and we believe this is out of scope of the current paper, as it would probably deserve a publication on its own. Our goal in the discussion was to highlight possible further investigation on this aspect.

Concerning the triggering effect of the karstic reservoir, we acknowledge that this is a main part of uncertainty in the model.

As modeled, the release of the karstic reservoir is very similar to the failure of a temporary dam, as in the karstic model, the water is stored in a karstic reservoir, which is then suddenly released by syphon effect. There is no evidence that it is solely due to karstic behavior. We mentioned this point through the Discussion section and will be showed in the following comments. We have deeply modified the Discussion section. Firstly, the two subsections have been unified forming a unique storyline where hydrological response, rainfall-runoff modelling, damage assessment and geomorphic change are integrated. We believe this new version of the MS is bridging the gap between them, because we have introduced several paragraphs especially focused on the predictability of this kind of flash-flood events in order to better join these different issues. In addition, the new Figure 1 is also useful to better understand the integrated approach.

This is the paragraph written to better join the different parts of the Discussion section (see Lines 636-646 of the revised MS):

"*At present, Mallorca does not have any sort of early warning system to assist flood risk management, and nor of course has*

*Sant Llorenç des Cardassar. Similarly, no hydrometeorological early warning was issued by the competent authorities, as the Balearic Islands have no operational hydrological control network releasing real-time information on discharges. In October 2018, Sant Llorenç des Cardassar was one of the four municipalities in Mallorca with a flood risk emergency plan. However, it was not operational at the time the emergency was declared. As a result, the population was completely unaware of how to defend themselves, even during the emergency phase, although Sant Llorenç des Cardassar municipality had significant social*

*vulnerability to floods, as most of the casualties were tourists and the elderly*".

Secondly, we have addressed a qualitative –but also quantitative– discussion about the role played by rainfall intensity and its spatial distribution, complex geology and land cover disturbances, following the suggestion provided by the Anonymous Referee 1#. For this purpose, we have also modified the subsection 2.1. Study area (see Lines 155-165 of the revised MS), where a deeper assessment of permeability in lithology materials as well as a diachronic evaluation of land uses evolution and perturbations (i.e., wildfires) is sustaining the discussion on the role of physical parameters generating the flash flood.

"*The lithology is mainly composed of marls intercalated with limestone (60% of the area) of the Medium-Upper Jurassic (Dogger), dolomites (22% of the area) of the Upper Triassic and Lower Jurassic, and pelagic limestone marls (14% of the area) of the Lower Cretaceous (Fig. 2d). This lithological composition determines the surface water/groundwater interaction. On the one hand, a high degree of fracturing, fissuring and karstification of limestone favours percolation through karstic*

*aquifers. On the other hand, the imperviousness of Dogger and Cretaceous marls (74% of the area) does not allow percolation, enabling runoff generation. The main land use in 2012 was agriculture (58%), mostly located in lowland areas. Forest (26%) and scrubland (17%) were predominant at headwaters. Terraced fields still occupied 10% of the catchment, although most of them were abandoned (Fig. 2e). In 1956, natural vegetation covered 21% of the catchment. This rose to 42% in 2012 due to an afforestation process of former agricultural land in the second half of the twentieth century. In combination with other*

*factors, afforestation triggered a higher fire risk: two wildfires burnt an area of 1.7 km$^2$: 17% in 1983 and 83% in 2011 (Balearic Forestry and Soil Conservation Service, [http://xarxaforestal.caib.es](http://xarxaforestal.caib.es); Fig. 2e)*".

We have also placed special emphasis on the sudden increase in discharge from 120 to 442 m$^3$ s$^{-1}$, which has resulted from the combination of all these physical parameters. Please, see the Lines 583-607 of the revised MS:

"*This runoff response resulted from the combination of rainfall intensity and its spatial distribution, complex geology and land*

*cover disturbances in generating a high $Q_{peak}$ (i.e., 442 m$^3$ s$^{-1}$) with high potential for generating geomorphological changes. Thus, the $Q_{peak}$ unit obtained (i.e., 19 m$^3$ s$^{-1}$ km$^2$) can be classified as the third highest value of all the reported values in Marchi et al. (2010) and the highest of those values obtained from streamflow measurements in a hydrometric station and not by post-event analysis. The hydrologic response analysis in the course of a flash flood shows how storm structure and evolution result in a scale-dependent flood response (Borga et al., 2007). Consequently, spatial rainfall organisation, geology combined with*

*orography and land cover disturbances led to pronounced contrasts in the flood response at the Begura de Salma River. Spatial rainfall on the catchment scale showed that the highest accumulation at the beginning of the storm was located at the*

*headwaters of the catchment (at 15:00 h), whilst during the last part of the event the most important rainfall amounts were located in the downstream part. Examination of the flood response illustrated how the extent and the position of the karst terrain (Zanon et al., 2010) and soil conservation practices (Calsamiglia et al., 2018; Tarolli et al., 2014) provided major*

*geological and anthropogenic control of runoff response. Impervious materials cover 74% of the Begura de Salma River catchment, mostly located at the headwaters, which are responsible for the highest values of topographic torrentiality (Estrany and Grimalt, 2014b), facilitating rapid overland flow generation. During the first part of the storm, when the highest rainfall amounts affected the headwaters, runoff response was delayed by the laminar effect of check-dam terraces massively constructed over Cretaceous marls (Calvo-Cases et al., 2020) and by the predominance of percolation in those areas covered*

*by limestone, mostly in the intermediate parts of the catchment. During the last part of the event, when the highest rainfall intensities were in the downstream part, the excess of soil infiltration capacity and the collapse of headwater check-dam structures triggered the sudden increase in discharge from 120 to 442 $m^3$ $s^{-1}$ in only15 minutes at the hydrometric station. Moreover, the increase of 5 $km^2$ (21% of the catchment area, see more details in section 2.1) of natural vegetation since the 1960s as a result of afforestation processes, increased fuel loads and the risk of wildfires led to 1.7 $km^2$ (7% of the catchment)*

*being burnt since 1980. The removal of vegetation by fires has a similar effect (less interception, less soil storage), which has been experimentally documented after major fires. These factors are a major reason why the history of the steady devastation of plant cover in the Mediterranean is likely to enhance flood risk (Wainwright and Thornes, 2004) and increase desertification tendencies*".

**Another important point is that authors should take care of the English language and grammar. At the end of the review, I provide some examples, limited to the Abstract and the Introduction, but a thorough review should be carried out throughout the paper.**

The authors we are grateful with detailed suggestions on English language and grammar. Following this advice, we have requested an external review on English language by a professional native speaker (see attached the certificate).

**Finally, please find below a list of other specific comments. I hope that my comments help to improve the quality of the paper.**

We thank to the reviewer for her/his dedication on providing accurate specific comments, which have all been carefully addressed.

**Abstract: it could be much more concise, avoiding unnecessary comments (e.g., "comprehensive analyses of catastrophic events are crucial…"). It is of the foremost importance that the abstract is as much straightforward as possible**

We think that the abstract is explaining in a concise way the different issues. However, we have deleted these unnecessary comments.

**L24: maybe remote sensing is better**

It has been changed.

**L31: Copernicus EMS: it's better to avoid acronyms in the abstract without explanation**

The acronym has been removed by "Emergency Management Service". See Line 30 of the modified MS.

**L45: also the interaction with the (warming) sea surface is an extremely important and peculiar feature of the Mediterranean area (e.g., Cassola et al., 2016; Avolio et al., 2019)**

We have modified this sentence in order to add the "warm of sea surface" as a driven factor, as well one these references. See Lines 42-44 of the modified MS:

"*However, catastrophic flash floods are much more frequent in some parts of the Mediterranean region than in the rest of Europe due to the interaction between geomorphology, climate, vegetation and the warm sea surface (Cassola et al., 2016), all combining to create a flood-prone environment*".

The new reference:

Cassola, F., Ferrari, F., Mazzino, A. and Miglietta, M. M.: The role of the sea on the flash floods events over Liguria (northwestern Italy), Geophys. Res. Lett., 43(7), 3534–3542, doi:10.1002/2016GL068265, 2016.

**L79: ok, but the main uncertainty in predictability is linked not only to hydrological uncertainty but also (mostly, I would say) to the meteorological uncertainty. This aspect should be also introduced.**

As the reviewer has also recognized in some comments focused on the Discussion section, it is not the main aim of our study. Despite this, we addressed meteorological uncertainty in the Discussion section. However, following the own recommendations of the reviewer, we have reduced the meteorological uncertainty in the Discussion section and added the issue within the Introduction section with the following sentence in Lines 67-69 of the modified version:

"*The main source of uncertainty is related to the spatio-temporal scales of rainfall pattern. The forecasting of intense thunderstorms by numerical weather prediction systems to provide accurate rainfall information is particularly challenging (Alfieri et al., 2015; Collier, 2007)*".

References used in this comment:

Alfieri, L., Berenguer, M., Knechtl, V., Liechti, K., Sempere-Torres, D. and Zappa, M.: Flash Flood Forecasting Based on Rainfall Thresholds, in Handbook of Hydrometeorological Ensemble Forecasting, edited by Duan Q., F. Pappenberger, A.

Collier, C. G.: Flash flood forecasting: What are the limits of predictability?, Q. J. R. Meteorol. Soc., 133(622), 3–23, doi:10.1002/qj.29, 2007.

**L109: since the structure of the paper is complex, a brief introduction to the next Sections could be useful**

We believe that the specific objectives deployed in the last part of section 1. Introduction are providing the conceptual structure of the paper that is performed by the typical structure of sections in a scientific paper such as Materials and methods, Results and Discussion. In addition, t the first paragraph of the section 2. Materials and Methods is really useful to provide a comprehension of the paper structure. It is true that the explanation provided in this first paragraph is in terms of methods, but they are completely related with the structure of the paper. However, to provide more consistence to the structure, we have completed this first paragraph of the section 2. Materials and methods, as follows (Lines 125-127 of the new MS version):

"*Finally, high-resolution digital elevation models (HR-DEM) were generated by LiDAR 2014 data from the Spanish National Geographic Institute and by imagery captured through a low-cost UAV just six days after the catastrophe to calculate a sediment connectivity index (IC) and measure geomorphic changes (Fig. 1)*".

In addition, we have designed and performed a new figure with a workflow of the different steps and their relation to the objectives of the manuscript. This is the new Figure 1, captioned as "Methodological workflow of the research study".

**L136: please explain what you exactly mean with "torrentiality". This index could be ignored by most of the audience**

We have added some words for provide a clear explanation to most of the audience, in Lines 148-149 of the revised MS, as follows:

"*...which is topographically computed as a coefficient between the number of first-order streams and catchment area, multiplied by the drainage density; cf. Strahler, 1964)...*".

**L193: a reference is needed to justify the sentence**

We added two references to justify the sentence, Lines 208-212 of the revised MS:

"*Two pair of coefficients were tested: (i) the pair a=200 and b=1.6 was tested because AEMET commonly uses these coefficients to obtain near-real time rainfall estimations ([http://www.aemet.es/es/eltiempo/observacion/radar/ayuda](http://www.aemet.es/es/eltiempo/observacion/radar/ayuda); last access: 15 May 2020) from the same radar data that we used in this research; (ii) the pair a=300 and b=1.4 was tested because the NWS in the USA uses it at operational level (Fulton et al., 1998) and it is argued that these coefficients perform*

*better in a convective environment than the first ones (e.g. Seo et al., 2020)*".

References used in this comment

Fulton, R.A., Breidenbach, J.P., Seo, D.-J., Miller, D.A., O'Bannon, T.: The WSR-88D rainfall algorithm. Weather Forecast, 13, 377–395, 1998.

Seo, B-C, Krajewski, W.F., Qi, Y.: Utility of Vertically Integrated Liquid Water Content for Radar-Rainfall Estimation: Quality     Control     and     Precipitation     Type     Classification.     Atmospheric     Research,     236,     104800, doi:10.1016/j.atmosres.2019.104800, 2020.

**L200: Please add in a figure (Fig. 2?) the radar location**

The location of the radar has been added in Figure 2b, formerly Figure 1b.

**L201: "due to these effects". What effects? Not clear. Please explain and justify with adequate reference(s)**

We agree with the reviewer that the sentence was not enough clear. We have restructured the sentence as follows (see Lines 218-223 of the reviewed MS):

"*Mountains may partially or totally block the electromagnetic radar signal and affect radar reflectivity and precipitation estimations(Germann and Joss, 2004). The study area is mountainous, but with low maximum altitudes (~400 m.a.s.l.). This low elevation combined with the regional orography, the distance of the Begura de Salma River from the radar (~50 km), the*

*0.5º azimuth of the PPI used, and the altitude of the radar location (113 m.a.s.l.) avoided any topographic interference with the radar signal. Thus, no orographic blocking reflectivity correction technique was needed*".

References used in this comment

Germann, U. and Joss, J. (2004) Operational measurements of precipitation in mountainous terrain. In: Weather Radar:

Principles and Advanced Applications. Meischner, Peter. (Ed). Springer, Berlin, pp 52-77

**LL206-213: this paragraph is not clear. What were the driving data for this analysis? Those provided by the rain gauges surrounding the catchments? If so, the authors should provide some proofs about the reliability of their analysis (e.g., a scatterplot, even in the Supplement). Furthermore, if rainfall data are gridded in 2x2 square cells, why in fig. 6 it looks like they are spatially interpolated? Maybe, because they are related to the GSM-SOCONT scheme (for which the authors refer to inverse distance weighting)?**

We have rewrote the paragraph in order to clarify this issue. We have also generated a scatterplot in which the two estimations are compared against the observed rainfall (the new Figure S1).

The new paragraph in Lines 227-234 of the revised MS:

"*With the set of coefficients a=300 and b=1.4, the maximum amount of estimated rainfall using radar data clearly underestimated the observed rainfall, with a PBIAS of –50.6% and an estimation of ca. 149 mm as the maximum rainfall amount, compared with the 257 mm recorded at the Sant Llorenç des Cardassar rain gauging station (see Fig. 2c). Instead of using the recorded rainfall in gauging stations to calibrate the radar-based rainfall, a correction method of the rainfall estimation based on spatial resampling was posited here. Accordingly, the 2\*2 km spatial resolution of radar data was resampled by assigning to each grid cell the value of the maximum amount of estimated rainfall at 1\*1 km. By this method, the regression coefficient reached $R^2$= 0.8, a PBIAS of only +2.6% and 258 mm as the maximum estimated rainfall amount, which fitted the rainfall observed at that point (Fig. S1)*".

The estimated rainfall from gridded radar data was directly used in the SOCONT scheme without inverse distance weighting. Only the temperature is interpolated. This has been clarified as follows (see lines 365-367 of the revised MS):

"*Resampled 1 km resolution radar data (see subsection 2.3) were used in the model to obtain precipitation for each elevation band by including all 1 km resolution points falling within each elevation band*".

**L218: When introduced, the acronym MEDhyCON (such as any other) should be explained**

We have introduced the whole meaning of the acronym MEDhyCON within the Introduction section: MEDiterranean hydrological CONnectivity Research Group.

**L251: "a second hydrograph was designed": with what Q values?**

A second hydrograph was designed, ranging also between 2 to 512 m$^3$s$^{-1}$, with additional flow steps. The first hydrograph consisted of nine different flow values, whereas the second one consisted of nineteen different values or steps, including intermediate values, and some other significant discharge values, such as the maximum channel capacity, and the value at which the presence of the bridges start influencing the hydraulics of the system. All values are represented as points in Figure

5 SDRC (formerly Figure 4). Besides, we have modified the sentence explaining the second hydrograph design. In the former version of the MS (Lines 250-253) was:

"*Under these conditions and with this designed hydrograph, a first approach to the SDRC was obtained. In order to improve the accuracy of the SDRC, a second hydrograph was designed, also containing the Q values of the first hydrograph*".

The new version of the sentence (see Lines 270-273 of the revised MS) is:

"*These conditions and this designed hydrograph gave a first approach to the SDRC. To improve the accuracy of the SDRC, a second hydrograph was designed with nineteen Q values, also containing the previous nine Q values of the first hydrograph, new intermediate values and some other significant Q values such as the maximum channel capacity and the Q value at which bridges influence the hydraulics*".

**L262: I would like some more details about the percentages of obstruction detected and how they were calibrated. Do they rely (only) on the three ground control points? Why those points? When/how were the water levels measured on them? Does the simulation consider 0% obstruction up to below the bankfull and 85% immediately after? No transient state?**

The percentages of obstruction detected were estimated from post-event photographs and calibrated using maximum water stages observed in situ 10 hours after the event within a period of time in which the high-water marks were still preserved. These marks –fully representatives of the overbank flows– were mapped by using a dGPS Leica 1200 through these three ground control points with high precision. Therefore, intermediate data is not available to compare results in the simulation of the transient state, and maximum water stage must to be compared with maximum obstruction results. Furthermore, the obstruction mechanism was very fast causing that we hydraulically simulated that bridges' obstruction does not occur when water depths are even close to the low chord of the bridge's deck. However, when the firsts floating elements start clogging the bridge opening, the obstruction rapidly increases. In addition, this obstruction was maintained throughout the flash-flood event duration, as it could be checked observing post-event pictures.

Nevertheless, for providing more clarity about the influence on hydraulics of bridges' obstruction, see the answer to the previous comment and its related modification within the MS. In addition, we have added the following sentence (see Lines 283-284 of the revised version of the MS) previous to the sentence "*85% at Bridge 1, 40% at Bridge 2 and with no obstruction for the other ones*":

"*These post-event pictures and maximum water stages observed in situ 10 h after the event was useful to estimate the obstruction percentages of these two bridges*".

In addition, this sentence "*Accordingly, ground control points in three representative locations around the hydrometric station were selected (Fig. 3b), also considering the maximum WS reached in the hydrometric station (4.55 m)*" (Lines 337-341 of the former version of the MS) was also modified to better explain the support of the three selected ground control points (see Lines 290-293 of the new version of the MS):

"*Maximum WS observed in situ 10 hours after the event, within a period of time in which the high-water marks were still* preserved, were mapped through ground control points. Three of them were selected as representative locations around the hydrometric station (Fig. 4b), and the maximum WS reached at the hydrometric station (4.55 m) was also included*".

**L321: in my opinion, the best tool for this kind of assessment would have been a complete 2D hydraulic study. Please discuss briefly your choice and its advantages (e.g., it's time-saving, etc.)**

We agree with this observation, but flow direction here used were useful to firstly assess the role of hydraulics in damages. Furthermore, a detailed and complete 2D hydraulic modelling in which urban flooding dynamics are being investigated. All the buildings and urban elements of Sant Llorenç des Cardassar village will be introduced in the 2D model by using a high-resolution digital elevation model established from LiDAR technology two months after the catastrophe. As we pointed previously out, this complete 2D hydraulic study is out of scope of this first study; although the results of flow direction here developed will be compared to the 2D hydraulic study.

The sentence of Line 321 of the former MS has been modified to reinforce this argument (currently in Lines 344-346 of the revised version of the MS):

"*Second, the flow direction in the urban network was calculated with Arc Hydro Tools (ESRI, 2019). This gave a preliminary* assessment of the role of hydraulic processes in physical damage. Due to the flow direction, this is mainly related to the velocity vector component perpendicular to the building element surface (Amirebrahimi et al., 2016)*".

Reference used in this comment:

Amirebrahimi, S., Rajabifard, A., Mendis, P. and Ngo, T.: A framework for a microscale flood damage assessment and visualization for a building using BIM–GIS integration, Int. J. Digit. Earth, 9(4), 363–386, doi:10.1080/17538947.2015.1034201, 2016.

**Table 1 needs more explanation. Terms like IPmax should be explained**

This comment is rising that NHESS audience is beyond the catchment hydrology expertise. Accordingly, we have added a detailed description of each parameter at the beginning of subsection 3.2 Hydrological response of the flash flood, as follows (Lines 428-439 of the revised version of the MS):

"*The hydrological response of the flash flood was analysed through variables derived from the rainfall (Table 1a, 7 variables)* and runoff (Table 1b, 9 variables) of the catchment: Event rainfall duration: duration from the beginning of rainfall until

*stopped it; Time of maximum rainfall: time of the highest rainfall intensity; Centroid storm: central time of the rainfall event;*

*Average radar rainfall: mean rainfall obtained by radar; $IP_{max}$ average radar: average of the highest rainfall intensities*

*obtained from radar rainfall points; $IP_{max}$ radar: highest rainfall intensity obtained from radar data; IP average radar:*

*average of rainfall intensities obtained from radar rainfall points; Runoff: discharge volume amount divided by the catchment*

*area; Runoff ratio: ratio between runoff and rainfall, also known as runoff coefficient when is expressed in percentage; Event*

*duration: duration of the flood event; $Q_{max}$: peak discharge; Time $Q_{max}$: time of the peak discharge; T centroid storm – T $Q_{max}$:*

*duration between the time of the rainfall centroid and the time of the discharge peak; $Q_{average}$: discharge average during the*

*flood event; Unit peak discharge: peak discharge divided by catchment area, allowing the comparison of peak discharge*

*independently from catchment size; Reduced Unit peak discharge: discharge peak divided by catchment area in square*

*kilometres elevated by 0.6. The exponent was obtained from Gaume et al. (2009), who applied this parameter to compare*

*reduced unit peak discharge from different flash-flood events*".

**Figure 7 is not very clear. Maybe it could be divided into more figures. However: in Fig. 7a, are the red dashed polygons**
**all derived from the Copernicus EMS? Also zone 2 and 3? Do the latter perfectly correspond to the Government survey?**
**Figs. 7b and 7c are not very readable/useful, in my opinion.**

The Figure 7 is the Figure 8 in the revised MS. Once clarified this structure detail, we must recognize that the legend in Fig.
7a was not comprehensible due to "Affected zones" did not help to observe what are the source for determining the affected
zones; i.e., Copernicus EMS or Government survey. As a result, we have modified the sub-figures "a", "b", and "c". With this
modification, we believe that sub-figures "b" and "c" are totally useful due to are areas not detected by Copernicus EMS and
we explain throughout the main text their damage level supported by these sub-figures.

**LL510-511: I guess it is Fig. 8c.**

Exactly, many thanks for the accurate detection of this error. It has been changed, in the reviewed version of the MS is Fig.
9c.

**LL535-536: "Despite these antecedent…as reported by these authors." Why?**

We have modified this sentence and the following one in order to improve the explanation. The former version (Lines 535-
538) was:

"*Despite these antecedent wetness conditions, the runoff coefficient of the event (i.e., 36%) was analogous to the median runoff*
*coefficient under average wetness conditions (37%) than dry ones (20%), as reported by these authors. This response*

*illustrated the key role of rainfall intensity in the generation of a high $Q_{peak}$ (i.e., 442 $m^3$ $s^{-1}$) with a high potential to generate geomorphological changes*".

The new version (Lines 581-585 of the revised MS) is:

"*Despite these dry prior conditions, the runoff coefficient of the event (i.e., 35%) was analogous to the median runoff coefficient under average wetness conditions (37%) reported by Marchi et al. (2010), rather than dry ones (20%). This runoff response resulted from the combination of rainfall intensity and its spatial distribution, complex geology and land cover disturbances in generating a high $Q_{peak}$ (i.e., 442 $m^3$ $s^{-1}$) with high potential for generating geomorphological changes*".

**LL547-548: it's not clear why the authors need to adjust the initial conditions manually. Does the model not perform well if used for long periods?**

We thank the reviewer for his/her critical reading. Actually, the model is not able to reproduce the event when it is running for
a longer period. Therefore, the model was run starting in 2015 until the flash-flood event with meteorological data. When using the initial conditions from the long-term run and the radar data, the model is not able to reproduce the event. The extraordinariness of the flash-flood event and the few flood events recorded since 2015 by the hydrometric station did not allow to calibrate the model for the event. In other words, by using specifically calibrated initial conditions, the model is more an event-based model rather than a classical hydrological model. This issue is thoroughly mentioned in the Discussion section,
and we have added a sentence reinforcing it (Lines 614-616 of the revised MS):

"*In this context, the initial condition $H(t_0)$ was manually adjusted, as numerical models applied to simulate catchment runoff response are often unsuccessfully implemented for Mediterranean-climate catchments due to their very heterogeneous responses over time and space (Merheb et al., 2016)*".

References used in this comment

Merheb, M., Moussa, R., Abdallah, C., Colin, F., Perrin, C. and Baghdadi, N.: Hydrological response characteristics of Mediterranean catchments at different time scales: a meta-analysis, Hydrol. Sci. J., 61(14), 2520–2539, doi:10.1080/02626667.2016.1140174, 2016.

**LL556-585: the discussion introduces many arguments in a general and qualitative way. I suggest to skip/reduce much of the discussion (especially that about the weather predictability, which is not addressed in this study) and/or try to quantify the different effects (please refer to my main concern).**

We are very grateful for this key comment, because it is a great opportunity for going beyond the simple description and join
the different issues addressed within this manuscript. Accordingly, we have further and deeply assessed the physical parameters conditioning the hydrological response of the Begura de Salma River catchment; i.e., historical and current land uses / land cover, soil conservation structures, the affection of wildfires and perviousness of lithology throughout the catchment.

**English language and grammar review:**

**L25: at the catchment scale**

"the" has been added in the reviewed MS.

**L26: peak discharge of 442…**

"of" has been added in the reviewed MS.

**L28: "i.e." not needed**

We have deleted "i.e." before the catchment surface area.

**L38: For that reason, they usually affect/impact basins…**

The beginning of this sentence has been modified following your advice. In the reviewed version of the MS: "For that reason, they usually impact basins…"

**L46: very close to the coastline**

We have changed "closeness" by "close".

**L48: "scarce soils". What do the authors exactly mean?**

We have deleted the sentence "Another cause is the reduced vegetation cover and scarce soils" to avoiding confusion also considering that this part of the Discussion has been completely rewritten. Besides, the following sentence of the same paragraph was explaining this process in a better way (Lines 604-607 of the revised MS): "*The removal of vegetation by fires*

*has a similar effect (less interception, less soil storage), which has been experimentally documented after major fires. These factors are a major reason why the history of the steady devastation of plant cover in the Mediterranean is likely to enhance flood risk (Wainwright and Thornes, 2004) and increase desertification tendencies*".

**L51: elucidates**

The "s" has been added to "elucidate".

**L51: "the hydrological processes from an extreme flood"? Do you mean the hydrological processes activating during**
**an extreme flood or so?**

This was the former version of the sentence:

"*Characterising the response of a catchment during flash flood events is important because elucidate the hydrological processes from an extreme flood and their dependency on catchment properties and flood severity (Borga et al., 2007).*"

And this is the new version of the sentence (Lines 48-50 of the reviewed MS):

"*Characterising the response of a catchment during an extreme flash-flood is important because elucidates the hydrological activating processes and their dependency on catchment properties and flood severity (Borga et al., 2007*".

**L54: small spatial scale usually means low-resolution (e.g., scale 1:50000 is a smaller scale than 1:500). Please rephrase to make the sentence clearer**

In order to avoid confusion, we have changed "small" by "limited": see Line 54 of the revised MS.

**LL67-68: in order to reduce the uncertainty of the Q estimate, I suppose**

We have added "the uncertainty of".

**L69: "…also adding that…" please rephrase**
We have deleted "…also adding that predictability is lowered…", being changed by "…conditioned by…". The new version of the sentence is as follows (Lines 73-74 of the revised version of MS):

"*Flash-flood events are also conditioned by high non-linearity in the hydrological response relating to threshold effects (Braud et al., 2014). Therefore, the predictability of such events remains low*".

**L87: regarding vulnerability**

We have deleted the beginning of the sentence "Particularly regarding to vulnerability", because it was also redundant.

**L88: "understanding…are developed". Understanding is not developed. Please rephrase**

We have changed "understanding" by "assessment of".

**L92: evaluates**

It has been changed.

**L97: "with that Copernicus EMS one". Not clear**

We have changed the sentence, being now "*In addition, a comparison of 'ground-based' assessment and 'remote-based'*
*Copernicus EMS may shed light on the accuracy of this rapid and helpful tool for assessing most catastrophic flash floods*".
Lines 103-105 revised MS.

**L112: why "precipitation as well as the Q"? I would write either "Precipitation and discharge" or "P and Q"**

We have changed by "Precipitation and discharge".

**Captions: please check also all figure captions (e.g., "left" and "right" pictures in Fig. 3, in Figure 7 there are two references to (f))**

In the former Figure 3 (Figure 4 in the revised MS), "left" and "right" was wrongly indicating the dates of the pictures. We have changed by "d" and "e" letters for more internal coherence of the figure and avoiding confusion.
In the former Figure 7 (Figure 8 in the revised MS), we have deleted "(f) Damage level of the buildings and plots by zones".

**Supplement: please check grammar also here (e.g., L'independent; "Schema"). The caption of Figure S2 should declare the meaning of the variables.**
The grammar has been reviewed: "L'independant" and "Scheme".
We have also added the meaning of the variables in the caption of the Figure S2.

 **RC2 - Anonymous Referee #2

**SUMMARY:**

**This manuscript analyses a flash flood event in a small catchment in the North-Eastern part of the Spanish Island of Mallorca, that left 13 people dead and caused severe damages to local properties. The analysis looks into four main aspects of the event, namely the meteorological conditions, the hydrological and hydraulic response, the damage assessment and a geomorphological analysis with the aim to improve the understanding of the drivers of this respective event. The authors conducted field measurements on the geomorphology few days after the event and present those findings alongside measurements of the rainfall, discharge and a damage assessment of a severely hit village in the catchment based on ground-based records and remote sensing information. The authors use hydraulic and hydrologic models to model the runoff processes in the catchment. The presented data and results are discussed by topic and summarized in the conclusion.**

We are very grateful by the accurate summary provided by the reviewer.

**GENERAL COMMENTS:**

**The paper is very interesting to read and provides important information on frequently underreported local flash flood events. The four aspects of the event are presented in great detail with very detailed information on the technical background of the data collection and modelling.**

The authors kindly appreciate the comments from Reviewer#2, which have helped to improve the MS.

**However, overall the paper appears very fragmented with little connection between the different analysis. From reading the paper I was not able to fully understand how the presented data sets and models relate to each other and what are the main conclusions from the analysis. While the authors claim that their study uses an "[...] integrated approach with meteorological, hydrological, geomorphological, damage and risk data analysis" (L616f), the different analysis are presented largely isolated and independent including the discussion. Here, it would help if the authors would A) provide an overview figure that shows how the data sets and models are linked and B) A joint discussion that highlights how the individual results are linked and how this contributes to a better understanding of what made the event so devastating.**

We agree with the Anonymous Referee #2 comments, which are –at the same time– fitted with some of the concerns highlighted by the Anonymous Referee #1. We are also very pleased to the referee for providing constructive ideas that –we believe– have improved the new version of the MS.

Accordingly, also applying a suggestion provided by the Anonymous Referee #1, we have created an overview figure showing the links between data and models. It is the new Figure 1.

The Discussion section has also been completely restructured without subsections to facilitate the combination of the individual results contributing to an integrated comprehension of the devastation. Firstly, the two subsections have been unified forming a unique storyline where hydrological response, rainfall-runoff modelling, damage assessment and geomorphic change are integrated. We believe this new version of the MS is bridging the gap between them, because we have introduced several paragraphs especially focused on the predictability of this kind of flash-flood events in order to better join these different issues. In addition, the new Figure 1 is also useful to better understand the integrated approach.

Secondly, we have addressed a qualitative –but also quantitative– discussion about the role played by rainfall intensity and its spatial distribution, complex geology and land cover disturbances, following the suggestion provided by the Anonymous Referee 1#. For this purpose, we have also modified the subsection 2.1. Study area, where a deeper assessment of permeability in lithology materials as well as a diachronic evaluation of land uses evolution and perturbations (i.e., wildfires) is sustaining the discussion on the role of physical parameters generating the flash flood (see Lines 155-165 of the revised MS):

*"The lithology is mainly composed of marls intercalated with limestone (60% of the area) of the Medium-Upper Jurassic*

*(Dogger), dolomites (22% of the area) of the Upper Triassic and Lower Jurassic, and pelagic limestone marls (14% of the area) of the Lower Cretaceous (Fig. 2d). This lithological composition determines the surface water/groundwater interaction. On the one hand, a high degree of fracturing, fissuring and karstification of limestone favours percolation through karstic aquifers. On the other hand, the imperviousness of Dogger and Cretaceous marls (74% of the area) does not allow percolation, enabling runoff generation. The main land use in 2012 was agriculture (58%), mostly located in lowland areas. Forest (26%)*

*and scrubland (17%) were predominant at headwaters. Terraced fields still occupied 10% of the catchment, although most of them were abandoned (Fig. 2e). In 1956, natural vegetation covered 21% of the catchment. This rose to 42% in 2012 due to an afforestation process of former agricultural land in the second half of the twentieth century. In combination with other factors, afforestation triggered a higher fire risk: two wildfires burnt an area of 1.7 km$^2$: 17% in 1983 and 83% in 2011 (Balearic Forestry and Soil Conservation Service, [http://xarxaforestal.caib.es](http://xarxaforestal.caib.es); Fig. 2e)".*

We have also placed special emphasis on the sudden increase in discharge from 120 to 442 m$^3$ s$^{-1}$, which has resulted from the combination of all these physical parameters. Please, see the Lines 583-607 of the revised MS:

*"This runoff response resulted from the combination of rainfall intensity and its spatial distribution, complex geology and land cover disturbances in generating a high $Q_{peak}$ (i.e., 442 m$^3$ s$^{-1}$) with high potential for generating geomorphological changes. Thus, the $Q_{peak}$ unit obtained (i.e., 19 m$^3$ s$^{-1}$ km$^2$) can be classified as the third highest value of all the reported values in Marchi*

*et al. (2010) and the highest of those values obtained from streamflow measurements in a hydrometric station and not by post-event analysis. The hydrologic response analysis in the course of a flash flood shows how storm structure and evolution result in a scale-dependent flood response (Borga et al., 2007). Consequently, spatial rainfall organisation, geology combined with orography and land cover disturbances led to pronounced contrasts in the flood response at the Begura de Salma River. Spatial rainfall on the catchment scale showed that the highest accumulation at the beginning of the storm was located at the*

*headwaters of the catchment (at 15:00 h), whilst during the last part of the event the most important rainfall amounts were*

*located in the downstream part. Examination of the flood response illustrated how the extent and the position of the karst terrain (Zanon et al., 2010) and soil conservation practices (Calsamiglia et al., 2018; Tarolli et al., 2014) provided major geological and anthropogenic control of runoff response. Impervious materials cover 74% of the Begura de Salma River catchment, mostly located at the headwaters, which are responsible for the highest values of topographic torrentiality (Estrany*

*and Grimalt, 2014b), facilitating rapid overland flow generation. During the first part of the storm, when the highest rainfall amounts affected the headwaters, runoff response was delayed by the laminar effect of check-dam terraces massively constructed over Cretaceous marls (Calvo-Cases et al., 2020) and by the predominance of percolation in those areas covered by limestone, mostly in the intermediate parts of the catchment. During the last part of the event, when the highest rainfall intensities were in the downstream part, the excess of soil infiltration capacity and the collapse of headwater check-dam*

*structures triggered the sudden increase in discharge from 120 to 442 $m^3$ $s^{-1}$ in only15 minutes at the hydrometric station. Moreover, the increase of 5 $km^2$ (21% of the catchment area, see more details in section 2.1) of natural vegetation since the 1960s as a result of afforestation processes, increased fuel loads and the risk of wildfires led to 1.7 $km^2$ (7% of the catchment) being burnt since 1980. The removal of vegetation by fires has a similar effect (less interception, less soil storage), which has been experimentally documented after major fires. These factors are a major reason why the history of the steady devastation*

*of plant cover in the Mediterranean is likely to enhance flood risk (Wainwright and Thornes, 2004) and increase desertification tendencies*".

**It also appears that there is quite a disconnect between the results, discussion and conclusion sections, where topics**
**such as driving factors of the damage in urban areas are for the first time explicitly mentioned in the conclusions, while the previous chapters mainly focus on the methodological aspects of the damage assessments. Similarly, language and grammar vary considerably throughout the paper and rigorous copy editing is necessary prior to accepting the manuscript for publication. Given the otherwise interesting and very relevant contribution the paper makes in the field of flash flood post event studies, I recommend considering the manuscript for publication after major revisions.**

We have tried to connct the different sections considering the driving factors of the damage. Likewise, we have unified the language and grammar also developing a deep review by a professional (see attached a certificate).

**Specific comments**
**Structure**
**Introduction**
**The introduction is very technical and has a very narrow focus on flash flood processes. It also appears to address a lot of specific subjects in no particular order rather then leading to the research questions the authors are aiming to**

**answer. Restructuring the introduction so it clearly leads to the research questions and highlights the importance of**
**the work would therefore really improve the quality of the paper. As this is not the first study of its kind, I would also**
**recommend including a literature review on previous post event studies (both flash flood related and potentially other**
**natural hazards) and their findings. This would give the reader the opportunity to better evaluate the contribution of**
**the paper to the scientific discourse and what knowledge gaps it addresses.**

We have modified the Introduction section amplifying the activating hydrological processes during a flash flood, including extra scientific literature on previous post event studies with a special emphasis on those analysing the role played by physical parameters in runoff generation in these catastrophic events (see Lines 48-61 of the revised MS):

"*Characterising the response of a catchment during an extreme flash-flood event is important because it clarifies flood severity and the activating hydrological processes and their dependency on natural and anthropogenic catchment properties (Borga et al., 2007). Numerous studies have tried to determine these driving factors (Braud et al., 2014), in which geological heterogeneities associated with the presence of karst features are crucial in Mediterranean catchments (Vannier et al., 2016; Wainwright and Thornes, 2004). Likewise, flash floods are closely related to land use: the devastation of plant cover in the Mediterranean may increase the risk of flooding because bare soil leads to larger runoff coefficients (Wainwright and Thornes, 2004). However, the limited spatial and temporal scales of flash floods make these events particularly difficult to monitor and document. In the case of rainfall monitoring, the spatial scales of the events are in general much smaller than the sampling potential offered by apparently dense rain networks (Borga et al., 2008; Amponsah et al., 2016). In the case of streamflow monitoring, there is a lack of flash-flood discharge (Q) data from stream gauge observations (Marchi et al., 2010), although Q data are crucial to obtaining representative hydrometric values and characterizing the runoff response of such extreme flash-flood events (Borga et al., 2008). As a result, further field observations and modelling studies are required in order to assess the interdependencies of flash-flood drivers and, thus, better understand and reproduce the active hydrological processes (Sofia and Nikolopoulos, 2020)*".

**Description of the study area**
**For the sake of readability, I would recommend separating the meteorological conditions that lead to the event from**
**the actual description of the study area.**

We have created a new subsection, specifically entitled "2.2 Meteorological context of the 9$^{th}$ October 2018 rain event", being in the revised version of the MS completely separated from the Study area description, as follows:

"*2.2 Meteorological context of the 9$^{th}$ October 2018 rain event*
*The 9th October storm affected the two northernmost catchments of the Llevant County; i.e., the Ca n'Amer and Canyamel Rivers (Fig. 2) with 9 and 4 casualties, respectively, and significant damage. The synoptic situation was like the situations generating flash-flood events in the Western Mediterranean (Fig. 3a). A cut-off low at mid-level was located in the eastern*

*part of the Iberian Peninsula and shallow low-level pressure was affecting the same region, driving warm and wet air from the Mediterranean Sea to the Balearic Islands and the eastern part of the Iberian Peninsula. This occurred in early October, when the sea surface temperature is close to its annual maximum in the Western Mediterranean, providing high quantities of*

*moisture. Moreover, the cut-off low showed a typical divergence at mid-level on its eastern flank, affecting the Balearic Islands and favouring the development of deep convection. Convection started on the sea between the Balearic Islands and the Iberian Peninsula (Figures 3b and 3c) and, due to SW winds at mid-tropospheric levels, the convective cells started to move towards the Balearic Islands, where they triggered the flash-flood event after a heavy rainfall episode (Figures 3d and 3e)".*

**Conclusion**

**The conclusion appears to be quite detached from the rest of the manuscript addressing several points that have not been previously mentioned in the manuscript but are important to fully understand the analysis. For example, how the meteorological, hydrological, geomorphological, damage and risk data analysis are linked. Or what the actual damage**

**driving factor in urban areas are based on the different findings.**

We believe that the modifications on the other sections are now helping to better understand those points that in the first version of the MS have not been previously mentioned. In the case of damage driving factors in the Sant Llorenç des Cardassar village, we have added some paragraphs in the Discussion sections, as follows (see Lines 639-646 of the revised MS):

"*At present, Mallorca does not have any sort of early warning system to assist flood risk management, and nor of course has*

*Sant Llorenç des Cardassar. Similarly, no hydrometeorological early warning was issued by the competent authorities, as the Balearic Islands have no operational hydrological control network releasing real-time information on discharges. In October 2018, Sant Llorenç des Cardassar was one of the four municipalities in Mallorca with a flood risk emergency plan. However, it was not operational at the time the emergency was declared. As a result, the population was completely unaware of how to defend themselves, even during the emergency phase, although Sant Llorenç des Cardassar municipality had significant social*

*vulnerability to floods, as most of the casualties were tourists and the elderly*".

**Rainfall**

**This paper focusses on the hydrological response as a main driver of the flash floods and the authors argue in the**

**introduction that "the uncertainty in hydrological modelling can be large and hydrological models often need to be calibrated [...]. Therefore, the predictability of such events remain low also adding that predictability is lowered by a high non-linearity in the hydrological response related to threshold effects". This implies that the uncertainty in the hydrological models are a key barrier in the predictability of flash floods. However, most other studies on flash floods and flash flood early warning systems find the spatio-temporal uncertainties in the rainfall prediction to be the largest**

**obstacle in accurately forecasting and modelling flash floods (see for example Alfieri et al. 2017). This issue is also**
**addressed in the description of the rainfall data, but the authors do not report to what extend the results of the subsequent hydrological and hydraulic models are sensitive to the uncertainties of the rainfall input. Therefore, I would recommend adding a short sensitivity analysis in regard to the rainfall input to the discussion section. It would also be interesting to see to what extend the results vary between the radar and gauge data.**

We thank the reviewer on this very pertinent and interesting comment. We fully agree that the spatio-temporal uncertainty in the rainfall data is the main source of uncertainty in flash floods, an issue also commented by the Anonymous Referee #1. We recognize that the text in our manuscript was misleading and we have accordingly modified the Introduction section (see Lines 67-69 of the revised MS):

"*The main source of uncertainty is related to the spatio-temporal scales of rainfall pattern. The forecasting of intense*
*thunderstorms by numerical weather prediction systems to provide accurate rainfall information is particularly challenging (Alfieri et al., 2015; Collier, 2007)*".

Otherwise, the scientific issue aroused from this comment is very challenging. We believe that analysing the uncertainty sources of the hydrological model in this particular paper is out of scope, would mislead the reader also considering this topic would deserve a paper on its own. However, we have specifically minimized this uncertainty source by using a spatially
interpolated rainfall from gauging stations in order to understand –from a modelling point of view– the processes leading to this event. It is clear that the estimated radar rainfall data used in our piece of work have provided a better spatial interpolation of the rainfall event than using only the radar data (see Fig. S1 and the Lines 232-234 in the subsection *2.3. Rainfall data* of the revised MS). Moreover, the radar data was adjusted with data from San Llorenç des Cardassar station as described in this same supplementary figure (Fig. SF1) so that indirectly the rainfall data is already used in the model. Definitely, it would be
interesting to further analyse the results of the uncalibrated radar data in these kind of extreme rain storms but it should be a future scientific work.

**Risk management and early warning**
**Given the high casualties and damage during this event it would be important to also cover the vulnerability of assets**
**and people in the case study area for a comprehensive analysis of the damaging factors. This aspect however is only very briefly mentioned in the discussion and conclusion. Key questions would include: did people in the village receive some sort of early warning? Are their any risk management strategies in place apart from the mentioned flood zones? Discussing these aspects would also help to conclude with more specific recommendations for the improvement of risk management practises.**

Nowadays, Mallorca Island does not have implemented any sort of early warning system to flood risk management, nor has the Sant Llorenç des Cardassar village. However, the Spanish Meteorological Agency (AEMET) applying the National Plan for the Observation of Adverse Meteorological Phenomena ("Meteoalerta"; http://www.aemet.es/en/lineas_de_interes/meteoalerta), forecasted storms and intense rains. On 9[th] October 2018 at 01:58 a.m., AEMET issued a yellow warning for rainfall amounts up to 20 mm in one hour in Balearic Islands, extending the duration of this warning until 12:00 p.m. on 10[th] October. At 6:53 p.m. on 9[th] October, the warning was raised to the orange level, 50 mm in one hour, in the east of Mallorca and in the north and northeast of Mallorca, and finally, at 10:07 p.m., reaching the red level, 220 mm in one hour, in the easternmost part of the north of Mallorca, being extended until 02:00 a.m. on 10[th] October. These different warnings clearly demonstrated that the storm was not well meteorological forecasted, due to both orange and red warnings were delivered when the disaster had already occurred.

Likewise, no hydrometeorological early warning was issued by the competent authorities, also considering that the Balearic Islands do not have both an operational hydrological control network and real-time information on rainfall intensity through automatic weather stations in the headwaters of Begura de Salma River catchment.

The municipality of Sant Llorenç des Cardassar was one of the four municipalities in Mallorca with municipal flood risk emergency plan, implemented previously to the catastrophe. However, it was not operational at the time the emergency was declared. The population was completely unaware of how to defend preventively themselves, even during the emergency phase.

Regarding the social vulnerability, the Sant Llorenç des Cardassar municipality has 8,406 inhabitants, being 23% foreigners (IBESTAT, 2018). In addition, a very high tourist capacity including dispersed rural vacation homes; as well ageing population is > 16%. All these variables increase the social vulnerability, a fact that has been proven due to the most of the casualties were tourists and ageing population during the catastrophe. As a result, any measure to improve social vulnerability should be focused on increasing the knowledge of natural risks by foreign resident population and tourists.

Almost two years after the catastrophe, its consequences have been applied as learnt lessons in preventing the risk. The City Council applying several actions (https://www.santllorenc.es/ca/noticies-ajuntament/lajuntament-de-sant-llorenc-presenta-una-modificacio-puntual-del-seu-planejament-urbanistic-despres-de-la-torrentada-en-vista-a-millorar-la-seguretat-) such as significant modifications within the Urban Planning for reducing the exposed areas to flood risk in urbanized areas, multidisciplinary studies to improve knowledge of the population exposure level, and regulations to reduce the exposure of assets in the case of new flash floods. In addition, works on hydraulic infrastructures focused to restore the previous condition or even lowering the channel roughness, involving a risk increase. Despite population has more information on how to deal with possible new events, the implementation of a hydrometeorological alert system would be more effective, facilitating the monitoring of potential flash floods.

We have refocused the Discussion section with several paragraphs in the Discussion section, as we have previously pointed out.

**Damage classes**

**In Figure 7(e), the distribution of the damage classes for the three different zones and the total of all zones are shown. It seems that the total does not correspond to the sum of the three zones as the by far largest group in total are houses being "Damaged & Non habitable" with 260 houses, while the sum in this group for all three zones is 37 homes. That might be either an error or it should clearly be stated what is meant by "Total".**

We must apologize by this mindless error caused by a mixing of cells in the Excel spreadsheet, although the explanation within the main text was correct. Certainly, "Damaged & Non habitable" summed 37 for all three zones, not 260 houses. This 260 houses are "Damaged and habitable". We have pertinently modified in the new Figure 8e.

**Sediment connectivity and geomorphic change**

**While using the sediment connectivity to support search and rescue missions after flash flood events is a very innovative approach, it is not entirely clear what one can learn from the sediment volume calculation. Discussing this number in the context of the other analysis and its implications for a better understanding of the flash flood processes would help to further improve the manuscript. It would also be interesting to learn what is the accuracy of the mentioned approach given the different spatial resolutions and accuracies between the 2014 and 2018 surface models. Can changes in volume**

**attributed to this specific event or does this number also include other changes to the geomorphology (both human and natural) that happened between 2014 and 2018? I would also recommend to clearly separate the sediment connectivity analysis that was used to support the search and rescue efforts and the geomorphic change detection to make clear that the two analysis had different aims.**

We must thank again to the referee for these relevant and appropriate questions, allowing an improvement of one of the key issues of our piece of work.

In order to provide the necessary clearness to the readers about what are the real aims from both methods, we have split the methods subsection *2.6 Sediment connectivity and geomorphic change detection*, being currently *2.6 Sediment connectivity* and *2.7 Geomorphic change detection*. In addition, both methods in the new Figure 1 are also clearly separated, although also emphasizing their own relationship.

We have modified the following sentence (see Lines 328-329 of the previous MS version):

"*Firstly, and taking into consideration the emergency situation, the index of (water and sediment) connectivity at the catchment scale was applied to find out the areas with the greatest sediment deposition potential where victims could have been buried by the flash flood*".

This is the new sentence (see Lines 351-353 of the revised MS), in which is better explaining the main aim of the IC

implementation in terms of Emergency rescue tasks:

"*Firstly, and taking into account the emergency situation, the index of (water and sediment) connectivity at the catchment scale was applied to find the areas with the greatest sediment deposition potential, which were where victims could have been buried by the flash flood*".

Within the new subsection *2.8 Geomorphic change detection*, we have reinforced its main purpose in the first paragraph, as follows (see Lines 367-372 of the revised MS):

"*HR-DEMs facilitate the improvement of sediment connectivity as a powerful tool to determine preferential flow-paths and those areas with the greatest potential sediment deposition. The evaluation of the flash-flood landform signature by UAVs is the second part of creating a tool for a rapid response of post-catastrophe search and rescue tasks by applying hydrogeomorphological precision techniques. The estimation of overbank sedimentation allowed the calibration of the predicted large sedimentation by IC mapping and its reliability in detecting sites where victims might be buried by flood sediment*".

In order to clarify that no changes in volume occurred between 2014 and 2018, we have added a new sentence in the new subsection 2.8 (see Lines 396-398 of the revised MS):

"*It is worth noting that no geomorphic changes were observed between 2014 and October 2018 by photointerpretation of aerial imagery (PNOA, 2015) and the continuous measurement of water stages since January 2015, with no overbanked flood events*".

Finally, the accuracy of the approach was settled, as mentioned in the first MS version, by an assessment of RMSE in xyz. However, we have reinforced this method adding "*and located on surfaces not modified by the flash flood*".

**Additional comments**

**As mentioned earlier, the manuscript would benefit from English language copy editing. Instead of giving point-by-point corrections I would like to provide a a few examples, which I find difficult to understand:**

The authors we are grateful with detailed suggestions on English language and grammar. Following this advice, we have requested an external review on English language by a professional native speaker (see attached the certificate).

**L 51f: "Characterising the response of a catchment during flash flood events is important because elucidate the hydrological processes from an extreme flood and their dependency on catchment properties and flood severity (Borga et al., 2007)" should probably be: "Characterising the response of a catchment during flash flood events is important because it helps understanding the hydrological processes of extreme floods and their dependency in regard to the properties of the catchment and the severity of the event."**

The Anonymous Referee #1 had also suggested a change in English language style within the sentence.

**L 112f: "[…] was developed affording the analysis of the rainfall-runoff processes at small spatial scale during this extreme event." I did not understand what "affording" means in this context.**

We have changed "affording" by "through".

**L125: "high-energy environment" I did not understand what "high-energy" means in this context**

We believe that "high-energy environment" is a concept completely accepted in the argot of Earth sciences and is comprehensible in the context developed within this sentence. However, we have modified the beginning of the sentence including the geographical context: "*In such a high-energy environment,…*".

**L156: "under a recurrent affection of wildfires": does that mean that these areas are regularly affected by wildfires or that these areas are prone to wildfires?**

These areas are both prone to wildfires and have been affected by wildfires twice in the last 30 years. In addition, one of the
main concerns provided by both referees are related with the physical processes conditioning the hydrological response of the flash flood event. In this way, we have improved the description of the subsection 2.1. Study area in order to better contextualize the influence of lithology, land uses and wildfires in the hydrological response during the extreme flash flood event (see Lines 155-165 of the revised MS):

"*The lithology is mainly composed of marls intercalated with limestone (60% of the area) of the Medium-Upper Jurassic*
*(Dogger), dolomites (22% of the area) of the Upper Triassic and Lower Jurassic, and pelagic limestone marls (14% of the area) of the Lower Cretaceous (Fig. 2d). This lithological composition determines the surface water/groundwater interaction. On the one hand, a high degree of fracturing, fissuring and karstification of limestone favours percolation through karstic aquifers. On the other hand, the imperviousness of Dogger and Cretaceous marls (74% of the area) does not allow percolation, enabling runoff generation. The main land use in 2012 was agriculture (58%), mostly located in lowland areas. Forest (26%)*
*and scrubland (17%) were predominant at headwaters. Terraced fields still occupied 10% of the catchment, although most of them were abandoned (Fig. 2e). In 1956, natural vegetation covered 21% of the catchment. This rose to 42% in 2012 due to an afforestation process of former agricultural land in the second half of the twentieth century. In combination with other factors, afforestation triggered a higher fire risk: two wildfires burnt an area of 1.7 km$^2$: 17% in 1983 and 83% in 2011 (Balearic Forestry and Soil Conservation Service, http://xarxaforestal.caib.es; Fig. 2e)*".

**To whom it may concern**

I certify that I have revised and corrected the grammar, structure and style of the following article:

**Title**:

**Hydrogeomorphological analysis and modelling for a comprehensive understanding of flash-flood damaging processes: the 9th October 2018 event in northeastern Mallorca**
**Authored by**:

Joan Estrany, Maurici Ruiz-Pérez, Raphael Mutzner, Josep Fortesa, Beatriz Nácher-Rodríguez, Miquel Tomàs-Burguera, Julián García-Comendador, Xavier Peña, Adolfo Calvo-Cases, Francisco J. Vallés-Morán

Michael Eaude
Torrent de Can Mariner 7-1
Barcelona 08031
Spain
NIF: X-01117301-F

Signed May 18, 2020:

**MS Records**

> **nhess-2019-304**   Submitted on 13 Sep 2019
>
> **Hydrogeomorphological analysis and modelling for a comprehensive understanding of flash-flood damaging processes: The 9th October 2018 event in North-eastern Mallorca**
>
> Joan Estrany, Maurici Ruiz-Pérez, Raphael Mutzner, Josep Fortesa, Beatriz Nácher-Rodríguez, Miquel Tomàs-Burguera, Julián García-Comendador, Xavier Peña, Adolfo Calvo-Cases, and Francisco J. Vallés-Morán
>
> First Contact: Joan Estrany Bertos
>
> Agreed licence: Creative Commons Attribution 4.0 International
>
> Handling Editor: Kai Schröter
> Manuscript Type: Research article
>
> **Status: File Upload (NHESS)    Iteration: Major Revision**

**Interactive Discussion**

**Editor Decision: Reconsider after major revisions (further review by editor and referees)** (27 May 2020) by Kai Schröter

Comments to the Author:

Response of the Authors to the Editor Comments (in blue):

Comments to the Author:

Dear Joan Estrany and co-authors,

Thank you very much again for submitting your brief communication manuscript 'Hydrogeomorphological analysis and modeling for a comprehensive understanding of flash-flood damaging processes: The 9th October 2018 event in North-eastern Mallorca'. The authors we kindly appreciate the comments from the Associate Editor as well as from the two anonymous referees which have helped to deeply improve the MS.

Two referee reports and a short comment from the scientific community are available which raises important issues with the current version of your manuscript. Major concerns exist regarding the links of individual aspects addressed in the study (meteorological, hydrological, geomorphological, damage, and risk data analysis) which appear to be considered rather separately. In addition, very detailed suggestions are made by the reviewers on how to improve the comprehensibility of your study and achieve a more clear structure. Reading your responses to these comments I am positive that a revised version of your manuscript will address these points appropriately.

We have addressed all the major concerns highlighted by both reviewers and also integrated constructive ideas that –we believe- have improved the comprehensibility of the MS with a clearer structure. We are also very pleased for the positive reaction of the Associate Editor to our responses to the referees' comments.

The main concern based on the lack of continuity in the narrative –taking into account that we had applied an integrated approach with meteorological, hydrological, geomorphological, damage and risk data analysis– of the paper has been addressed creating an overview figure showing the links between data and models, the new Figure 1. This Figure 1 has also been improved following one of the comments of the Associate Editor.

The second idea provided by the Anonymous Referees for addressing a more integrated comprehension has been focused in merging the two subsections of the Discussion section to facilitate the combination of the individual results. As a result, the discussion on hydrological response, rainfall-runoff modelling, damage assessment and geomorphic change are integrated in a unique section.

Also in this same Discussion section, we have also introduced several paragraphs especially focused on the predictability of this kind of flash-flood events and developed a qualitative –but also quantitative– discussion about the role played by rainfall intensity and its spatial distribution, complex geology and land cover disturbances, following the suggestion provided by the Anonymous Referee 1#. For this purpose, we have also modified the subsection 2.1. Study area, where a deeper assessment of permeability in lithology materials as well as a diachronic evaluation of land uses evolution and perturbations (i.e., wildfires) is sustaining the discussion on the role of physical parameters generating the flash flood.

Another main modifications have been performed in the Introduction section, where we have amplified the activating hydrological processes during a flash flood, including extra scientific literature on previous post-event studies with a special emphasis on those analysing the role played by physical parameters in runoff generation.

In addition please make sure that legends and other information given in your Figures are readable (e.g. new Figure 1 with the workflow and Figure 8 (previously Figure 7) are not readable).
We have modified the new Figure 1 slightly changing its structure and also increasing the size of the text to provide more clarity. In addition, the text size of the legend and other information have been also increased in the Figure 8.

Please also note that this decision for major revisions does not necessarily imply acceptance of the manuscript in the journal NHESS, and it still depends on your reply and edits to your manuscript, as well as on the reviewer comments of the revised version. As a next step, I kindly ask you to provide a revised marked up (track changes) version of your manuscript to make clear how you include the changes in response to both referee reports and the short comment. Additionally, you have to upload the new

We have submitted the revised version of our MS and also the supplementary information with tracking changes.

[revised manuscript text omitted]
 9[th] October 2018 violent flash-flood  in  Llevant County of Mallorca. Source: Pol (2019).

| Damage level | Flood risk cartographyies | | | | | | Copernicus Emergency Management System | | Total |
|---|---|---|---|---|---|---|---|---|---|
| | *10 years* | *%* | *100 years* | *%* | *500 years* | *%* | *Affected area* | *%* | |
| **COLLAPSED** | 5 | 50 | 8 | 80 | 9 | 90 | 10 | 100 | 10 |
| **DAMAGED & HABITABLE** | 52 | 20 | 107 | 41 | 141 | 54 | 225 | 86 | 261 |
| **DAMAGED & NOT HABITABLE** | 15 | 41 | 26 | 70 | 31 | 84 | 36 | 97 | 37 |
| **DAMAGED PLOT** | 6 | 35 | 9 | 53 | 9 | 53 | 13 | 76 | 17 |
| **DAMAGED & RESTRINCTEDGE D USE** | 19 | 28 | 39 | 58 | 47 | 70 | 63 | 94 | 67 |
| **TOTAL** | 97 | 25 | 189 | 48 | 237 | 60 | 347 | 89 | 392 |

**Table 3** Damaged buildings in the village of Sant Llorenç des Cardassar caused by the violent flash-flood oin 9[th] October 2018 and those encompassed in the official flood risk maps for 10, 100 and 500 years recurrence periods.

**Figures and Cayptions**

[Figure]

**Figure 1** Workflow of the experimental design.

[Figure]

**Figure 2** Main characteristics of affected basins during the 9th October 2018 flash-flood. (a) Location of Mallorca in the western Mediterranean. (b) Topography and fluvial network of Mallorca  with the location of the  main basins affected: Canyamel and Ca n'Amer rivers. (c) Blanquera and Begura de Salma headwater river catchments within the Ca n'Amer River, with location of rainfall and hydrometric stations, and radar rainfall points derived from a regular mesh of 1x1 km. Source: https://opendata.aemet.es. Background: aerial photography and DEM data (PNOA, 2015). (d) Lithology of both Blanquera and Begura de Salma catchments. (e) Land uses and terraced areas for the same headwater catchments. Source: Corine Land Cover (2018).

[Figure]

**Figure 3**2 (a) Surface pressure and 500-hPa height analyses at 1200 UTC on 9ᵗʰ October 2018 Source: **http://wetter3.de**; i.e. , at the beginning of the precipitation event. Satellite image at (b) 12.00UTC and (d) 15.00UTC Source: http://www.sat24.com. EUMETSAT and radar images at the same hours (c and e) Source: http://www.aemet.es.

[Figure]

**Figure 43** (a) Aerial view of the concrete channeling of the Begura de SaumàSalma River of concrete channelization that crossinges Sant Llorenç des Cardassar village and the location of bridges. (b) Detailed aerial view of the very beginning of this concrete channeling.zation where the hydrometric station is located. The photographs show a view of the Bridge 1 from the hydrometric station when (the right pictured) was installed the digital equipment was installed, on 10th June 2015 and (the left onee) a few hours after the flash flood, on the 10th October 2018. Background: aerial photography and DEM data (PNOA, 2015).

[Figure]

**Figure 54** Stage-discharge rating curve performed by means of two-dimensional hydraulic modelling with two differentiated sections in according withto the influence of the Bridge 1 (see Figure 3a) and its potential obstruction.

[Figure]

| Hydrological year | Rainfall (mm) | Runoff (mm) | Qmax (m³ s⁻¹) | Qavg (m³ s⁻¹) |
|---|---|---|---|---|
| 2014-15 | 823 | 2.0 | 2.8 | 0.002 |
| 2015-16 | 363 | 0.1 | 0.2 | 0.000 |
| 2016-17 | 825 | 12.9 | 4.8 | 0.009 |
| 2017-18 | 763 | 0.8 | 0.4 | 0.001 |
| 2018-19 (until April) | 500 | 91.8 | 442.0 | 0.118 |
| Average study period (2014-2018) | 655 ± 211 | 21.5 | --- | 0.026 ± 0.1 |
| Average long term (1968-2018) | 652 ± 176 | --- | --- | --- |

**Figure 56** Discharge at 15-min intervals measured atin the MEDhyCON hydrometric station located at the beginning of the concrete channelizationg of the Begura de SaumàSalma River in Sant Llorenç des Cardassar. Likewise, the daily rainfall measured at the AEMET-B630 Ses Pastores during the monitored period (10$^{th}$ January 2015-30$^{th}$ September 2018), priorevious to the catastrophic flash flood of 9$^{th}$ October 2018. Bottom set table: Rainfall, runoff and peak discharge for hydrological years during study period. Rainfall data is are from AEMET-B630 Ses Pastores, located 10.5 km from the Begura de SaumàSalma catchment outlet and representative of the rainfall dynamics of the Llevant Ranges headwater parts.

[Figure]

**Figure 6** Map of isohyets of the rain storm  of 9th October 2018 in the two headwater catchments of the Ca n'Amer River; i.e. the Blanquera and Begura de Salma rivers. Source: 10-minute radar images obtained from the web https://opendata.aemet.es/. The inset figure illustrates the observed discharge measured at the MEDhyCON hydrometric station as well as the result of the rainfall-runoff simulation using a modified version of the GR3 model. Background: aerial photography and DEM data (PNOA, 2015).

[Figure]

**Figure 8̶7** (a) Map of the damage level classification of buildings and water stage reached in the different affected zones at Sant Llorenç des Cardassar according to the Balearic Islands Autonomous Government in comparison with the flood delimitation c̶a̶r̶r̶i̶e̶d̶ ̶o̶u̶t̶ by Copernicus EMS. Flow direction and hydrological connectivity in the affected zones (b) 1,̶ ̶a̶n̶d̶ 2,̶ ̶a̶s̶ ̶w̶e̶l̶l̶ ̶a̶s̶ ̶(̶c̶)̶and 3 in the Sant Llorenç des Cardassar urban network. (c̶d) Economic activities at building scale in the urban area of Sant Llorenç des Cardassar and the delimitation of affected zones by the flash-flood. (d̶e) Damage level classification of buildings in the different affected zones at Sant Llorenç des Cardassar. (̶f̶)̶ ̶D̶a̶m̶a̶g̶e̶ ̶l̶e̶v̶e̶l̶ ̶o̶f̶ ̶t̶h̶e̶ ̶b̶u̶i̶l̶d̶i̶n̶g̶s̶ ̶a̶n̶d̶ ̶p̶l̶o̶t̶s̶ ̶b̶y̶ ̶z̶o̶n̶e̶s̶.̶ (e̶f) Official flood risk maps and flood delimitation by Copernicus EMS at the Sant Llorenç des Cardassar village with the location of buildings affected b̶u̶i̶l̶d̶i̶n̶g̶s̶ by the flash -̶flood o̶c̶c̶u̶r̶r̶e̶d̶ of 9th October 2018. Background: aerial photography (PNOA, 2015). In Fig. 7d, the source of land uses at the urban plot scale is the General Directorate for the Cadastre (http://www.sedecatastro.gob.es/).

[revised manuscript text omitted]

Figure S32 Scheme of the modified infiltration model including a karstic component. Only the relevant parameters are shown, being $Q_{GR}$: outflow from the infiltration reservoir; $H_{GR}$: level in infiltration reservoir (State variable); $H_{GR\_threshold}$: height of the onset of karstic behaviour; $H_{GR\_max}$: Maximum height of infiltration reservoir $i_{karst}$: karstic release coefficient; ETR: Evapotranspiration; $i_{Inf}$: Infiltration intensity; $i_{Net\ to\ SWIMM}$: surface runoff intensity; P,T: Precipitation, Temperature

---

## Author Response (AR2)

**Hydrogeomorphological analysis and modelling for a comprehensive understanding of flash-flood damageing processes: tThe 9th October 2018 event in North-eastern Mallorca**

Joan Estrany[1,2*], Maurici Ruiz-Pérez[1,2,3], Raphael Mutzner[4], Josep Fortesa[1,2], Beatriz Nácher-Rodríguez[5], Miquel Tomàs-Burguera[6], Julián García-Comendador[1,2], Xavier Peña[4], Adolfo Calvo-Cases[7], Francisco J. Vallés-Morán[5]

[1]Mediterranean Ecogeomorphological and Hydrological Connectivity Research Team (http://medhycon.uib.cat), Department of Geography, University of the Balearic Islands, Carretera de Valldemossa Km 7.5, 07122 Palma, Balearic Islands, Spain

[2]Institute of Agro-Environmental and Water Economy Research –INAGEA, University of the Balearic Islands, Carretera de Valldemossa Km 7.5, 07122, Palma, Balearic Islands, Spain

[3]Service of GIS and Remote Sensing, University of the Balearic Islands, 07122 Palma, Balearic Islands, Spain

[4]Hydrique Engineers (http://www.hydrique.ch), Le Mont sur Lausanne, Vaud 1052, Switzerland

[5]Universitat Politècnica de València, Camí de Vera, s/n, València, 46022, Spain

[6]Estación Experimental de Aula Dei (EEAD-CSIC), Avenida Montañana, 1005, 50059 Zaragoza, Spain

[7]Inter-University Institute for Local Development (IIDL) Department of Geography, University of Valencia, Av. Blasco Ibáñez 28, 46010, Valencia, Spain

*Correspondence to: Joan Estrany (joan.estrany@uib.cat)

**Abstract.** A flash-flood event hit the north-eastern part of Mallorca Island on 9th October 2018, causing 13 casualties. Mallorca is prone to catastrophic flash floods acting on a scenario of deep landscape transformation caused by Mediterranean tourist resorts. As global change may exacerbate devastating flash floods, analyses of catastrophic events are crucial to support effective prevention and mitigation measures. Field-based, remote-sensing and modelling techniques were used in this study to evaluate

rainfall-runoff processes at the catchment scale linked to hydrological modelling. Continuous streamflow monitoring data revealed a peak discharge of 442 m$^3$ s$^{-1}$ with an unprecedented runoff response. This exceptional behaviour triggered the natural disaster as a combination of heavy rainfall (249 mm in 10 h), karstic features and land cover disturbances in the Begura de Salma River catchment (23 km$^2$). Topography-based connectivity indices and geomorphic change detection were used as rapid post-catastrophe decision-making tools, playing a key role during the rescue search. These hydrogeomorphological precision techniques were combined with the Copernicus Emergency Management Service and 'ground-based' damage assessment, which showed very accurately the damage-driving factors in the village of Sant Llorenç des Cardassar. The main challenges in the future are to readapt hydrological modelling to global change scenarios, implement an early flash-flood warning system and take adaptive and resilient measures on the catchment scale.

**1 Introduction**

Flash floods are related  to high-intensity precipitation, mainly  convective in origin and with  restricted spatio-temporal occurrence. For this reason, they usually strike basins <1000 km$^2$ with response times of a few hours or less. These spatial and temporal dimensions of flash-flood events are directly linked to (1) geomorphometric characteristics of the catchments and (2) activation mechanisms of runoff as a combination of intense rainfall, soil moisture and soil hydraulic properties (Versini et al., 2013). Land use modification, urbanization and recurrent wildfires can alter these activation mechanisms and the potential for flash-flood casualties and damage. In Europe, 40% of flood-related casualties in the period 1950–2006 are due to flash floods (Barredo, 2007). However, catastrophic flash-floods are much more frequent in some parts of the Mediterranean region than in the rest of Europe, due to the interaction between geomorphology, climate, vegetation, and the warm sea surface (Cassola et al., 2016), all combining to create a flood-prone environment. The abrupt reliefs surrounding the Mediterranean Sea are very close to the coastline, shaping small and torrential catchments where the convergence of low-level atmospheric flows and the uplift of warm wet air masses drifting from the Mediterranean Sea to the coast generates heavy downpours in very short time-spans (Gaume et al., 2009).

Characterising the response of a catchment during an extreme flash-flood event is important because it

55   clarifies flood severity and the activating hydrological processes and their dependency on natural and anthropogenic catchment properties (Borga et al., 2007). Numerous studies have tried to determine these driving factors (Braud et al., 2014), for which geological heterogeneity associated with the presence of karst features is crucial in Mediterranean catchments (Vannier et al., 2016; Wainwright and Thornes, 2004). Likewise, flash floods are closely related to land use: the devastation of plant cover in the

60   Mediterranean may increase the risk of flooding because bare soil leads to larger runoff coefficients (Wainwright and Thornes, 2004). However, the limited spatial and temporal scales of flash- floods make these events particularly difficult to monitor and document. In the case of rainfall monitoring, the spatial scales of the events are in general much smaller than the sampling potential offered by apparently dense rain networks (Borga et al., 2008; Amponsah et al., 2016). In the case of streamflow monitoring, there is

65   a lack of flash- flood discharge ($Q$) data from stream gauge observations (Marchi et al., 2010), although $Q$ data are crucial to obtaining representative hydrometric values and characterising the runoff response of such extreme flash-flood events (Borga et al., 2008). As a result, further field observations and modelling studies are required in order to assess the interdependencies of flash-flood drivers and, thus,  to understand better and reproduce the active hydrological processes (Sofia and Nikolopoulos,

70   2020).

Earlier flash- flood forecasting systems were based on the Flash Flood Guidance (Georgakakos, 1986), which calculated the a- priori amount of rainfall needed to trigger specific $Q$ at the outlet of a catchment, depending on prior wetness conditions. At present, semi-distributed or distributed hydrological models are more widely used for such forecasting purposes (Artinyan et al., 2016; Gourley et al., 2010; Miao et

75   al., 2016; Nguyen et al., 2016), whilst probabilistic and ensemble modelling assess the uncertainty of flash- flood forecasting systems (Hardy et al., 2016). However,  uncertainty in hydrological modelling may be great. The main source of uncertainty  relates to the spatio-temporal scales of the 
[revised manuscript text omitted]
 channel that takes the river through the village. C,losing a drainage basin of 23 km$^2$, it is located ca. 50 m upstream from the first of the five bridges that cross the river within the village (see Bridge 1 in Fig. 4b). At the cross-section where the hydrometric station is located, the channel bed is 70.25 m.a.s.l., whilst the top of its bank channels are is 72.00 m.a.s.l.

The transformation of water stage (hereinafter WS; m) to $Q$ (m$^3$ s$^{-1}$) through the stage/discharge rating curves (hereinafter SDRC) into the qualitative range (from low to high $Q$ conditions) is broadly developed by power and  polynomial equations, characterised by physical-based parameters.  In the absence of direct flow measurements for  $
[revised manuscript text omitted]
 used to simulate catchment runoff response are often unsuccessful for Mediterranean-climate catchments due to  their 
[revised manuscript text omitted]